# On diffusion-based generative models and their error bounds: The log-concave case with full convergence estimates

**Stefano Bruno**                                                             *sbruno@ed.ac.uk*
*School of Mathematics*
*University of Edinburgh*

**Ying Zhang**                                                  *yingzhang@hkust-gz.edu.cn*
*Fintech Thrust*
*The Hong Kong University of Science and Technology (Guangzhou)*

**Dong-Young Lim**                                                         *dlim@unist.ac.kr*
*Department of Industrial Engineering*
*Artificial Intelligence Graduate School*
*Ulsan National Institute of Science and Technology*

**Ömer Deniz Akyildiz**                                      *deniz.akyildiz@imperial.ac.uk*
*Department of Mathematics*
*Imperial College London*

**Sotirios Sabanis**                                                      *s.sabanis@ed.ac.uk*
*School of Mathematics*
*University of Edinburgh*
*The Alan Turing Institute*
*National Technical University of Athens*

**Reviewed on OpenReview:** *https://openreview.net/forum?id=zjxKrb4ehr*

## Abstract

We provide full theoretical guarantees for the convergence behaviour of diffusion-based generative models under the assumption of strongly log-concave data distributions while our approximating class of functions used for score estimation is made of Lipschitz continuous functions avoiding any Lipschitzness assumption on the score function. We demonstrate via a motivating example, sampling from a Gaussian distribution with unknown mean, the powerfulness of our approach. In this case, explicit estimates are provided for the associated optimization problem, i.e. score approximation, while these are combined with the corresponding sampling estimates. As a result, we obtain the best known upper bound estimates in terms of key quantities of interest, such as the dimension and rates of convergence, for the Wasserstein-2 distance between the data distribution (Gaussian with unknown mean) and our sampling algorithm. Beyond the motivating example and in order to allow for the use of a diverse range of stochastic optimizers, we present our results using an $L^2$-accurate score estimation assumption, which crucially is formed under an expectation with respect to the stochastic optimizer and our novel auxiliary process that uses only known information. This approach yields the best known convergence rate for our sampling algorithm.

## 1 Introduction

Diffusion-based generative models, also known as score-based generative models (SGMs) (Song & Ermon, 2019; Song et al., 2021; Sohl-Dickstein et al., 2015; Ho et al., 2020), have become over the past few years one of the most popular approaches in generative modelling due to their empirical successes in data generation

tasks. These models have achieved state-of-the-art results in image generation (Dhariwal & Nichol, 2021; Rombach et al., 2022), audio generation (Kong et al., 2020) and inverse problems (Chung et al., 2022; Song et al., 2022; Cardoso et al., 2024; Boys et al., 2024) outperforming other generative models like generative adversarial networks (GANs) (Goodfellow et al., 2014), variational autoencoders (VAEs) (Kingma & Welling, 2014), normalizing flows (Rezende & Mohamed, 2015) and energy-based methods (Zhao et al., 2017).

SGMs generate approximate data samples from high-dimensional data distributions by combining two diffusion processes, a forward and a backward in time process. The former process is used to iteratively and smoothly transform samples from the unknown data distribution into (Gaussian) noise, while the associated backward in time process reverses the noising procedure and generates new samples from the starting unknown data distribution. A key role in these models is played by the score function, i.e. the gradient of the log-density of the solution of the forward process, which appears in the drift of the stochastic differential equation (SDE) associated to the backward process. Since this quantity depends on the unknown data distribution, an estimator of the score has to be constructed during the noising step using score-matching techniques (Hyvärinen, 2005; Vincent, 2011). These techniques have the advantage of not suffering from known problems of traditional pushforward generative models, such as mode collapse (Salmona et al., 2022).

The widespread applicability and success of SGMs have been accompanied by a growing interest in the theoretical understandings of these models, particularly in their convergence theory e.g. in Block et al. (2020); De Bortoli et al. (2021); Bortoli (2022); Lee et al. (2022); Yang & Wibisono (2022); Kwon et al. (2022); Liu et al. (2022); Oko et al. (2023); Lee et al. (2023); Chen et al. (2023a;d); Li et al. (2024); Pedrotti et al. (2024); Conforti et al. (2025); Benton et al. (2024), with further works appearing after the first version of our preprint e.g. in Tang & Zhao (2024); Strasman et al. (2025); Mimikos-Stamatopoulos et al. (2024). At its core, this new generative modeling approach combines optimisation and sampling procedures – specifically, the approximation of the score and the creation of new samples, which make its theoretical analysis both an interesting and a rich challenge.

Some of the recent advances in the study of the theoretical properties of SGMs concentrate around the sampling procedure by assuming suitable control for the score estimation procedure. For instance, the analysis in Lee et al. (2022); Chen et al. (2023d) assumes that the score estimate is $L^2$-*accurate*, meaning that the $L^2$ error between the score and its estimate is small, and provides estimates in total variation (TV) distance. Under the same assumption, the more recent contribution in Conforti et al. (2025) establishes non-asymptotic bounds in Kullback Leibler (KL) divergence by assuming finite relative Fisher of data distribution with respect to a Gaussian distribution.

The main drawback of the aforementioned $L^2$-accurate (and in some cases $L^\infty$-accurate) score estimation assumption is that the corresponding expectation is given with respect to density of the solution of the forward process, which depends on the unknown data distribution.

Our approach introduces a novel auxiliary process that relies solely on known information and uses the density of the solution of this process in the $L^2$-accurate score estimation. To further highlight the powerfulness of our approach, we present a motivating example on the case of sampling from a Gaussian distribution with unknown mean. Full theoretical estimates for the convergence properties of the SGM are provided in Wasserstein-2 distance, while our choice of stochastic optimizer for the score approximation is a simple Langevin-based algorithm. Our estimates are explicit and contain the best known optimal dependencies in terms of dimension and rate of convergence (Theorem 1). To the best of the authors' knowledge, these are the first such explicit results with transparent dependence on the parameters involved in the sampling and optimization combined procedures of the diffusion models. By connecting the diffusion models with the theoretical guarantees of machine learning optimizers via standard stochastic calculus tools, the results in Theorem 1, together with the bounds achieved in the more general setting in Theorem 10, provide the theoretical justification for the empirical success of the diffusion models.

We close this introductory section by highlighting some other, alternative approaches which were recently developed. One may consult Yang & Wibisono (2023) for an approach based on the assumption that the score estimation error has a sub-Gaussian tail. This is a stronger assumption than $L^2$-*accurate*. In Chen et al. (2023b), non-asymptotic bounds in TV and Wasserstein distance of order 2 are derived when the data distribution is supported on a low-dimensional linear subspace. Finally, convergence guarantees in TV were

developed in Chen et al. (2023c) for the probability flow ordinary differential equation (ODE) implementation of SGMs under the $L^2$-accurate score estimate assumption.

*Notation.* Let $(\Omega, \mathcal{F}, \mathbb{P})$ be a fixed probability space. We denote by $\mathbb{E}[X]$ the expectation of a random variable $X$. For $1 \leq p < \infty$, $L^p$ is used to denote the usual space of $p$-integrable real-valued random variables. The $L^p$-integrability of a random variable $X$ is defined as $\mathbb{E}[|X|^p] < \infty$. Fix an integer $M \geq 1$. For an $\mathbb{R}^M$-valued random variable $X$, its law on $\mathcal{B}(\mathbb{R}^M)$, i.e. the Borel sigma-algebra of $\mathbb{R}^M$ is denoted by $\mathcal{L}(X)$. Let $T > 0$ denotes some time horizon. For a positive real number $a$, we denote its integer part by $\lfloor a \rfloor$. The Euclidean scalar product is denoted by $\langle \cdot, \cdot \rangle$, with $|\cdot|$ standing for the corresponding norm (where the dimension of the space may vary depending on the context). Let $\mathbb{R}_{>0} := \{x \in \mathbb{R} |\ x > 0\}$. Let $f : \mathbb{R}^M \to \mathbb{R}$ be a continuously differentiable function. The gradient of $f$ is denote by $\nabla f$. For any integer $q \geq 1$, let $\mathcal{P}(\mathbb{R}^q)$ be the set of probability measures on $\mathcal{B}(\mathbb{R}^q)$. For $\mu, \nu \in \mathcal{P}(\mathbb{R}^M)$, let $\mathcal{C}(\mu, \nu)$ denote the set of probability measures $\zeta$ on $\mathcal{B}(\mathbb{R}^{2M})$ such that its respective marginals are $\mu$ and $\nu$. For any $\mu$ and $\nu \in \mathcal{P}(\mathbb{R}^M)$, the Wasserstein distance of order 2 is defined as

$$W_2(\mu, \nu) = \left( \inf_{\zeta \in \mathcal{C}(\mu, \nu)} \int_{\mathbb{R}^M} \int_{\mathbb{R}^M} |x - y|^2 \ \mathrm{d}\zeta(x, y) \right)^{\frac{1}{2}}.$$

## 2 Technical Background

In this section, we provide a brief summary behind the construction of score-based generative models (SGMs) based on diffusion introduced in Song et al. (2021). The fundamental concept behind SGMs centers around the use of an ergodic (forward) diffusion process to diffuse the unknown data distribution $\pi_\mathsf{D} \in \mathcal{P}(\mathbb{R}^M)$ to a known prior distribution and then learn a backward process to transform the prior to the target distribution $\pi_\mathsf{D}$ by estimating the score function of the forward process. In our analysis, we focus on the forward process $(X_t)_{t \in [0,T]}$ given by an Ornstein-Uhlenbeck (OU) process

$$\mathrm{d}X_t = -X_t \ \mathrm{d}t + \sqrt{2} \ \mathrm{d}B_t, \quad X_0 \sim \pi_\mathsf{D}, \tag{1}$$

where $(B_t)_{t \in [0,T]}$ is an $M$-dimensional Brownian motion and we assume that $\mathbb{E}[|X_0|^2] < \infty$. The process 1 is chosen to match the forward process in the original paper (Song et al., 2021), which is also referred to as Variance Preserving Stochastic Differential Equation. The noising process 1 can also be represented as follows

$$X_t \overset{\mathrm{a.s.}}{=} m_t X_0 + \sigma_t Z_t, \quad m_t = e^{-t}, \quad \sigma_t^2 = 1 - e^{-2t}, \quad Z_t \sim \mathcal{N}(0, I_M), \tag{2}$$

where $\overset{\mathrm{a.s.}}{=}$ denotes almost sure equality and $I_M$ denotes the identity matrix of dimension $M$. Under mild assumptions on the target data distribution $\pi_\mathsf{D}$ (Haussmann & Pardoux, 1986; Cattiaux et al., 2023), the backward process $(Y_t)_{t \in [0,T]} = (X_{T-t})_{t \in [0,T]}$ is given by

$$\mathrm{d}Y_t = (Y_t + 2\nabla \log p_{T-t}(Y_t)) \ \mathrm{d}t + \sqrt{2} \ \mathrm{d}\bar{B}_t, \quad Y_0 \sim \mathcal{L}(X_T), \tag{3}$$

where $\{p_t\}_{t \in [0,T]}$ is the family of densities of $\{\mathcal{L}(X_t)\}_{t \in (0,T]}$ with respect to the Lebesgue measure, $\bar{B}_t$ is an another Brownian motion independent of $B_t$ in 1 defined on $(\Omega, \mathcal{F}, \mathbb{P})$. However, the sampling is done in practice from the invariant distribution of the forward process, which, in this case, is a standard Gaussian distribution. Therefore, the backward process 3 becomes

$$\mathrm{d}\widetilde{Y}_t = (\widetilde{Y}_t + 2\ \nabla \log p_{T-t}(\widetilde{Y}_t)) \ \mathrm{d}t + \sqrt{2} \ \mathrm{d}\bar{B}_t, \quad \widetilde{Y}_0 \sim \pi_\infty = \mathcal{N}(0, I_M). \tag{4}$$

Here, we have $\widetilde{Y}_0 \overset{\mathrm{a.s.}}{=} Z_T$. Since $\pi_\mathsf{D}$ is unknown, the score function $\nabla \log p_t$ in 3 cannot be computed exactly. To address this issue, an estimator $s(\cdot, \theta^*, \cdot)$ is *learned* based on a family of functions $s : [0,T] \times \mathbb{R}^d \times \mathbb{R}^M \to \mathbb{R}^M$ parametrized in $\theta$, aiming at approximating the score of the ergodic (forward) diffusion process over a fixed time window $[0,T]$. In practice, the functions $s$ are neural networks and in particular cases like the motivating example in Section 3.1, the functions $s$ can be wisely designed. The optimal value $\theta^*$ of the parameter $\theta$ is determined by optimizing the following score-matching objective

$$\mathbb{R}^d \ni \theta \mapsto \mathbb{E}\left[ \int_0^T |\nabla \log p_t(X_t) - s(t, \theta, X_t)|^2 \ \mathrm{d}t \right]. \tag{5}$$

To account for numerical instability issues for training and sampling at $t = 0$ as observed in practice in Song et al. (2021, Appendix C) and for the possibility that the integral of the score function in 5 may diverge when $t = 0$ (see Appendix A), a discretised version of the score-matching optimization problem is usually considered

$$\text{minimize} \quad \mathbb{R}^d \ni \theta \mapsto U(\theta) := \int_{\epsilon}^{T} \frac{\kappa(t)}{T - \epsilon} \int_{\mathbb{R}^M} |\nabla \log p_t(x) - s(t, \theta, x)|^2 \, p_t(x) \, dx \, dt, \tag{6}$$

where $\epsilon > 0$ and $\kappa : [0, T] \to \mathbb{R}_{>0}$ is a weighting function. The score-matching objective $U$ in 6 can be rewritten via denoising score matching (Vincent, 2011) as follows

$$U(\theta) = \mathbb{E}[\kappa(\tau)|\sigma_\tau^{-1} Z + s(\tau, \theta, m_\tau X_0 + \sigma_\tau Z)|^2] + C, \tag{7}$$

where the expectation is over $\tau \sim \text{Uniform}([\epsilon, T])$, $X_0 \sim \pi_D$ and $Z \sim \mathcal{N}(0, I_M)$, and where $C \in \mathbb{R}$ is a constant independent of $\theta$ (see Appendix A for the derivation with the OU representation 30). The stochastic gradient $H : \mathbb{R}^d \times \mathbb{R}^m \to \mathbb{R}^d$ of 6 deduced using 7 is given by

$$H(\theta, \mathbf{x}) = 2\kappa(t) \sum_{i=1}^{M} \left( \sigma_t^{-1} z^{(i)} + s^{(i)}(t, \theta, m_t x_0 + \sigma_t z) \right) \nabla_\theta s^{(i)}(t, \theta, m_t x_0 + \sigma_t z), \tag{8}$$

where $\mathbf{x} = (t, x_0, z) \in \mathbb{R}^m$ with $m = 2M + 1$. As contribution to this analysis, we introduce an auxiliary process $(Y_t^{\text{aux}})_{t \in [0,T]}$ containing the approximating function $s$ depending on the (random) estimator of $\theta^*$ denoted by $\hat{\theta}$, for $t \in [0, T]$,

$$dY_t^{\text{aux}} = (Y_t^{\text{aux}} + 2 \, s(T - t, \hat{\theta}, Y_t^{\text{aux}})) \, dt + \sqrt{2} \, d\bar{B}_t, \quad Y_0^{\text{aux}} \sim \pi_\infty = \mathcal{N}(0, I_M). \tag{9}$$

The process 9 will play an important role in the derivation of the nonasymptotic estimates in Wasserstein distance of order two between the target data distribution and the generative distribution of the diffusion model. Indeed, it connects the backward process 4 and the numerical scheme 11, which facilitates the analysis of the convergence of the diffusion model (see Appendix C.2 and Appendix D.2 for more details). For this reason, we introduce a sequence of stepsizes $\{\gamma_k\}_{k \in \{0, \ldots, K\}}$ such that $\sum_{k=0}^{K} \gamma_k = T$. For any $k \in \{0, \ldots, K\}$, let $t_{k+1} = \sum_{j=0}^{k} \gamma_j$ with $t_0 = 0$ and $t_{K+1} = T$. Let $\gamma_k = \gamma \in (0, 1)$ for each $k = 0, \ldots, K$. The discrete process $(Y_k^{\text{EM}})_{k \in \{0, \ldots, K+1\}}$ of the Euler–Maruyama approximation of 9 is given, for any $k \in \{0, \ldots, K\}$, as follows

$$Y_{k+1}^{\text{EM}} = Y_k^{\text{EM}} + \gamma(Y_k^{\text{EM}} + 2 \, s(T - t_k, \hat{\theta}, Y_k^{\text{EM}})) + \sqrt{2\gamma} \, \bar{Z}_{k+1}, \quad Y_0^{\text{EM}} \sim \pi_\infty = \mathcal{N}(0, I_M), \tag{10}$$

where $\{\bar{Z}_k\}_{k \in \{0, \ldots, K+1\}}$ is a sequence of independent $M$-dimensional Gaussian random variables with zero mean and identity covariance matrix. We emphasize that the approximation 10 is the one chosen in the original paper Song et al. (2021). Finally, the continuous-time interpolation of 10, for $t \in [0, T]$, is given by

$$d\widehat{Y}_t^{\text{EM}} = (\widehat{Y}_{\lfloor t/\gamma \rfloor \gamma}^{\text{EM}} + 2 \, s(T - \lfloor t/\gamma \rfloor \gamma, \hat{\theta}, \widehat{Y}_{\lfloor t/\gamma \rfloor \gamma}^{\text{EM}})) \, dt + \sqrt{2} \, d\bar{B}_t, \quad \widehat{Y}_0^{\text{EM}} \sim \pi_\infty = \mathcal{N}(0, I_M), \tag{11}$$

where $\mathcal{L}(\widehat{Y}_k^{\text{EM}}) = \mathcal{L}(Y_k^{\text{EM}})$ at grid points for each $k \in \{0, \ldots, K + 1\}$.

## 3 Main Results

Before introducing the main assumptions of the paper, we present a motivating example.

### 3.1 A Motivating Example: Full Estimates for Multivariate Gaussian Initial Data with Unknown Mean

In this section, we consider the case where the data distribution follows a multivariate normal distribution with unknown mean and identity covariance, i.e., $X_0 \sim \pi_D = \mathcal{N}(\mu, I_d)$ for some unknown $\mu \in \mathbb{R}^d$ with $M = d$. We show that, by using diffusion models, we are able to generate new data from an approximate distribution that is close to $\pi_D$. More precisely, we provide a non-asymptotic convergence bound with explicit constants

in Wasserstein-2 distance between the law of the diffusion model and $\pi_{\mathsf{D}}$, which can be made arbitrarily small by appropriately choosing key parameters on the upper bound. In this example, the estimation of the score function reduces to the estimation of the unknown mean by the methods of convex optimization with optimal dependence of the dimension combined with the most efficient sampling method for high-dimensional Gaussian data.

In this setting by using 1, we can derive the score function given by

$$\nabla \log p_t(x) = -x + m_t \mu, \tag{12}$$

which can be approximated using

$$s(t, \theta, x) = -x + m_t \theta, \quad (t, \theta, x) \in [0, T] \times \mathbb{R}^d \times \mathbb{R}^d. \tag{13}$$

To obtain an approximated score, i.e., to obtain an optimal value of $\theta$ in 13, we opt for a popular class of algorithms called stochastic gradient Langevin dynamics (SGLD) to solve the optimisation problem 6. In addition, we choose the weighting function $\kappa(t) = \sigma_t^2$ as in Song & Ermon (2019) and we set $\epsilon = 0$ in 6. Using the approximating function 13 in 8, we can obtain the following expression for the stochastic gradient

$$\begin{aligned} H(\theta, \mathbf{x}) &= 2\sigma_t^2 \sum_{i=1}^{d} \left( \sigma_t^{-1} z^{(i)} + (-m_t x_0^{(i)} - \sigma_t z^{(i)} + m_t \theta^{(i)}) \right) m_t e_i \\ &= 2\sigma_t^2 m_t \left( \sigma_t^{-1} z - m_t x_0 - \sigma_t z + m_t \theta \right), \end{aligned} \tag{14}$$

where $\mathbf{x} = (t, x_0, z) \in \mathbb{R}^m$ with $m = 2d + 1$ and $e_i$ denotes the unit vector with $i$-th entry being 1. Fixing the so-called inverse temperature parameter $\beta > 0$, the associated SGLD algorithm is given by

$$\theta_0^\lambda := \theta_0, \quad \theta_{n+1}^\lambda = \theta_n^\lambda - \lambda H(\theta_n^\lambda, \mathbf{X}_{n+1}) + \sqrt{2\lambda/\beta}\, \xi_{n+1}, \quad n \in \mathbb{N}_0, \tag{15}$$

where $\lambda > 0$ is often called the stepsize or gain of the algorithm, $(\xi_n)_{n \in \mathbb{N}_0}$ is a sequence of standard Gaussian vectors and

$$(\mathbf{X}_n)_{n \in \mathbb{N}_0} = (\tau_n, X_{0,n}, Z_n)_{n \in \mathbb{N}_0}, \tag{16}$$

is a sequence of i.i.d. random variables generated as follows. For each $n \in \mathbb{N}_0$, we sample $\tau_n$ from Uniform$([0, T])$ such that $\mathcal{L}(\tau_n) = \mathcal{L}(\tau)$, sample $X_{0,n}$ from $\pi_{\mathsf{D}} = \mathcal{N}(\mu, I_d)$ such that $\mathcal{L}(X_{0,n}) = \mathcal{L}(X_0)$, and sample $Z_n$ from $\mathcal{N}(0, I_d)$ such that $\mathcal{L}(Z_n) = \mathcal{L}(Z)$. In addition, we consider the case where $\theta_0$, $(\xi_n)_{n \in \mathbb{N}_0}$, and $(\mathbf{X}_n)_{n \in \mathbb{N}_0}$ in 15 are independent and we have $\mathbb{E}[H(\theta, \mathbf{X}_{n+1})] = \nabla U(\theta)$.

Throughout this section, we fix

$$0 < \lambda \le \min\{\mathbb{E}[\sigma_\tau^2 m_\tau^2]/(4\mathbb{E}[\sigma_\tau^4 m_\tau^4]), 1/(2\mathbb{E}[\sigma_\tau^2 m_\tau^2])\}. \tag{17}$$

Theorem 1 states the non-asymptotic (upper) bounds between the generative distribution of the diffusion model $\mathcal{L}(\widehat{Y}_{K+1}^{\mathrm{EM}})$ and the data distribution $\pi_{\mathsf{D}}$. An overview of the proof can be found in Appendix C.2.

**Theorem 1.** *Under the setting described in this section, then, for any $T > 0$ and $\gamma \in (0, 1/2]$,*

$$\begin{aligned} &W_2(\mathcal{L}(\widehat{Y}_{K+1}^{EM}), \pi_{\mathsf{D}}) \\ &\le \sqrt{2}e^{-2T}(\sqrt{\mathbb{E}\left[|X_0|^2\right]} + \sqrt{d}) \\ &\quad + (\sqrt{4/3} + 2\sqrt{33})(e^{-n\lambda\mathbb{E}[\sigma_\tau^2 m_\tau^2]}\sqrt{\mathbb{E}[|\theta_0 - \theta^*|^2]} + \sqrt{dC_{\mathsf{SGLD},1}/\beta} + \sqrt{\lambda C_{\mathsf{SGLD},2}}) \\ &\quad + \gamma(\sqrt{18d} + \sqrt{132|\theta^*|^2}), \end{aligned} \tag{18}$$

*where $C_{\mathsf{SGLD},1}$ and $C_{\mathsf{SGLD},2}$ are given explicitly in Table 1. In addition, the result in 18 implies that for any $\delta > 0$, if we choose $T > T_\delta$, $\beta \ge \beta_\delta$, $0 < \lambda \le \lambda_\delta$, $n \ge n_\delta$, and $0 < \gamma < \gamma_\delta$, then*

$$W_2(\mathcal{L}(\widehat{Y}_{K+1}^{EM}), \pi_{\mathsf{D}}) < \delta,$$

*where $T_\delta, \beta_\delta, \lambda_\delta, n_\delta$ and $\gamma_\delta$ are given explicitly in Table 1.*

Table 1: Explicit expressions for the constants in Theorem 1.

| CONSTANT | FULL EXPRESSION |
|---|---|
| $C_{\mathsf{SGLD},1}$ | $1/\mathbb{E}[\sigma_\tau^2 m_\tau^2]$ |
| $C_{\mathsf{SGLD},2}$ | $4\mathbb{E}[\sigma_\tau^4 m_\tau^2 (\sigma_\tau^{-1}|Z| + m_\tau|X_0| + \sigma_\tau|Z| + m_\tau|\theta^*|)^2]/\mathbb{E}[\sigma_\tau^2 m_\tau^2]$ |
| $T_\delta$ | $2^{-1}\ln\left(4\sqrt{2}\left(\sqrt{\mathbb{E}[|X_0|^2]} + \sqrt{d}\right)/\delta\right)$ |
| $\beta_\delta$ | $144d(\sqrt{4/3} + 2\sqrt{33})^2/(\delta^2\mathbb{E}[\kappa(\tau)m_\tau^2])$ |
| $\lambda_\delta$ | $\min\left\{\mathbb{E}[\sigma_\tau^2 m_\tau^2]/(4\mathbb{E}[\sigma_\tau^4 m_\tau^4]), 1/(2\mathbb{E}[\sigma_\tau^2 m_\tau^2]), \delta^2\mathbb{E}[\sigma_\tau^2 m_\tau^2]\right.$ $\left.\times(576(\sqrt{4/3} + 2\sqrt{33})^2\mathbb{E}[\sigma_\tau^4 m_\tau^2\left(\sigma_\tau^{-1}|Z| + m_\tau|X_0| + \sigma_\tau|Z| + m_\tau|\theta^*|\right)^2])^{-1}\right\}$ |
| $n_\delta$ | $(\lambda\mathbb{E}[\sigma_\tau^2 m_\tau^2])^{-1}\ln\left(12(\sqrt{4/3} + 2\sqrt{33})\sqrt{\mathbb{E}[|\theta_0 - \theta^*|^2]}/\delta\right)$ WITH FIXED $\lambda$ ($\leq \lambda_\delta$) |
| $\gamma_\delta$ | $\min\left\{\delta/(4(18d + 132|\theta^*|^2)^{1/2}), 1/2\right\}$ |

**Remark 2.** *The result in Theorem 1 achieves the optimal rate of convergence of order one for Euler or Milstein schemes of SDEs with constant diffusion coefficients. Furthermore, one notes that the dependence of the dimension on the upper bound in 18 is in the form of $\sqrt{d}$. To the best of the authors' knowledge, the result in Theorem 1 is the first convergence bound where the parameters involved in the sampling and optimization steps of the diffusion models appear explicitly. In the optimization procedure, we use SGLD algorithm 15 to solve the problem 6. Since the stochastic gradient $H$ in 14 is strongly convex by Proposition 13, it has been proved, for instance in Barkhagen et al. (2021), that, for large enough $\beta > 0$, the output of SGLD is an almost minimizer of 6 when $n$ is large. Thus, in the diffusion model, we set $\hat{\theta} = \theta_n^\lambda$ indicating $\nabla \log p_t(x) \approx s(t, \hat{\theta}, x) = s(t, \theta_n^\lambda, x)$ for large values of $n$ and for all $t$ and $x$. Crucially, this allows us to use the established convergence results for SGLD to deduce a sampling upper bound for $W_2(\mathcal{L}(\widehat{Y}_{K+1}^{EM}), \pi_{\mathsf{D}})$ with explicit constants in 18. Consequently, this bound can be controlled by any given precision level $\delta > 0$ by appropriately choosing $T, \beta, \lambda, n$ and $\gamma$.*

This motivating example has focused on exploring the convergence of diffusion-based generative models using a Langevin-based algorithm, specifically SGLD, which is well-known for its theoretical guarantees in achieving global convergence. However, in the general case discussed in Section 3.2, we do not prescribe a specific optimizer to choose to minimise the distance between $\hat{\theta}$ and $\theta^*$.

## 3.2 General Case

In this section, we derive the full non-asymptotic estimates in Wasserstein distance of order two between the target data distribution $\pi_{\mathsf{D}}$ and the generative distribution of the diffusion model under the assumptions stated below. As explained in Section 2 (see also Appendix A), it could be necessary in the general setting to restrict $t \in [\epsilon, T]$ for $\epsilon \in (0, 1)$ in 6. Therefore, we truncate the integration in the backward diffusion at $T - \epsilon$ and run the process $(Y_t)_{t \in [0, T-\epsilon]}$.

### 3.2.1 Assumptions for the General Case

In the motivating example in Section 3.1, we have chosen the SGLD algorithm to solve the optimisation problem 6. Other algorithms, such as ADAM (Kingma & Ba, 2015), TheoPouLa (Lim & Sabanis, 2024) and stochastic gradient descent (Jentzen et al., 2021), can be chosen as long as they satisfy the following assumption. Fix $\epsilon > 0$.

**Assumption 1.** *Let $\theta^*$ be a minimiser[1] of 6 and let $\hat{\theta}$ be the (random) estimator of $\theta^*$ obtained through some approximation procedure such that $\mathbb{E}[|\hat{\theta}|^4] < \infty$. There exists $\widetilde{\varepsilon}_{AL} > 0$ such that*

$$\mathbb{E}[|\hat{\theta} - \theta^*|^2] < \widetilde{\varepsilon}_{AL}.$$

---

[1]The score-matching optimization problem 6 is not necessarily (strongly) convex.

**Remark 3.** *As a consequence of Assumption 1, we have* $\mathbb{E}[|\hat{\theta}|^2] < 2\widetilde{\varepsilon}_{AL} + 2|\theta^*|^2$.

We consider the following assumption on the data distribution.

**Assumption 2.** *The data distribution* $\pi_D$ *has a finite second moment, is strongly log-concave, and* $\int_{\epsilon}^{T} |\nabla \log p_t(0)|^2 \, dt < \infty$.

**Remark 4.** *As a consequence of Assumption 2 and the preservation of strong log-concavity under convolution, see, e.g., Saumard & Wellner (2014, Proposition 3.7), there exists* $L_{MO} : [0, T] \to (0, \infty]$ *such that for all* $t \in [0, T]$ *and* $x, \bar{x} \in \mathbb{R}^M$, *we have*

$$\langle \nabla \log p_t(x) - \nabla \log p_t(\bar{x}), x - \bar{x} \rangle \leq -L_{MO}(t)|x - \bar{x}|^2. \tag{19}$$

*The function* $L_{MO}(t)$ *in 19 has a lower bound for all* $t \in [0, T]$, *which we denote by* $\widehat{L}_{MO}$. *Moreover, Assumption 2 with the estimate 19 implies that the processes in 3 and in 4 have a unique strong solution, see, e.g., Krylov (1991, Theorem 1).*

Next, we consider the following assumption on the approximating function $s$ which is used in Remark 12.

**Assumption 3.a.** *The function* $s : [0, T] \times \mathbb{R}^d \times \mathbb{R}^M \to \mathbb{R}^M$ *is continuously differentiable in* $x \in \mathbb{R}^M$. *Let* $D_1 : \mathbb{R}^d \times \mathbb{R}^d \to \mathbb{R}_+$, $D_2 : [0, T] \times [0, T] \to \mathbb{R}_+$ *and* $D_3 : [0, T] \times [0, T] \to \mathbb{R}_+$ *be such that* $\int_{\epsilon}^{T} \int_{\epsilon}^{T} D_2(t, \bar{t}) \, dt \, d\bar{t} < \infty$ *and* $\int_{\epsilon}^{T} \int_{\epsilon}^{T} D_3(t, \bar{t}) \, dt \, d\bar{t} < \infty$. *For* $\alpha \in \left[\frac{1}{2}, 1\right]$ *and for all* $t, \bar{t} \in [0, T]$, $x, \bar{x} \in \mathbb{R}^M$, *and* $\theta, \bar{\theta} \in \mathbb{R}^d$, *we have that*

$$|s(t, \theta, x) - s(\bar{t}, \bar{\theta}, \bar{x})| \leq D_1(\theta, \bar{\theta})|t - \bar{t}|^\alpha + D_2(t, \bar{t})|\theta - \bar{\theta}| + D_3(t, \bar{t})|x - \bar{x}|,$$

*where* $D_1$, $D_2$ *and* $D_3$ *have the following growth in each variable: i.e. there exist* $\mathsf{K}_1$, $\mathsf{K}_2$, *and* $\mathsf{K}_3 > 0$ *such that for each* $t, \bar{t} \in [0, T]$ *and* $\theta, \bar{\theta} \in \mathbb{R}^d$,

$$|D_1(\theta, \bar{\theta})| \leq \mathsf{K}_1(1 + |\theta| + |\bar{\theta}|), \qquad |D_2(t, \bar{t})| \leq \mathsf{K}_2(1 + |t|^\alpha + |\bar{t}|^\alpha),$$
$$|D_3(t, \bar{t})| \leq \mathsf{K}_3(1 + |t|^\alpha + |\bar{t}|^\alpha).$$

By adding a further condition on the gradient of $s$ in Assumption 3.a we are able to achieve the optimal rate of convergence in Theorem 10 below.

**Assumption 3.b.** *Let* $s$ *be as in Assumption 3.a and there exists* $\mathsf{K}_4 > 0$ *such that, for all* $x, \bar{x} \in \mathbb{R}^M$ *and for any* $k = 1, \ldots M$,
$$|\nabla_x s^{(k)}(t, \theta, x) - \nabla_{\bar{x}} s^{(k)}(t, \theta, \bar{x})| \leq \mathsf{K}_4(1 + 2|t|^\alpha)|x - \bar{x}|.$$

**Remark 5.** *Let* $\mathsf{K}_{Total} := \mathsf{K}_1 + \mathsf{K}_2 + \mathsf{K}_3 + |s(0, 0, 0)| > 0$. *Using Assumption 3.b, we have*

$$|s(t, \theta, x)| \leq \mathsf{K}_{Total}(1 + |t|^\alpha)(1 + |\theta| + |x|).$$

We postpone the proof of Remark 5 to Appendix D.1.

**Remark 6.** *Assumption 3.a and 3.b impose Lipschitz continuity on a family of approximating functions* $s(\cdot, \cdot, \cdot)$ *with respect to the input variables,* $t$ *and* $x$, *as well as the parameters* $\theta$. *It is well-known that the continuity properties of neural networks with respect to* $t$, *and* $x$ *are largely determined by the activation function at the last layer. For instance, neural networks with activation functions such as hyberbolic tangent and sigmoid functions at the last layer satisfy the Lipschitz continuity with respect to* $t$ *and* $x$ *(Virmaux & Scaman, 2018; Fazlyab et al., 2019).*

For the following assumption on the score approximation, we let $\hat{\theta}$ be as in Assumption 1 and we let $(Y_t^{\text{aux}})_{t \in [0, T]}$ be the auxiliary process defined in 9.

**Assumption 4.** *There exists* $\varepsilon_{SN} > 0$ *such that*

$$\mathbb{E}\int_0^{T-\epsilon} |\nabla \log p_{T-r}(Y_r^{aux}) - s(T - r, \hat{\theta}, Y_r^{aux})|^2 \, dr < \varepsilon_{SN}. \tag{20}$$

**Remark 7.** *We highlight that the expectation in Assumption 4 is taken over the auxiliary process 9. This density is known since the approximating function s and the estimator $\hat{\theta}$ are known. To the best of authors' knowledge, this is a novelty with respect to previous works (Bortoli, 2022; Chen et al., 2023d; Lee et al., 2022; 2023; Chen et al., 2023a; Conforti et al., 2025; Benton et al., 2024) which consider the unknown density of the forward process (or its numerical discretization).*

**Remark 8.** *Assumption 4 is satisfied, along with Assumption 3.b, for data distributions satisfying Assumption 2, beyond the multivariate Gaussian case discussed in the motivating example. Indeed,*

$$\mathbb{E} \int_0^{T-\epsilon} |\nabla \log p_{T-r}(Y_r^{aux}) - s(T-r, \hat{\theta}, Y_r^{aux})|^2 \; dr$$

$$\leq 2 \, \mathbb{E} \int_0^{T-\epsilon} |\nabla \log p_{T-r}(Y_r^{aux}) - s(T-r, \theta^*, Y_r^{aux})|^2 \; dr \tag{21}$$

$$+ 2 \, \mathbb{E} \int_0^{T-\epsilon} |s(T-r, \theta^*, Y_r^{aux}) - s(T-r, \hat{\theta}, Y_r^{aux})|^2 \; dr.$$

*If $\theta^*$ is such that $s(t, \theta^*, x) = \nabla \log p_t(x)$ as in the motivating example in Section 3.1, then the first term on the right-hand side of 21 vanishes. Otherwise, we expect that the first term on the right-hand side above to be small. Indeed, by the definition of strong log-concavity, see e.g. Saumard & Wellner (2014, Definition 2.9), we have*

$$\nabla \log p_t(x) = \nabla \log(g(x)) + \nabla \log(\phi_t(x)), \tag{22}$$

*where $g$ is some log-concave function and $\phi_t$ is a multivariate Gaussian density. If $g$ is a multivariate logistic distribution[2] (Malik & Abraham, 1973), then its score function is given by*

$$\frac{\partial}{\partial x_k} \log(g(x)) = -1 - (M+1) \frac{-\exp(-x_k)}{1 + \sum_{k=1}^{M} \exp(-x_k)}, \qquad -\infty < x_k < \infty, \qquad k = 1, \ldots, M, \tag{23}$$

*while the Hessian of $\log(g(x))$ is bounded (see Appendix B for more details). Therefore, the score function of the multivariate logistic distribution 23 is Lipschitz, and, as a consequence, $\nabla \log p_t(x)$ in 22 is still Lipschitz. Thus, we expect to have a good control on the first term on the right-hand side of 21 since the function $s$ satisfying Assumption 3.b approximates a Lipschitz score function. In general, there exists a function*

$$c(t, x) := \nabla \log p_t(x) - s(t, \theta^*, x), \qquad t > 0, \; x \in \mathbb{R}^M, \tag{24}$$

*and therefore one has to define a log-concave function $g$ such that the first term on the right-hand side of 21 is small. Clearly, this is a problem specific challenge and it may not always have a good solution. The second term on the right-hand side of 21 is controlled by Assumption 3.b and Assumption 1 (see Appendix B for more details).*

**Remark 9.** *We conduct a numerical experiment to show the convergence of diffusion-based generative models under Assumption 1, 2, 3.b and 4. We consider the case where $X_0 \sim \pi_{\mathsf{D}} = \mathcal{N}(\mu, I_d)$ with $\mu = (-1.2347, -0.89244)$ and $d = 2$. We use SGLD algorithm 15 to solve the optimization problem 7, with $\kappa(t) = \sigma_t^2$, $\epsilon = 0$, $T = 2$, $\lambda = 5 \times 10^{-5}$, $\beta = 10^{12}$, and $n = 4 \times 10^4$. At each iteration, 128 mini-batch samples are used to estimate the stochastic gradient. Then, we generate samples using the Euler-Maruyama approximation 10 with $\gamma = 10^{-3}$ and $s$ given in 13. At each iteration $n$, we evaluate the quality of $100,000$ generated samples using the Wasserstein distance of order two. In addition, we compute the $L^2$ error between the score function and the approximated function $s$ with $\theta_n^\lambda$, using the auxiliary process as in Assumption 4. Figure 1 (a) demonstrates the error in Assumption 4 vanishes as the generative model converges, where the degree of convergence is measured by $W_2(\mathcal{L}(Y_K^{EM}), \pi_{\mathsf{D}})$. This empirical observation justifies Assumption 4. Moreover, we explore the relationship between the quality of generated samples and the error using 7 via denoising score matching in Figure 1 (b).*

---

[2]The multivariate logistic distribution is an example of elliptical distribution widely used in portfolio risk management (Xiao & Valdez, 2015; Owen & Rabinovitch, 1983).

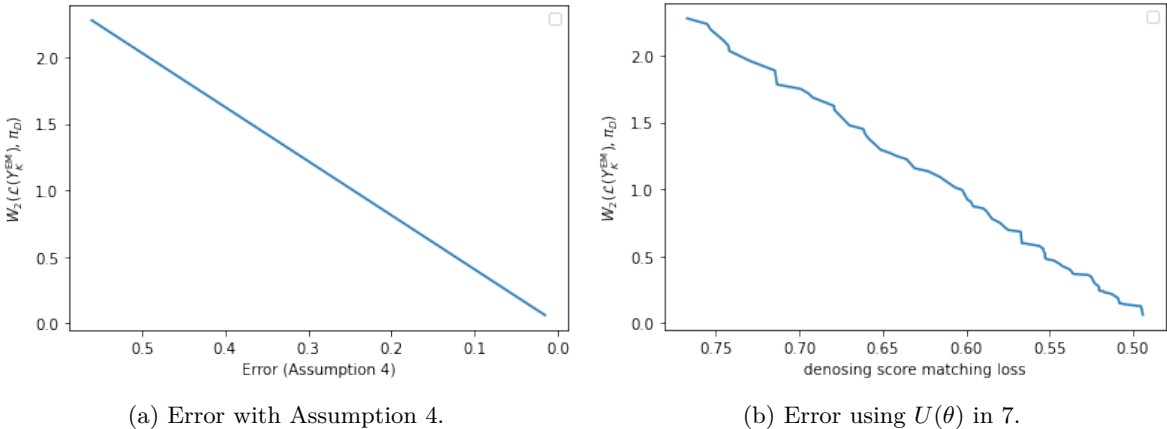

(a) Error with Assumption 4.      (b) Error using $U(\theta)$ in 7.

Figure 1: The quality of generated samples with respect to (a) the error with Assumption 4 and (b) the error obtained through denoising score matching using $U(\theta)$ in 7.

### 3.2.2 Full Estimates for the General Case

The main result under the general setting is stated as follows. An overview of the proof can be found in Appendix D.2.

**Theorem 10.** *Let Assumptions 1, 2, 3.b and 4 hold. Then, there exist constants $C_1$, $C_2$, $C_3$ and $C_4 > 0$ such that for any $T > 0$ and $\gamma, \epsilon \in (0,1)$,*

$$W_2(\mathcal{L}(Y_K^{EM}), \pi_D) \leq C_1\sqrt{\epsilon} + C_2 e^{-2\widehat{L}_{MO}(T-\epsilon)-\epsilon} + C_3(T,\epsilon)\sqrt{\varepsilon_{SN}} + C_4(T,\epsilon)\gamma^\alpha, \tag{25}$$

*where $C_1$, $C_2$, $C_3$ and $C_4$ are given explicitly in Table 4 (Appendix E). In addition, the result in 25 implies that for any $\delta > 0$, if we choose $0 \leq \epsilon < \epsilon_\delta$, $T > T_\delta$, $0 < \varepsilon_{SN} < \varepsilon_{SN,\delta}$ and $0 < \gamma < \gamma_\delta$ with $\epsilon_\delta$, $T_\delta$, $\varepsilon_{SN,\delta}$, and $\gamma_\delta$ given in Table 4, then*

$$W_2(\mathcal{L}(Y_K^{EM}), \pi_D) < \ \delta.$$

**Remark 11.** *The error bounds in 25 are not as good as the ones provided in Theorem 1 due to the general form of the approximating function $s$. We emphasize that the optimal rate of convergence of order $\alpha \in \left[\frac{1}{2}, 1\right]$ for the Euler or Milstein scheme of SDEs with constant diffusion coefficients is achieved using the Lipschitz continuity on the derivative of $s$ in Assumption 3.b. In the explicit expression of $C_4$ in Table 4, the dependence of the dimension is $O(M)$ due to numerical techniques from Kumar & Sabanis (2019) used in the proof of Theorem 10 to achieve the optimal rate of convergence.*

**Remark 12.** *If we replace Assumption 3.b with Assumption 3.a in Theorem 10, then the bound 25 becomes*

$$W_2(\mathcal{L}(Y_K^{EM}), \pi_D) \leq C_1\sqrt{\epsilon} + C_2 e^{-2\widehat{L}_{MO}(T-\epsilon)-\epsilon} + C_3(T,\epsilon)\sqrt{\varepsilon_{SN}} + \widetilde{C}_4(T,\epsilon)\gamma^{1/2},$$

*where $C_1$, $C_2$, $C_3$ are the same as in Theorem 10 and $\widetilde{C}_4(T,\epsilon)$ contains a better dependence on the dimension, namely $O(\sqrt{M})$, than the one achieved in Theorem 10, and it is explicitly provided in Table 4 (Appendix E). Although this relaxation achieves the same dependence on the data dimension as in the motivating example in Theorem 1, it leads to a worse rate of convergence of order $1/2$.*

## 4 Related Work and Comparison

We describe assumptions and results of various existing works in our notation and framework to facilitate the comparisons with our results which are provided in 2-Wasserstein distance. Beyond being theoretical relevant, the choice of the use of this metric is motivated by its applications in generative modeling. A popular performance metric currently used to examine the quality of the images produced by generative models is the Fréchet Inception Distance (FID) which was originally introduced in Heusel et al. (2017). This

metric measures the Fréchet distance between the distribution of generated samples and the distribution of real samples, assuming Gaussian distributions, which is equivalent to the Wasserstein distance of order two. Our results provided under this metric are therefore also relevant for practical applications.

We can classify the previous approaches based on their assumptions on two key quantities: (i) score approximation error and (ii) assumptions on the data distribution. Based on these approaches, we give a brief overview of some of the most relevant recent contributions in the field.

### 4.1 Score Approximation Assumptions

The early work of De Bortoli et al. (2021) requires an $L^\infty$-bound on the score approximation, which is in contrast with the $L^2$ nature of the score-matching optimization problem 6. Most of the recent analysis of score-based generative models, see, e.g., Chen et al. (2023d); Lee et al. (2022; 2023); Chen et al. (2023a); Conforti et al. (2025); Benton et al. (2024), have considered assumptions (in $L^2$) on the absolute error of the following type, i.e. for any $k \in \{0, \dots, N-1\}$, there exists $\underline{\varepsilon} > 0$

$$\mathbb{E}\left[|\nabla \log p_{t_k}(X_{t_k}) - s(t_k, \hat{\theta}, X_{t_k})|^2\right] \leq \underline{\varepsilon}, \tag{26}$$

where the expectation is taken with respect to the unknown $\{p_t\}_{t \in [0,T]}$ and $\hat{\theta}$ is a deterministic quantity. In Chen et al. (2023a); Conforti et al. (2025); Benton et al. (2024), the assumption 26 is written in integral (averaged) form. In Lee et al. (2023), the bound in 26 is not uniform over $t$, i.e. $\underline{\varepsilon}_t := \frac{\varepsilon}{\sigma_t^4}$ and this allows the score function to grow as $\frac{1}{\sigma_t^2}$ as $t \to 0$. The authors in Conforti et al. (2025) use the relative $L^2$-score approximation error such as

$$\mathbb{E}\left[|2\nabla \log \tilde{p}_{t_k}(X_{t_k}) - \tilde{s}(t_k, \hat{\theta}, X_{t_k})|^2\right] \leq \underline{\varepsilon}\, \mathbb{E}\left[|2\nabla \log \tilde{p}_{t_k}(X_{t_k})|^2\right], \tag{27}$$

where the expectation is with respect to the unknown density of the law of $X_t$ against the Gaussian distribution $\pi_\infty(x)$, i.e. $\tilde{p}_t(x) := p_t(x)/\pi_\infty(x)$, and $\tilde{s}(t, \theta, x)$ in 27 is the approximating function for the score function of $\tilde{p}_t$. A pointwise assumption in Bortoli (2022), given by

$$|\nabla \log p_t(x) - s(t, \hat{\theta}, x)| \leq C(1 + |x|)/\sigma_t^2, \tag{28}$$

for $C \geq 0$, is considered under the manifold (compact) setting. The assumption 28 takes into account the explosive behaviour of the score function as $t \to 0$. In an attempt to obtain a weaker control than 28, the assumption Bortoli (2022, A5) is used, namely

$$\mathbb{E}\left[|\nabla \log p_{T-t_k}(Y_k^{\mathrm{EI}}) - s(T - t_k, \hat{\theta}, Y_k^{\mathrm{EI}})|^2\right] \leq C^2 \mathbb{E}\left[(1 + |Y_k^{\mathrm{EI}}|^2)\right]/\sigma_{T-t_k}^4. \tag{29}$$

We note that, unlike 26 and 27, the expectation in 29 is taken with respect to the algorithm $Y_k^{\mathrm{EI}}$ given by the exponential integrator (EI) discretization scheme[3], which has known density. However, the bounds of Bortoli (2022, Theorem H.1) in Wasserstein distance of order one derived under this assumption scale exponentially in problem parameters such as the diameter of the manifold $\mathfrak{M}$ and the inverse of the early stopping time parameter $\epsilon$, i.e. $\exp(O(\mathrm{diam}(\mathfrak{M})^2/\epsilon))$. Furthermore, an assumption similar to 29 is considered in a very recent and concurrent result[4] Gao et al. (2023):

$$\sup_{k=1,\dots,K} \left(\mathbb{E}\left[|\nabla \log p_{T-(k-1)\eta}(Y_k^{\mathrm{EI}}) - s(T - (k-1)\eta, \hat{\theta}, Y_k^{\mathrm{EI}})|^2\right]\right)^{1/2} \leq \underline{\varepsilon},$$

where $\eta > 0$ is the stepsize and $T = K\eta$ with $K \in \mathbb{N}$.

We emphasize that the existing results in the literature do not take the expectation with respect to the stochastic optimizer $\hat{\theta}$. This, together with the use of our novel auxiliary process 9 that uses only known information, allows us to deduce state-of-the-art bounds in the Wasserstein distance of order two in the following sense. The bounds scale polynomially in the data dimension $M$, i.e. $O(\sqrt{M})$ as shown in Theorem 1 and Remark 12, and achieve the optimal rate of convergence: of order one in Theorem 1 and of order $\alpha \in \left[\frac{1}{2}, 1\right]$ in Theorem 10.

---

[3]This analysis can be extended to the Euler-Maruyama numerical scheme in $Y_k^{\mathrm{EM}}$ 10.

[4]The concurrent paper Gao et al. (2023) appeared few days earlier when the first draft of this work was made available online.

## 4.2 Assumptions on the Data Distribution

The vast majority of the results available in the literature are in KL divergence and TV distance. For two general data distributions $\mu$ and $\nu$, there is no known relationship between their KL divergence and their $W_2$. However, for strongly log-concave data distributions, a bound on the Wasserstein distance of order two in terms of KL divergence follows from an extension of Talagrand's inequality Gozlan & Léonard (2010, Corollary 7.2).

Some convergence results in different metrics can be deduced under the following types of assumptions. Convergence bounds in Wasserstein distance of order one with exponential complexity has been obtained in Bortoli (2022) under the so-called manifold hypothesis, namely assuming that the target distribution is supported on a lower-dimensional manifold or is given by some empirical distribution. Moreover, the results in TV distance in Lee et al. (2022) and in KL divergence Yang & Wibisono (2023) assumed that the data distribution satisfies a logarithmic Sobolev inequality and the score function is Lipschitz resulting in convergence bounds characterized by polynomial complexity. By replacing the requirement that the data distribution satisfies a functional inequality with the assumption that $\pi_D$ has finite KL divergence with respect to the standard Gaussian and by assuming that the score function for the forward process is Lipschitz, the authors in Chen et al. (2023d) managed to derive bounds in TV distance which scale polynomially in all the problem parameters. By requiring only the Lipschitzness of the score at the initial time instead of the whole trajectory, the authors in Chen et al. (2023a, Theorem 2.5) managed to show, using an exponentially decreasing then linear step size, convergence bounds in KL divergence with quadratic dimensional dependence and logarithmic complexity in the Lipschitz constant. In the work by Benton et al. (2024), the authors provide convergence bounds in KL divergence that are linear in the data dimension, up to logarithmic factors, using early stopping and assuming finite second moments of the data distribution. A careful examination of Benton et al. (2024, Proof of Theorem 1 and Corollary 1) reveals that the authors still require the uniqueness of solutions for the backward SDE 3. For instance, either measurability and boundedness of the drift (Stroock & Varadhan, 1997) or integrability of the drift (Röckner & Zhao, 2023) should be imposed if the uniqueness of weak solutions is considered. Therefore, additional conditions on the score function, depending on the type of uniqueness of solutions considered, are still needed in their bounds in Theorem 1 and Corollary 1.

Assuming the finiteness of the second moment of the data distribution and using an EI discretization scheme with constant and exponentially decreasing step sizes, the authors in Conforti et al. (2025, Corollary 2.4) derive a KL divergence bound with early stopping, which scales linearly in the data dimension up to logarithmic factors. In terms of dependence of the data dimension, this bound is slightly worse than the bound in $W_2$ provided in Remark 12, which does not include the logarithmic dependence of $M$. Bounds in KL without early stopping have been derived in Conforti et al. (2025) for data distributions with finite Fisher information with respect to the standard Gaussian distribution. The bounds in Conforti et al. (2025, Theorem 2.1 and 2.2) scale linearly in the Fisher information when an EI discretization scheme with constant step size is used and logarithmically in the Fisher information when an exponential-then-constant step size Conforti et al. (2025, Theorem 2.3) is chosen. We note that Conforti et al. (2025, Theorem 2.1 and 2.2) cannot achieve the optimal rate of convergence of Theorem 1 in the motivating example.

We summarise the results of Chen et al. (2023a); Benton et al. (2024); Conforti et al. (2025) and compare them to ours in Table 2 and Table 3, making the distinction based on whether the early stopping rule is applied. In addition, a careful examination of Chen et al. (2023a, Proof of Theorem 2.2) reveals that the authors require the uniqueness of solutions for the backward SDE 3. For instance, when strong solutions are considered, this implies that the score function should be (at least) monotone in the space variable, and a suitable integrability condition in $t$ (similar to our Assumption 2) is still needed to guarantee uniqueness of the solution (see, e.g., Krylov (1991, Theorem 1)). For weak solutions, we refer to the discussion of Benton et al. (2024, Proof of Theorem 1 and Corollary 1) above.

We emphasize that in Theorem 10, we do not assume the score function to be Lipschitz continuous with a uniformly bounded Lipschitz constant. This is particularly useful for future work in nonconvex settings, where the upper bound estimates will be independent of the (potentially large) Lipschitz constant of the score function, which could otherwise hide additional dimensional dependencies. The requirement

$\int_\epsilon^T |\nabla \log p_t(0)|^2 dt < \infty$ in Assumption 2 is weaker than the Lipschitz assumption on the score function, but it is still difficult to verify in practical applications.

As pointed out in Chen et al. (2023d, Section 4), some type of log-concavity assumption on the data distribution is needed to derive polynomial convergence rates in 2-Wasserstein distance. This justifies the need for our Assumption 2. The concurrent result Gao et al. (2023) has a similar assumption. For completeness, we also mention that the results in the same metric, which have appeared after the first version of our preprint on arXiv (e.g., Tang & Zhao (2024); Strasman et al. (2025)), continue to assume strong log-concavity of the data distribution.

Table 2: Summary of previous bounds without early stopping and our result in Theorem 1. Bounds expressed in terms of the number of steps required to guarantee an error of at most $\delta$ in the stated metric. The relative Fisher information of the target $\pi_D$ against standard Gaussian measure $\pi_\infty$ is denoted by $\mathrm{FI}(\pi_D|\pi_\infty)$. All the bounds assume that $\pi_D$ has finite second moments.

| Optimization problem solved | Regularity condition | Metric | Complexity | Reference |
|---|---|---|---|---|
| No | $\forall t, \quad \nabla \log p_t \quad L$-Lipschitz | $\sqrt{\mathrm{KL}(\mathcal{L}(\widehat{Y}_{K+1}^{\mathrm{EI}})\|\pi_D)}$ | $\tilde{O}\left(\frac{ML^2}{\delta}\right)$ | Chen et al. (2023a, Theorem 2.1) |
| No | Conforti et al. (2025, H2), i.e. $\mathrm{FI}(\pi_D|\pi_\infty) < \infty$ | $\sqrt{\mathrm{KL}(\mathcal{L}(\widehat{Y}_{K+1}^{\mathrm{EI}})\|\pi_D)}$ | $\tilde{O}\left(\sqrt{\frac{M+\mathbb{E}[|X_0|^2]}{\delta}}\log^2(\mathfrak{L})\right)$, with $\mathfrak{L} := M^{-1}\mathrm{FI}(\pi_D|\pi_\infty)$ | Conforti et al. (2025, Theorem 2.3) |
| Yes | $\pi_D \sim \mathcal{N}(\mu, I_M)$ | $W_2(\mathcal{L}(\widehat{Y}_{K+1}^{\mathrm{EM}}), \pi_D)$ | $\tilde{O}(\frac{\sqrt{M}}{\delta})$ | Theorem 1 |

Table 3: Summary of previous bounds with early stopping and our results in Remark 12. The results in Chen et al. (2023a, Theorem 2.2), Benton et al. (2024, Theorem 1), and Conforti et al. (2025, Corollary 2.4) are stated for the smoothed version of the data, denoted by $\pi_D^\epsilon$ and using the score approximation assumption 26 with $\underline{\varepsilon}$. Therefore, an additional error should be added to their bounds, as the distance between $\pi_D^\epsilon$ and $\pi_D$ scales with $\sqrt{M}$ in $W_2$ (see Appendix D.2).

| Assumption on the data | Metric | Error bound | Reference |
|---|---|---|---|
| Finite second moments of $\pi_D$ and 26. See Section 4.2 for conditions on $\nabla \log p_t$ for the uniqueness of $Y_t$. | $\sqrt{\mathrm{KL}(\mathcal{L}(\widehat{Y}_K^{\mathrm{EI}})\|\pi_D^\epsilon)}$ | $\sqrt{(\mathbb{E}[|X_0|^2] + M)e^{-T}} + \sqrt{T\underline{\varepsilon}} + M(T + \log \epsilon^{-1})/\sqrt{N}$, (+ additional bounds between $\pi_D^\epsilon$ and $\pi_D$) | Chen et al. (2023a, Theorem 2.2) |
| Finite second moments of $\pi_D$ and 26. See Section 4.2 for conditions on $\nabla \log p_t$ for the uniqueness of $Y_t$. | $\sqrt{\mathrm{KL}(\mathcal{L}(\widehat{Y}_K^{\mathrm{EI}})\|\pi_D^\epsilon)}$ | $\sqrt{\underline{\varepsilon}} + \sqrt{\tilde{O}(M/N)} + \sqrt{Me^{-2T}}$, (+ additional bounds between $\pi_D^\epsilon$ and $\pi_D$) | Benton et al. (2024, Theorem 1) |
| Finite second moments of $\pi_D$ and 26. | $\sqrt{\mathrm{KL}(\mathcal{L}(\widehat{Y}_K^{\mathrm{EI}})\|\pi_D^\epsilon)}$ | $\sqrt{(\mathbb{E}[|X_0|^2] + M)e^{-T}} + \sqrt{T\underline{\varepsilon}} + [c[(M + \mathbb{E}[|X_0|^2])(\log(M+\mathbb{E}[|X_0|^2])+\log(\epsilon^{-1})+1)]]^{1/2}$, with $c, \epsilon \in (0, 1/2]$ (+ additional bounds between $\pi_D^\epsilon$ and $\pi_D$) | Conforti et al. (2025, Corollary 2.4) |
| Assumption 2 and Assumption 4. | $W_2(\mathcal{L}(\widehat{Y}_K^{\mathrm{EM}}), \pi_D)$ | $O(\sqrt{M})\sqrt{\epsilon} + O(\sqrt{M})e^{-2\widehat{L}_{\mathrm{MO}}(T-\epsilon)-\epsilon} + O(e^{(1+\zeta-2\widehat{L}_{\mathrm{MO}})(T-\epsilon)})\sqrt{\varepsilon_{\mathrm{SN}}} + O(\sqrt{M}e^{T^{2\alpha+1}}T^{2\alpha+1}\widetilde{\varepsilon}_{\mathrm{AL}}^{1/2})\gamma^{1/2}$ | Remark 12 |

**Acknowledgments**

This work has been supported by The Alan Turing Institute through the Theory and Methods Challenge Fortnights event "Accelerating generative models and nonconvex optimisation", which took place at The Alan Turing Institute headquarters. This work was made possible by a Research-in-Groups programme funded by the International Centre for Mathematical Sciences, Edinburgh. This work is supported by the Guangzhou-HKUST(GZ) Joint Funding Programs (No. 2024A03J0630 and No. 2025A03J3322). This work has received funding from the Ministry of Trade, Industry and Energy (MOTIE) and Korea Institute for Advancement of Technology (KIAT) through the International Cooperative R&D program (No.P0025828). S.S. was supported by the Alan Turing Institute under the EPSRC grant EP/N510129/1.

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

## Appendix

## A  Objective Function via Denoising Score Matching

In this section, we show that the objective function $U$ in 6 can be written into 7 using denoising score matching (Vincent, 2011) and the OU representation

$$X_t \overset{\mathrm{d}}{=} m_t X_0 + \sigma_t Z, \quad m_t = e^{-t}, \quad \sigma_t^2 = 1 - e^{-2t}, \quad Z \sim \mathcal{N}(0, I_M), \tag{30}$$

where $\overset{\mathrm{d}}{=}$ denotes equality in distribution. We start by noticing that

$$\begin{aligned}
&\int_{\mathbb{R}^M} \langle \nabla \log p_t(x), s(t, \theta, x) \rangle \ p_t(x) \ \mathrm{d}x \\
&= \int_{\mathbb{R}^M} \langle \nabla p_t(x), \ s(t, \theta, x) \rangle \ \mathrm{d}x \\
&= \int_{\mathbb{R}^M} \left\langle \nabla_x \int_{\mathbb{R}^M} p_0(\tilde{x}) p_{t|0}(x|\tilde{x}) \ \mathrm{d}\tilde{x}, \ s(t, \theta, x) \right\rangle \ \mathrm{d}x \\
&= \int_{\mathbb{R}^M} \left\langle \int_{\mathbb{R}^M} p_0(\tilde{x}) p_{t|0}(x|\tilde{x}) \nabla_x \log(p_{t|0}(x|\tilde{x})) \ \mathrm{d}\tilde{x}, \ s(t, \theta, x) \right\rangle \ \mathrm{d}x \\
&= \int_{\mathbb{R}^M} \int_{\mathbb{R}^M} p_{t,0}(x, \tilde{x}) \left\langle \nabla_x \log(p_{t|0}(x|\tilde{x})), \ s(t, \theta, x) \right\rangle \ \mathrm{d}x \ \mathrm{d}\tilde{x},
\end{aligned} \tag{31}$$

where $p_{t|0}$ is the density of the transition kernel associated with 1 and $p_{t,0}$ is the joint density of $X_t$ and $X_0$. Using 31 in the objective function given in 6 and the OU representation 30, we have

$$\begin{aligned}
U(\theta) &= \int_\epsilon^T \frac{\kappa(t)}{T - \epsilon} \int_{\mathbb{R}^M} \left( |\nabla \log p_t(x)|^2 - 2\langle \nabla \log p_t(x), s(t, \theta, x) \rangle + |s(t, \theta, x)|^2 \right) \ p_t(x) \ \mathrm{d}x \ \mathrm{d}t \\
&= \int_\epsilon^T \frac{\kappa(t)}{T - \epsilon} \int_{\mathbb{R}^M} \int_{\mathbb{R}^M} \left( |\nabla \log p_t(x)|^2 - 2\left\langle \nabla_x \log(p_{t|0}(x|\tilde{x})), \ s(t, \theta, x) \right\rangle \right. \\
&\qquad\qquad\qquad\qquad\qquad \left. + |s(t, \theta, x)|^2 \right) p_{t,0}(x, \tilde{x}) \ \mathrm{d}x \ \mathrm{d}\tilde{x} \ \mathrm{d}t \\
&= \int_\epsilon^T \frac{\kappa(t)}{T - \epsilon} \int_{\mathbb{R}^M} \int_{\mathbb{R}^M} |\nabla_x \log p_{t|0}(x|\tilde{x}) - s(t, \theta, x)|^2 p_{t,0}(x, \tilde{x}) \ \mathrm{d}x \ \mathrm{d}\tilde{x} \ \mathrm{d}t \\
&\quad + \int_\epsilon^T \frac{\kappa(t)}{T - \epsilon} \int_{\mathbb{R}^M} |\nabla \log p_t(x)|^2 p_t(x) \ \mathrm{d}x \ \mathrm{d}t \\
&\quad - \int_\epsilon^T \frac{\kappa(t)}{T - \epsilon} \int_{\mathbb{R}^M} \int_{\mathbb{R}^M} |\nabla_x \log p_{t|0}(x|\tilde{x})|^2 p_{t,0}(x, \tilde{x}) \ \mathrm{d}x \ \mathrm{d}\tilde{x} \ \mathrm{d}t \\
&= \mathbb{E}[\kappa(\tau)|\sigma_\tau^{-1} Z + s(\tau, \theta, m_\tau X_0 + \sigma_\tau Z)|^2] + \int_\epsilon^T \frac{\kappa(t)}{T - \epsilon} \int_{\mathbb{R}^M} |\nabla \log p_t(x)|^2 p_t(x) \ \mathrm{d}x \ \mathrm{d}t \\
&\quad - \int_\epsilon^T \frac{\kappa(t)}{T - \epsilon} \int_{\mathbb{R}^M} \int_{\mathbb{R}^M} |\nabla_x \log p_{t|0}(x|\tilde{x})|^2 p_{t,0}(x, \tilde{x}) \ \mathrm{d}x \ \mathrm{d}\tilde{x} \ \mathrm{d}t.
\end{aligned} \tag{32}$$

We emphasize that we may choose to restrict $t \in [\epsilon, T]$ with $\epsilon \in (0, 1)$ to prevent the divergence of the integrals on the right-hand side of 32.

## B  Additional Discussions about Assumption 4

In this section, we provide additional details about Remark 8, in which we used the multivariate logistic distribution (Malik & Abraham, 1973)

$$g(x) = M! \ \exp\left(-\sum_{k=1}^M x_k\right) \left(1 + \sum_{k=1}^M \exp(-x_k)\right)^{-(M+1)}, \qquad k = 1, \ldots, M, \quad -\infty < x_k < \infty,$$

and derived its score function in 23. The Hessian of $\log(g(x))$ is

$$
\begin{aligned}
\frac{\partial^2}{\partial x_k^2} \log(g(x)) &= -(M+1) \left( \frac{\exp(-x_k)}{(1 + \sum_{k=1}^M \exp(-x_k))} - \frac{\exp(-2x_k)}{(1 + \sum_{k=1}^M \exp(-x_k))^2} \right), \\
\frac{\partial^2}{\partial x_k \partial x_j} \log(g(x)) &= -(M+1) \left( \frac{\exp(-x_k)\exp(-x_j)}{(1 + \sum_{k=1}^M \exp(-x_k))^2} \right), \qquad j, k = 1, \dots, M, \quad j \neq k,
\end{aligned}
\tag{33}
$$

Since the Hessian 33 is bounded, we can conclude that the score function of the multivariate logistic distribution defined in 23 is Lipschitz continuous. By using the same arguments as in Remark 8, we can conclude that $\nabla \log p_t(x)$ defined in 22 is Lipschitz.

Using Assumption 3.b and Assumption 1, we have

$$
\begin{aligned}
& \mathbb{E} \int_0^{T-\epsilon} |\nabla \log p_{T-r}(Y_r^{\mathrm{aux}}) - s(T-r, \hat{\theta}, Y_r^{\mathrm{aux}})|^2 \, \mathrm{d}r \\
& \leq 2\mathbb{E} \int_0^{T-\epsilon} |\nabla \log p_{T-r}(Y_r^{\mathrm{aux}}) - s(T-r, \theta^*, Y_r^{\mathrm{aux}})|^2 \, \mathrm{d}r + 2\mathsf{K}_2^2 \int_0^{T-\epsilon} (1 + 2|T-r|^\alpha)^2 \, \mathbb{E}[|\hat{\theta} - \theta^*|^2] \, \mathrm{d}r \\
& \leq 2\mathbb{E} \int_0^{T-\epsilon} |\nabla \log p_{T-r}(Y_r^{\mathrm{aux}}) - s(T-r, \theta^*, Y_r^{\mathrm{aux}})|^2 \, \mathrm{d}r + 4\mathsf{K}_2^2(T-\epsilon)(1 + 4T^{2\alpha})\widetilde{\varepsilon}_{\mathrm{AL}} < \varepsilon_{\mathrm{SN}},
\end{aligned}
$$

where the error of the first term on the right-hand side above is expected to be small since the Lipschitz $\nabla \log p_t(x)$ is approximated by the function $s$ satisfying Assumption 1. This satisfies Assumption 4.

In addition, we remark that the motivating example in section 3.1 is a special case of the general setting. Indeed, Assumption 1 is satisfied due to Lemma 14 with $\hat{\theta} = \theta_n^\lambda$ being the $n$th-iterate of the SGLD algorithm 14 and $\theta^* := \mu$. Assumption 2 is satisfied by the score function 12. Assumption 3.b is satisfied by the approximating function 13 with $d = M$, $\alpha = 1$, $D_1(\theta, \bar{\theta}) := |\bar{\theta}|$, $D_2(t, \bar{t}) := e^{-t}$, $D_3(t, \bar{t}) := 1$, for any $\theta$, $\bar{\theta} \in \mathbb{R}^d$, $t, \bar{t} \in [0, T]$ and $\mathsf{K}_1 = \mathsf{K}_2 = \mathsf{K}_3 = \mathsf{K}_4 = 1$. Furthermore, both functions satisfy Assumption 4 with $d = M$. Indeed, by Assumption 1 with $\theta^* = \mu$, we have

$$
\begin{aligned}
& \mathbb{E} \int_0^{T-\epsilon} |\nabla \log p_{T-r}(Y_r^{\mathrm{aux}}) - s(T-r, \hat{\theta}, Y_r^{\mathrm{aux}})|^2 \, \mathrm{d}r \\
& \leq 2 \, \mathbb{E} \int_0^{T-\epsilon} |\nabla \log p_{T-r}(Y_r^{\mathrm{aux}}) - s(T-r, \theta^*, Y_r^{\mathrm{aux}})|^2 \, \mathrm{d}r \\
& \quad + 2 \, \mathbb{E} \int_0^{T-\epsilon} |s(T-r, \theta^*, Y_r^{\mathrm{aux}}) - s(T-r, \hat{\theta}, Y_r^{\mathrm{aux}})|^2 \, \mathrm{d}r \\
& = (e^{-2\epsilon} - e^{-2T}) \, \mathbb{E}[|\hat{\theta} - \theta^*|^2] < (e^{-2\epsilon} - e^{-2T})\widetilde{\varepsilon}_{\mathrm{AL}} < \varepsilon_{\mathrm{SN}}.
\end{aligned}
$$

## C  Proofs of the Results for the Multivariate Gaussian Initial Data with Unknown Mean

In this section, we provide the proof of Theorem 1. We start by introducing the results which will be used in the proof of Theorem 1.

### C.1  Preliminary Estimates

We provide the results for the SGLD algorithm 15 with $\lambda$ given in 17, $\beta > 0$, $C_{\mathsf{SGLD},1}$ and $C_{\mathsf{SGLD},2}$ given in Table 1, as well as for the auxiliary process 9, the discrete process 10, and the continuous-time interpolation 11 with $s$ given in 13 and $\gamma \in (0, 1/2]$.

**Proposition 13.** *For any $\theta, \bar{\theta} \in \mathbb{R}^d$ and $\mathbf{x} \in \mathbb{R}^m$,*

$$
\begin{aligned}
|H(\theta, \mathbf{x}) - H(\bar{\theta}, \mathbf{x})| &= 2\sigma_t^2 m_t^2 |\theta - \bar{\theta}|, \\
\langle H(\theta, \mathbf{x}) - H(\bar{\theta}, \mathbf{x}), \theta - \bar{\theta} \rangle &= 2\sigma_t^2 m_t^2 |\theta - \bar{\theta}|^2.
\end{aligned}
$$

Proposition 13 is derived from the definition of the stochastic gradient 14.

The proof of the following lemmas are postponed to Section C.3.

**Lemma 14.** *For any $n \in \mathbb{N}_0$,*

$$\mathbb{E}\left[|\theta_n^\lambda - \theta^*|^2\right] \leq (1 - 2\lambda\mathbb{E}[\sigma_\tau^2 m_\tau^2])^n \mathbb{E}\left[|\theta_0 - \theta^*|^2\right] + dC_{\mathsf{SGLD},1}/\beta + \lambda C_{\mathsf{SGLD},2}.$$

As a consequence of Lemma 14, we derive the following corollary.

**Corollary 15.** *For any $n \in \mathbb{N}_0$,*

$$\mathbb{E}\left[|\theta_n^\lambda|^2\right] \leq 2e^{-2n\lambda\mathbb{E}[\sigma_\tau^2 m_\tau^2]}\mathbb{E}\left[|\theta_0 - \theta^*|^2\right] + 2dC_{\mathsf{SGLD},1}/\beta + 2\lambda C_{\mathsf{SGLD},2} + 2|\theta^*|^2.$$

**Lemma 16.** *It holds that*

$$\sup_{t \geq 0} \mathbb{E}\left[|Y_t^{aux}|^2\right] \leq C_{\mathsf{aux}},$$

*where $C_{\mathsf{aux}} := (8/3)(e^{-2n\lambda\mathbb{E}[\sigma_\tau^2 m_\tau^2]}\mathbb{E}\left[|\theta_0 - \theta^*|^2\right] + dC_{\mathsf{SGLD},1}/\beta + \lambda C_{\mathsf{SGLD},2} + |\theta^*|^2) + 2d.$*

**Lemma 17.** *It holds that*

$$\sup_{k \in \mathbb{N}} \mathbb{E}\left[|Y_k^{EM}|^2\right] \leq C_{\mathsf{EM}},$$

*where $C_{\mathsf{EM}} := 3d + 20(e^{-2n\lambda\mathbb{E}[\sigma_\tau^2 m_\tau^2]}\mathbb{E}\left[|\theta_0 - \theta^*|^2\right] + dC_{\mathsf{SGLD},1}/\beta + \lambda C_{\mathsf{SGLD},2} + |\theta^*|^2).$*

**Lemma 18.** *For any $0 < \gamma \leq 1/2$, one obtains that*

$$\sup_{t \geq 0} \mathbb{E}\left[|\widehat{Y}_t^{EM} - \widehat{Y}_{\lfloor t/\gamma \rfloor \gamma}^{EM}|^2\right] \leq \gamma C_{\mathsf{EMose}},$$

*where $C_{\mathsf{EMose}} := 8d + 56(e^{-2n\lambda\mathbb{E}[\sigma_\tau^2 m_\tau^2]}\mathbb{E}\left[|\theta_0 - \theta^*|^2\right] + dC_{\mathsf{SGLD},1}/\beta + \lambda C_{\mathsf{SGLD},2} + |\theta^*|^2).$*

**Lemma 19.** *It holds that*

$$\sup_{t \geq 0} \mathbb{E}\left[|\widehat{Y}_t^{EM}|^2\right] \leq \widehat{C}_{\mathsf{EM}},$$

*where $\widehat{C}_{\mathsf{EM}} := 18d + 128(e^{-2n\lambda\mathbb{E}[\sigma_\tau^2 m_\tau^2]}\mathbb{E}\left[|\theta_0 - \theta^*|^2\right] + dC_{\mathsf{SGLD},1}/\beta + \lambda C_{\mathsf{SGLD},2} + |\theta^*|^2).$*

## C.2 Proof of the Main Result in the Motivating Example

*Proof of Theorem 1.* We derive the non-asymptotic estimate for $W_2(\mathcal{L}(\widehat{Y}_{K+1}^{EM}), \pi_{\mathsf{D}})$ using the splitting

$$\begin{aligned} W_2(\mathcal{L}(Y_{K+1}^{EM}), \pi_{\mathsf{D}}) &\leq W_2(\pi_{\mathsf{D}}, \mathcal{L}(Y_{t_{K+1}})) + W_2(\mathcal{L}(Y_{t_{K+1}}), \mathcal{L}(\widetilde{Y}_{t_{K+1}})) \\ &\quad + W_2(\mathcal{L}(\widetilde{Y}_{t_{K+1}}), \mathcal{L}(Y_{t_{K+1}}^{aux})) + W_2(\mathcal{L}(Y_{t_{K+1}}^{aux}), \mathcal{L}(Y_{K+1}^{EM})). \end{aligned} \tag{34}$$

Since $t_{K+1} = T$, we have $W_2(\pi_{\mathsf{D}}, \mathcal{L}(Y_{t_{K+1}})) = 0$. We provide upper bounds on the error made by approximating the initial condition of the backward process $Y_0 \sim \mathcal{L}(X_T)$ with $\widetilde{Y}_0 \sim \pi_\infty$, i.e. $W_2(\mathcal{L}(Y_{t_{K+1}}), \mathcal{L}(\widetilde{Y}_{t_{K+1}}))$, the error made by approximating the score function with $s$, i.e. $W_2(\mathcal{L}(\widetilde{Y}_{t_{K+1}}), \mathcal{L}(Y_{t_{K+1}}^{aux}))$, and the discretisation error, i.e. $W_2(\mathcal{L}(Y_{t_{K+1}}^{aux}), \mathcal{L}(Y_{K+1}^{EM}))$, separately.

**Upper bound on $W_2(\mathcal{L}(Y_{t_{K+1}}), \mathcal{L}(\widetilde{Y}_{t_{K+1}}))$.** Applying Itô's formula and using 3 and 4 with the score function given in 12, we have, for any $t \in [0, T]$,

$$\begin{aligned} \mathrm{d}|Y_t - \widetilde{Y}_t|^2 &= 2\langle Y_t - \widetilde{Y}_t, Y_t - \widetilde{Y}_t + 2(\nabla \log p_{T-t}(Y_t) - \nabla \log p_{T-t}(\widetilde{Y}_t))\rangle \mathrm{d}t \\ &= -2|Y_t - \widetilde{Y}_t|^2 \mathrm{d}t. \end{aligned} \tag{35}$$

Integrating and taking expectation both sides in 35 yields

$$\begin{aligned} \mathbb{E}[|Y_t - \widetilde{Y}_t|^2] &= \mathbb{E}[|Y_0 - \widetilde{Y}_0|^2] - 2\int_0^t \mathbb{E}[|Y_s - \widetilde{Y}_s|^2]\mathrm{d}s \\ &\leq e^{-2t}\mathbb{E}[|Y_0 - \widetilde{Y}_0|^2]. \end{aligned} \tag{36}$$

Using 36, the representation 2 with $Z_T \overset{\mathrm{d}}{=} \widetilde{Y}_0$ and $1 - \sigma_t \leq m_t$, we have

$$
\begin{aligned}
\mathbb{E}[|Y_{t_{K+1}} - \widetilde{Y}_{t_{K+1}}|^2] &\leq e^{-2t_{K+1}} \mathbb{E}[|Y_0 - \widetilde{Y}_0|^2] \\
&= e^{-2t_{K+1}} \mathbb{E}[|m_T X_0 + (\sigma_T - 1)\widetilde{Y}_0|^2] \\
&\leq 2e^{-4t_{K+1}} \left( \mathbb{E}[|X_0|^2] + d \right).
\end{aligned}
\tag{37}
$$

Using 37, we have

$$
\begin{aligned}
W_2(\mathcal{L}(Y_{t_{K+1}}), \mathcal{L}(\widetilde{Y}_{t_{K+1}})) &\leq \sqrt{\mathbb{E}[|Y_{t_{K+1}} - \widetilde{Y}_{t_{K+1}}|^2]} \\
&\leq \sqrt{2} e^{-2T} (\sqrt{\mathbb{E}[|X_0|^2]} + \sqrt{d}).
\end{aligned}
\tag{38}
$$

**Upper bound on** $W_2(\mathcal{L}(\widetilde{Y}_{t_{K+1}}), \mathcal{L}(Y_{t_{K+1}}^{\mathbf{aux}}))$. Applying Itô's formula and using the process 4 with the score function 12 and the process 9 with the approximating function 13, we have, for any $t \in [0, T]$,

$$
\begin{aligned}
\mathrm{d}|\widetilde{Y}_t - Y_t^{\mathrm{aux}}|^2 &= 2\langle \widetilde{Y}_t - Y_t^{\mathrm{aux}}, \widetilde{Y}_t - Y_t^{\mathrm{aux}} + 2(\nabla \log p_{T-t}(\widetilde{Y}_t) - s(T - t, \hat{\theta}, Y_t^{\mathrm{aux}}))\rangle \mathrm{d}t \\
&= -2|\widetilde{Y}_t - Y_t^{\mathrm{aux}}|^2 \mathrm{d}t + 4\langle \widetilde{Y}_t - Y_t^{\mathrm{aux}}, m_{T-t}(\mu - \hat{\theta})\rangle \mathrm{d}t.
\end{aligned}
\tag{39}
$$

Integrating and taking expectation on both sides in 39 and using that the minimiser $\theta^\star = \mu$, we have

$$
\mathbb{E}[|\widetilde{Y}_t - Y_t^{\mathrm{aux}}|^2] = -2\int_0^t \mathbb{E}[|\widetilde{Y}_s - Y_s^{\mathrm{aux}}|^2]\mathrm{d}s + 4\int_0^t \mathbb{E}[\langle \widetilde{Y}_s - Y_s^{\mathrm{aux}}, m_{T-s}(\theta^* - \hat{\theta})\rangle]\mathrm{d}s.
\tag{40}
$$

Differentiating both sides of 40 and using Young's inequality, we obtain

$$
\begin{aligned}
\frac{\mathrm{d}}{\mathrm{d}t} \mathbb{E}[|\widetilde{Y}_t - Y_t^{\mathrm{aux}}|^2] &= -2\mathbb{E}[|\widetilde{Y}_t - Y_t^{\mathrm{aux}}|^2] + 4\mathbb{E}[\langle \widetilde{Y}_t - Y_t^{\mathrm{aux}}, m_{T-t}(\theta^* - \hat{\theta})\rangle] \\
&\leq -\mathbb{E}[|\widetilde{Y}_t - Y_t^{\mathrm{aux}}|^2] + 4e^{-2T}e^{2t}\mathbb{E}[|\theta^* - \hat{\theta}|^2],
\end{aligned}
$$

which implies that

$$
\frac{\mathrm{d}}{\mathrm{d}t}(e^t \mathbb{E}[|\widetilde{Y}_t - Y_t^{\mathrm{aux}}|^2]) \leq 4e^{-2T}e^{3t}\mathbb{E}[|\theta^* - \hat{\theta}|^2].
$$

Integrating both sides and using Lemma 14 yields

$$
\begin{aligned}
&\mathbb{E}[|\widetilde{Y}_t - Y_t^{\mathrm{aux}}|^2] \\
&\leq (4/3)e^{-2T}(e^{2t} - e^{-t})\mathbb{E}[|\theta^* - \hat{\theta}|^2] \\
&\leq (4/3)e^{-2(T-t)}(e^{-2n\lambda \mathbb{E}[\sigma_\tau^2 m_\tau^2]}\mathbb{E}[|\theta_0 - \theta^*|^2] + dC_{\mathsf{SGLD},1}/\beta + \lambda C_{\mathsf{SGLD},2}).
\end{aligned}
\tag{41}
$$

Using 41, we have

$$
\begin{aligned}
W_2(\mathcal{L}(\widetilde{Y}_{t_{K+1}}), \mathcal{L}(Y_{t_{K+1}}^{\mathrm{aux}})) &\leq \sqrt{\mathbb{E}[|\widetilde{Y}_{t_{K+1}} - Y_{t_{K+1}}^{\mathrm{aux}}|^2]} \\
&\leq \sqrt{4/3}(e^{-2n\lambda \mathbb{E}[\sigma_\tau^2 m_\tau^2]}\mathbb{E}[|\theta_0 - \theta^*|^2] + dC_{\mathsf{SGLD},1}/\beta + \lambda C_{\mathsf{SGLD},2})^{1/2}.
\end{aligned}
\tag{42}
$$

**Upper bound on** $W_2(\mathcal{L}(Y_{t_{K+1}}^{\mathbf{aux}}), \mathcal{L}(\widehat{Y}_{K+1}^{\mathbf{EM}}))$. Applying Itô's formula and using the processes 9 and 11 with the approximating function $s$ given in 13, we have, for any $t \in [0, T]$,

$$
\begin{aligned}
\mathrm{d}|Y_t^{\mathrm{aux}} - \widehat{Y}_t^{\mathrm{EM}}|^2 &= 2\langle Y_t^{\mathrm{aux}} - \widehat{Y}_t^{\mathrm{EM}}, Y_t^{\mathrm{aux}} - \widehat{Y}_{\lfloor t/\gamma \rfloor \gamma}^{\mathrm{EM}}\rangle \mathrm{d}t \\
&\quad + 4\langle Y_t^{\mathrm{aux}} - \widehat{Y}_t^{\mathrm{EM}}, s(T - t, \hat{\theta}, Y_t^{\mathrm{aux}}) - s(T - \lfloor t/\gamma \rfloor \gamma, \hat{\theta}, \widehat{Y}_{\lfloor t/\gamma \rfloor \gamma}^{\mathrm{EM}})\rangle \mathrm{d}t \\
&= -2\langle Y_t^{\mathrm{aux}} - \widehat{Y}_t^{\mathrm{EM}}, Y_t^{\mathrm{aux}} - \widehat{Y}_{\lfloor t/\gamma \rfloor \gamma}^{\mathrm{EM}}\rangle \mathrm{d}t \\
&\quad + 4\langle Y_t^{\mathrm{aux}} - \widehat{Y}_t^{\mathrm{EM}}, (m_{T-t} - m_{T-\lfloor t/\gamma \rfloor \gamma})\hat{\theta}\rangle \mathrm{d}t.
\end{aligned}
\tag{43}
$$

Integrating both sides and taking expectation in 43 yields

$$
\begin{aligned}
\mathbb{E}\left[|Y_t^{\mathrm{aux}} - \widehat{Y}_t^{\mathrm{EM}}|^2\right] &= -2\int_0^t \mathbb{E}\left[|Y_s^{\mathrm{aux}} - \widehat{Y}_s^{\mathrm{EM}}|^2\right]\mathrm{d}s - 2\int_0^t \mathbb{E}\left[\left\langle Y_s^{\mathrm{aux}} - \widehat{Y}_s^{\mathrm{EM}}, \widehat{Y}_s^{\mathrm{EM}} - \widehat{Y}_{\lfloor s/\gamma\rfloor\gamma}^{\mathrm{EM}}\right\rangle\right]\mathrm{d}s \\
&\quad + 4\int_0^t \mathbb{E}\left[\left\langle Y_s^{\mathrm{aux}} - \widehat{Y}_s^{\mathrm{EM}}, (m_{T-s} - m_{T-\lfloor s/\gamma\rfloor\gamma})\hat{\theta}\right\rangle\right]\mathrm{d}s.
\end{aligned}
\tag{44}
$$

Differentiating both sides in 44, using Young's inequality and $m_{T-t} - m_{T-\lfloor t/\gamma\rfloor\gamma} \leq \gamma m_{T-t}$, we have

$$
\begin{aligned}
\frac{\mathrm{d}}{\mathrm{d}t}\mathbb{E}[|Y_t^{\mathrm{aux}} - \widehat{Y}_t^{\mathrm{EM}}|^2] &= -2\mathbb{E}[|Y_t^{\mathrm{aux}} - \widehat{Y}_t^{\mathrm{EM}}|^2] - 2\mathbb{E}\left[\left\langle Y_t^{\mathrm{aux}} - \widehat{Y}_t^{\mathrm{EM}}, \sqrt{2}\int_{\lfloor t/\gamma\rfloor\gamma}^t \mathrm{d}\overline{B}_s\right\rangle\right] \\
&\quad - 2\mathbb{E}\left[\left\langle Y_t^{\mathrm{aux}} - \widehat{Y}_t^{\mathrm{EM}}, \int_{\lfloor t/\gamma\rfloor\gamma}^t (-\widehat{Y}_{\lfloor s/\gamma\rfloor\gamma}^{\mathrm{EM}} + 2m_{T-\lfloor s/\gamma\rfloor\gamma}\hat{\theta})\mathrm{d}s\right\rangle\right] \\
&\quad + 4\mathbb{E}[\langle Y_t^{\mathrm{aux}} - \widehat{Y}_t^{\mathrm{EM}}, (m_{T-t} - m_{T-\lfloor t/\gamma\rfloor\gamma})\hat{\theta}\rangle] \\
&\leq -\mathbb{E}[|Y_t^{\mathrm{aux}} - \widehat{Y}_t^{\mathrm{EM}}|^2] - 2\mathbb{E}\left[\left\langle Y_t^{\mathrm{aux}} - \widehat{Y}_t^{\mathrm{EM}}, \sqrt{2}\int_{\lfloor t/\gamma\rfloor\gamma}^t \mathrm{d}\overline{B}_s\right\rangle\right] \\
&\quad + 2\gamma^2\mathbb{E}[(-\widehat{Y}_{\lfloor t/\gamma\rfloor\gamma}^{\mathrm{EM}} + 2m_{T-\lfloor t/\gamma\rfloor\gamma}\hat{\theta})^2] + 8\gamma^2 e^{-2(T-t)}\mathbb{E}[|\hat{\theta}|^2] \\
&\leq -\mathbb{E}[|Y_t^{\mathrm{aux}} - \widehat{Y}_t^{\mathrm{EM}}|^2] - 2\mathbb{E}\left[\left\langle Y_t^{\mathrm{aux}} - \widehat{Y}_t^{\mathrm{EM}}, \sqrt{2}\int_{\lfloor t/\gamma\rfloor\gamma}^t \mathrm{d}\overline{B}_s\right\rangle\right] \\
&\quad + 4\gamma^2\mathbb{E}[|\widehat{Y}_{\lfloor t/\gamma\rfloor\gamma}^{\mathrm{EM}}|^2] + 24\gamma^2\mathbb{E}[|\hat{\theta}|^2].
\end{aligned}
\tag{45}
$$

We derive an upper bound for the second term on the right-hand side in 45. Using Cauchy-Schwarz inequality, Itô formula applied to $tB_t$ and Young's inequality, we have

$$
\begin{aligned}
&- 2\mathbb{E}\left[\left\langle Y_t^{\mathrm{aux}} - \widehat{Y}_t^{\mathrm{EM}}, \sqrt{2}\int_{\lfloor t/\gamma\rfloor\gamma}^t \mathrm{d}\overline{B}_s\right\rangle\right] \\
&= -2\mathbb{E}\left[\left\langle (Y_t^{\mathrm{aux}} - Y_{\lfloor t/\gamma\rfloor\gamma}^{\mathrm{aux}}) - (\widehat{Y}_t^{\mathrm{EM}} - \widehat{Y}_{\lfloor t/\gamma\rfloor\gamma}^{\mathrm{EM}}), \sqrt{2}\int_{\lfloor t/\gamma\rfloor\gamma}^t \mathrm{d}\overline{B}_s\right\rangle\right] \\
&= -2\mathbb{E}\left[\left\langle \int_{\lfloor t/\gamma\rfloor\gamma}^t (-Y_s^{\mathrm{aux}} + 2m_{T-s}\hat{\theta})\mathrm{d}s, \sqrt{2}\int_{\lfloor t/\gamma\rfloor\gamma}^t \mathrm{d}\overline{B}_s\right\rangle\right] \\
&\quad + 2\mathbb{E}\left[\left\langle \int_{\lfloor t/\gamma\rfloor\gamma}^t (-\widehat{Y}_{\lfloor s/\gamma\rfloor\gamma}^{\mathrm{EM}} + 2m_{T-\lfloor s/\gamma\rfloor\gamma}\hat{\theta})\mathrm{d}s, \sqrt{2}\int_{\lfloor t/\gamma\rfloor\gamma}^t \mathrm{d}\overline{B}_s\right\rangle\right] \\
&= 2\mathbb{E}\left[\left\langle \int_{\lfloor t/\gamma\rfloor\gamma}^t (Y_s^{\mathrm{aux}} - \widehat{Y}_{\lfloor s/\gamma\rfloor\gamma}^{\mathrm{EM}})\mathrm{d}s, \sqrt{2}\int_{\lfloor t/\gamma\rfloor\gamma}^t \mathrm{d}\overline{B}_s\right\rangle\right] \\
&= 2\mathbb{E}\left[\left\langle \int_{\lfloor t/\gamma\rfloor\gamma}^t (Y_s^{\mathrm{aux}} - Y_{\lfloor s/\gamma\rfloor\gamma}^{\mathrm{aux}})\mathrm{d}s, \sqrt{2}\int_{\lfloor t/\gamma\rfloor\gamma}^t \mathrm{d}\overline{B}_s\right\rangle\right] \\
&= 2\mathbb{E}\left[\left\langle \int_{\lfloor t/\gamma\rfloor\gamma}^t \int_{\lfloor s/\gamma\rfloor\gamma}^s (-Y_\nu^{\mathrm{aux}})\mathrm{d}\nu\mathrm{d}s, \sqrt{2}\int_{\lfloor t/\gamma\rfloor\gamma}^t \mathrm{d}\overline{B}_s\right\rangle\right] \\
&\quad + 2\mathbb{E}\left[\left\langle \int_{\lfloor t/\gamma\rfloor\gamma}^t \sqrt{2}\int_{\lfloor s/\gamma\rfloor\gamma}^s \mathrm{d}\overline{B}_\nu\mathrm{d}s, \sqrt{2}\int_{\lfloor t/\gamma\rfloor\gamma}^t \mathrm{d}\overline{B}_s\right\rangle\right]
\end{aligned}
$$

$$\leq 2\sqrt{2}\left(\mathbb{E}\left[\left|\int_{\lfloor t/\gamma\rfloor\gamma}^{t}\int_{\lfloor s/\gamma\rfloor\gamma}^{s}(-Y_{\nu}^{\mathrm{aux}})\mathrm{d}\nu\mathrm{d}s\right|^{2}\right]\right)^{1/2}\left(\mathbb{E}\left[\left|\int_{\lfloor t/\gamma\rfloor\gamma}^{t}\mathrm{d}\overline{B}_{s}\right|^{2}\right]\right)^{1/2}$$
$$+4\mathbb{E}\left[\left\langle t\int_{\lfloor t/\gamma\rfloor\gamma}^{t}\mathrm{d}\overline{B}_{s}-\int_{\lfloor t/\gamma\rfloor\gamma}^{t}s\mathrm{d}\overline{B}_{s},\int_{\lfloor t/\gamma\rfloor\gamma}^{t}\mathrm{d}\overline{B}_{s}\right\rangle\right] \quad (46)$$
$$\leq \sqrt{2}\gamma^{5/2}\left(\sup_{t\geq0}\mathbb{E}\left[|Y_{t}^{\mathrm{aux}}|^{2}\right]+d\right)+2d\gamma^{2}.$$

Substituting 46 into 45 yields

$$\frac{\mathrm{d}}{\mathrm{d}t}\mathbb{E}[|Y_{t}^{\mathrm{aux}}-\widehat{Y}_{t}^{\mathrm{EM}}|^{2}]\leq -\mathbb{E}[|Y_{t}^{\mathrm{aux}}-\widehat{Y}_{t}^{\mathrm{EM}}|^{2}]+\sqrt{2}\gamma^{5/2}\sup_{t\geq0}\mathbb{E}[|Y_{t}^{\mathrm{aux}}|^{2}]$$
$$+\sqrt{2}d\gamma^{5/2}+2d\gamma^{2}+4\gamma^{2}\mathbb{E}[|\widehat{Y}_{\lfloor t/\gamma\rfloor\gamma}^{\mathrm{EM}}|^{2}]+24\gamma^{2}\mathbb{E}[|\hat{\theta}|^{2}].$$

Thus,

$$\frac{\mathrm{d}}{\mathrm{d}t}(e^{t}\mathbb{E}[|Y_{t}^{\mathrm{aux}}-\widehat{Y}_{t}^{\mathrm{EM}}|^{2}])\leq e^{t}\gamma^{2}\left(\sqrt{2}\sup_{t\geq0}\mathbb{E}[|Y_{t}^{\mathrm{aux}}|^{2}]+3d+4\mathbb{E}[|\widehat{Y}_{\lfloor t/\gamma\rfloor\gamma}^{\mathrm{EM}}|^{2}]+24\mathbb{E}[|\hat{\theta}|^{2}]\right). \quad (47)$$

Integrating both sides in 47 and using Lemma 16, Lemma 17 and Corollary 15, yield

$$\mathbb{E}[|Y_{t}^{\mathrm{aux}}-\widehat{Y}_{t}^{\mathrm{EM}}|^{2}]$$
$$\leq \sqrt{2}\gamma^{2}(2d+(8/3)(e^{-2n\lambda\mathbb{E}[\sigma_{\tau}^{2}m_{\tau}^{2}]}\mathbb{E}[|\theta_{0}-\theta^{*}|^{2}]+dC_{\mathsf{SGLD},1}/\beta+\lambda C_{\mathsf{SGLD},2}+|\theta^{*}|^{2}))$$
$$+3d\gamma^{2}+4\gamma^{2}(3d+20(e^{-2n\lambda\mathbb{E}[\sigma_{\tau}^{2}m_{\tau}^{2}]}\mathbb{E}[|\theta_{0}-\theta^{*}|^{2}]+dC_{\mathsf{SGLD},1}/\beta+\lambda C_{\mathsf{SGLD},2}+|\theta^{*}|^{2})) \quad (48)$$
$$+48\gamma^{2}(e^{-2n\lambda\mathbb{E}[\sigma_{\tau}^{2}m_{\tau}^{2}]}\mathbb{E}[|\theta_{0}-\theta^{*}|^{2}]+dC_{\mathsf{SGLD},1}/\beta+\lambda C_{\mathsf{SGLD},2}+|\theta^{*}|^{2})$$
$$\leq 18d\gamma^{2}+132\gamma^{2}(e^{-2n\lambda\mathbb{E}[\sigma_{\tau}^{2}m_{\tau}^{2}]}\mathbb{E}[|\theta_{0}-\theta^{*}|^{2}]+dC_{\mathsf{SGLD},1}/\beta+\lambda C_{\mathsf{SGLD},2}+|\theta^{*}|^{2}).$$

By using 48, we have

$$W_{2}(\mathcal{L}(Y_{t_{K+1}}^{\mathrm{aux}}),\mathcal{L}(\widehat{Y}_{K+1}^{\mathrm{EM}}))$$
$$\leq \sqrt{\mathbb{E}[|Y_{t_{K+1}}^{\mathrm{aux}}-\widehat{Y}_{t_{K+1}}^{\mathrm{EM}}|^{2}]} \quad (49)$$
$$\leq \gamma(18d+132(e^{-2n\lambda\mathbb{E}[\sigma_{\tau}^{2}m_{\tau}^{2}]}\mathbb{E}[|\theta_{0}-\theta^{*}|^{2}]+dC_{\mathsf{SGLD},1}/\beta+\lambda C_{\mathsf{SGLD},2}+|\theta^{*}|^{2}))^{1/2}.$$

**Final upper bound on** $W_{2}(\mathcal{L}(\widehat{Y}_{K+1}^{\mathbf{EM}}),\pi_{\mathsf{D}})$. Substituting 38, 42, 49 into 34 yields

$$W_{2}(\mathcal{L}(\widehat{Y}_{K+1}^{\mathrm{EM}}),\pi_{\mathsf{D}})$$
$$\leq \sqrt{2}e^{-2T}(\sqrt{\mathbb{E}[|X_{0}|^{2}]}+\sqrt{d})$$
$$+\sqrt{4/3}(e^{-2n\lambda\mathbb{E}[\sigma_{\tau}^{2}m_{\tau}^{2}]}\mathbb{E}[|\theta_{0}-\theta^{*}|^{2}]+dC_{\mathsf{SGLD},1}/\beta+\lambda C_{\mathsf{SGLD},2})^{1/2}$$
$$+\gamma(18d+132(e^{-2n\lambda\mathbb{E}[\sigma_{\tau}^{2}m_{\tau}^{2}]}\mathbb{E}[|\theta_{0}-\theta^{*}|^{2}]+dC_{\mathsf{SGLD},1}/\beta+\lambda C_{\mathsf{SGLD},2}+|\theta^{*}|^{2}))^{1/2} \quad (50)$$
$$\leq \sqrt{2}e^{-2T}(\sqrt{\mathbb{E}[|X_{0}|^{2}]}+\sqrt{d})$$
$$+(\sqrt{4/3}+2\sqrt{33})(e^{-2n\lambda\mathbb{E}[\sigma_{\tau}^{2}m_{\tau}^{2}]}\mathbb{E}[|\theta_{0}-\theta^{*}|^{2}]+dC_{\mathsf{SGLD},1}/\beta+\lambda C_{\mathsf{SGLD},2})^{1/2}$$
$$+\gamma(18d+132|\theta^{*}|^{2})^{1/2}.$$

The bound for $W_{2}(\mathcal{L}(\widehat{Y}_{K+1}^{\mathrm{EM}}),\pi_{\mathsf{D}})$ in 50 can be made arbitrarily small by appropriately choosing parameters including $T,\beta,\lambda,n$ and $\gamma$. More precisely, for any $\delta>0$, we first choose $T>T_{\delta}$ with $T_{\delta}$ given explicitly in Table 1 such that

$$\sqrt{2}e^{-2T}(\sqrt{\mathbb{E}[|X_{0}|^{2}]}+\sqrt{d})<\delta/4. \quad (51)$$

Next, we choose $\beta \geq \beta_\delta$ and $0 < \lambda \leq \lambda_\delta$ with $\beta_\delta$ and $\lambda_\delta$ given explicitly in Table 1, and, in the case where $\lambda = \lambda_\delta$, we choose $n \geq n_\delta$ with $n_\delta$ given explicitly in Table 1 such that

$$
\begin{aligned}
&(\sqrt{4/3} + 2\sqrt{33})(e^{-2n\lambda\mathbb{E}[\sigma_\tau^2 m_\tau^2]}\mathbb{E}[|\theta_0 - \theta^*|^2] + dC_{\mathsf{SGLD},1}/\beta + \lambda C_{\mathsf{SGLD},2})^{1/2} \\
&\leq (\sqrt{4/3} + 2\sqrt{33})\sqrt{d/(\beta\mathbb{E}[\sigma_\tau^2 m_\tau^2])} \\
&\quad + (\sqrt{4/3} + 2\sqrt{33})(4\lambda\mathbb{E}[\sigma_\tau^4 m_\tau^2(\sigma_\tau^{-1}|Z| + m_\tau|X_0| + \sigma_\tau|Z| + m_\tau|\theta^*|)^2]/(\mathbb{E}[\sigma_\tau^2 m_\tau^2]))^{1/2} \\
&\quad + (\sqrt{4/3} + 2\sqrt{33})e^{-n\lambda\mathbb{E}[\sigma_\tau^2 m_\tau^2]}\sqrt{\mathbb{E}[|\theta_0 - \theta^*|^2]} \\
&\leq \delta/12 + \delta/12 + \delta/12 = \delta/4.
\end{aligned}
\tag{52}
$$

Finally, we choose $0 < \gamma < \gamma_\delta$ with $\gamma_\delta$ given explicitly in Table 1 such that

$$
\gamma(18d + 132|\theta^*|^2)^{1/2} < \delta/4.
\tag{53}
$$

Using 51, 52 and 53 in 50, we obtain $W_2(\mathcal{L}(\widehat{Y}_{K+1}^{\mathrm{EM}}), \pi_{\mathsf{D}}) < \delta$. $\qquad\square$

### C.3 Proof of the Preliminary Results

We provide the proofs of the results of Appendix C.1.

*Proof of Lemma 14.* By 15, we have for any $n \in \mathbb{N}_0$,

$$
\begin{aligned}
|\theta_{n+1}^\lambda - \theta^*|^2 &= \left|\theta_n^\lambda - \theta^* - \lambda H(\theta_n^\lambda, \mathbf{X}_{n+1}) + \sqrt{2\lambda/\beta}\,\xi_{n+1}\right|^2 \\
&= \left|\theta_n^\lambda - \theta^*\right|^2 + 2\left\langle \theta_n^\lambda - \theta^*, -\lambda H(\theta_n^\lambda, \mathbf{X}_{n+1}) + \sqrt{2\lambda/\beta}\,\xi_{n+1}\right\rangle \\
&\quad + \left|-\lambda H(\theta_n^\lambda, \mathbf{X}_{n+1}) + \sqrt{2\lambda/\beta}\xi_{n+1}\right|^2 \\
&= \left|\theta_n^\lambda - \theta^*\right|^2 - 2\lambda\left\langle \theta_n^\lambda - \theta^*, H(\theta_n^\lambda, \mathbf{X}_{n+1}) - H(\theta^*, \mathbf{X}_{n+1})\right\rangle \\
&\quad - 2\lambda\left\langle \theta_n^\lambda - \theta^*, H(\theta^*, \mathbf{X}_{n+1})\right\rangle + 2\sqrt{2\lambda/\beta}\left\langle \theta_n^\lambda - \theta^*, \xi_{n+1}\right\rangle + \lambda^2\left|H(\theta_n^\lambda, \mathbf{X}_{n+1})\right|^2 \\
&\quad - 2\lambda\sqrt{2\lambda/\beta}\left\langle H(\theta_n^\lambda, \mathbf{X}_{n+1}), \xi_{n+1}\right\rangle + (2\lambda/\beta)\left|\xi_{n+1}\right|^2.
\end{aligned}
\tag{54}
$$

Taking conditional expectation on both sides in 54 yields

$$
\begin{aligned}
\mathbb{E}\left[|\theta_{n+1}^\lambda - \theta^*|^2 \mid \theta_n^\lambda\right] &= \left|\theta_n^\lambda - \theta^*\right|^2 - 2\lambda\mathbb{E}\left[\left\langle \theta_n^\lambda - \theta^*, H(\theta_n^\lambda, \mathbf{X}_{n+1}) - H(\theta^*, \mathbf{X}_{n+1})\right\rangle \mid \theta_n^\lambda\right] \\
&\quad + 2\lambda^2\mathbb{E}\left[\left|H(\theta_n^\lambda, \mathbf{X}_{n+1}) - H(\theta^*, \mathbf{X}_{n+1})\right|^2 \mid \theta_n^\lambda\right] \\
&\quad + 2\lambda^2\mathbb{E}\left[\left|H(\theta^*, \mathbf{X}_{n+1})\right|^2 \mid \theta_n^\lambda\right] + 2\lambda d/\beta.
\end{aligned}
$$

Recall that $0 < \lambda \leq \min\{\mathbb{E}[\sigma_\tau^2 m_\tau^2]/(4\mathbb{E}[\sigma_\tau^4 m_\tau^4]), 1/(2\mathbb{E}[\sigma_\tau^2 m_\tau^2])\}$. Using Proposition 13 and the stochastic gradient 14, we have

$$
\begin{aligned}
\mathbb{E}\left[|\theta_{n+1}^\lambda - \theta^*|^2 \mid \theta_n^\lambda\right] &\leq \left(1 - 2\lambda\mathbb{E}\left[\sigma_\tau^2 m_\tau^2\right]\right)|\theta_n^\lambda - \theta^*|^2 + 2\lambda d/\beta \\
&\quad - 2\lambda\mathbb{E}\left[\sigma_\tau^2 m_\tau^2\right]|\theta_n^\lambda - \theta^*|^2 + 8\lambda^2\mathbb{E}\left[\sigma_\tau^4 m_\tau^4\right]\left|\theta_n^\lambda - \theta^*\right|^2 \\
&\quad + 8\lambda^2\mathbb{E}\left[\sigma_\tau^4 m_\tau^2\left(\sigma_\tau^{-1}|Z| + m_\tau|X_0| + \sigma_\tau|Z| + m_\tau|\theta^*|\right)^2\right] \\
&\leq \left(1 - 2\lambda\mathbb{E}\left[\sigma_\tau^2 m_\tau^2\right]\right)|\theta_n^\lambda - \theta^*|^2 + 2\lambda d/\beta \\
&\quad + 8\lambda^2\mathbb{E}\left[\sigma_\tau^4 m_\tau^2\left(\sigma_\tau^{-1}|Z| + m_\tau|X_0| + \sigma_\tau|Z| + m_\tau|\theta^*|\right)^2\right].
\end{aligned}
$$

This implies that

$$
\mathbb{E}\left[|\theta_{n+1}^\lambda - \theta^*|^2\right] \leq \left(1 - 2\lambda\mathbb{E}\left[\sigma_\tau^2 m_\tau^2\right]\right)^{n+1}\mathbb{E}\left[|\theta_0 - \theta^*|^2\right] + dC_{\mathsf{SGLD},1}/\beta + \lambda C_{\mathsf{SGLD},2}.
$$

$\qquad\square$

*Proof of Lemma 16.* Applying Itô's formula and using the process 9 with $s$ given in 13, we have, for any $t \in [0, T]$,

$$\mathrm{d}|Y_t^{\mathrm{aux}}|^2 = -2|Y_t^{\mathrm{aux}}|^2\mathrm{d}t + 2\langle Y_t^{\mathrm{aux}}, 2m_{T-t}\hat{\theta}\rangle\mathrm{d}t + 2\langle Y_t^{\mathrm{aux}}, \sqrt{2}\mathrm{d}\overline{B}_t\rangle + 2d\mathrm{d}t.$$

Integrating both sides and taking expectation, we have

$$\mathbb{E}\left[|Y_t^{\mathrm{aux}}|^2\right] = -2\int_0^t \mathbb{E}\left[|Y_s^{\mathrm{aux}}|^2\right]\mathrm{d}s + 2\int_0^t \mathbb{E}[\langle Y_s^{\mathrm{aux}}, 2m_{T-s}\hat{\theta}\rangle]\mathrm{d}s + \mathbb{E}\left[|Y_0^{\mathrm{aux}}|^2\right] + 2dt.$$

Then, differentiating both sides, we have

$$\frac{\mathrm{d}}{\mathrm{d}t}\mathbb{E}\left[|Y_t^{\mathrm{aux}}|^2\right] = -2\mathbb{E}\left[|Y_t^{\mathrm{aux}}|^2\right] + 2\mathbb{E}[\langle Y_t^{\mathrm{aux}}, 2m_{T-t}\hat{\theta}\rangle] + 2d$$
$$\leq -\mathbb{E}\left[|Y_t^{\mathrm{aux}}|^2\right] + 4m_{T-t}^2\mathbb{E}[|\hat{\theta}|^2] + 2d,$$

which, by rearranging the terms, yields

$$\frac{\mathrm{d}}{\mathrm{d}t}(e^t\mathbb{E}\left[|Y_t^{\mathrm{aux}}|^2\right]) \leq 4e^{-2T}e^{3t}\mathbb{E}[|\hat{\theta}|^2] + 2de^t.$$

Integrating both side and using Corollary 15, we obtain

$$\mathbb{E}\left[|Y_t^{\mathrm{aux}}|^2\right] \leq e^{-t}(\mathbb{E}\left[|Y_0^{\mathrm{aux}}|^2\right] + (4/3)e^{-2T}(e^{3t}-1)\mathbb{E}[|\hat{\theta}|^2] + 2d(e^t-1))$$
$$\leq d(2-e^{-t}) + (4/3)e^{-2T}(e^{2t}-e^{-t})\mathbb{E}[|\hat{\theta}|^2]$$
$$\leq 2d + e^{-2(T-t)}(8/3)(e^{-2n\lambda\mathbb{E}[\sigma_\tau^2 m_\tau^2]}\mathbb{E}[|\theta_0-\theta^*|^2] + dC_{\mathsf{SGLD},1}/\beta + \lambda C_{\mathsf{SGLD},2} + |\theta^*|^2).$$

$\square$

*Proof of Lemma 17.* Using the process 10 with the approximating function $s$ given in 13, we have

$$|Y_{k+1}^{\mathrm{EM}}|^2 = |Y_k^{\mathrm{EM}} + \gamma(-Y_k^{\mathrm{EM}} + 2m_{T-k}\hat{\theta})|^2 + 2\gamma|\bar{Z}_{k+1}|^2 \tag{55}$$
$$+ 2\langle Y_k^{\mathrm{EM}} + \gamma(-Y_k^{\mathrm{EM}} + 2m_{T-k}\hat{\theta}), \sqrt{2\gamma}\bar{Z}_{k+1}\rangle.$$

Taking conditional expectation on both sides in 55, using the independence of $Y_k^{\mathrm{EM}}$ and $\bar{Z}_{k+1}$, Young's inequality and $0 < \gamma \leq 1/2$, we obtain

$$\mathbb{E}\left[|Y_{k+1}^{\mathrm{EM}}|^2 \mid Y_k^{\mathrm{EM}}\right] = |Y_k^{\mathrm{EM}}|^2 + 2\langle Y_k^{\mathrm{EM}}, \gamma(-Y_k^{\mathrm{EM}} + 2m_{T-t_k}\hat{\theta})\rangle$$
$$+ \gamma^2|Y_k^{\mathrm{EM}}|^2 + 4\gamma^2 m_{T-k}^2|\hat{\theta}|^2 - 2\gamma^2\langle Y_k^{\mathrm{EM}}, 2m_{T-t_k}\hat{\theta}\rangle + 2\gamma d$$
$$= (1-2\gamma)|Y_k^{\mathrm{EM}}|^2 + 4\gamma^2 m_{T-k}^2|\hat{\theta}|^2 + 2\gamma d$$
$$+ \gamma^2|Y_k^{\mathrm{EM}}|^2 + 2\gamma(1-\gamma)\langle Y_k^{\mathrm{EM}}, 2m_{T-k}\hat{\theta}\rangle$$
$$\leq (1-\gamma)|Y_k^{\mathrm{EM}}|^2 + 4\gamma(\gamma+2)m_{T-k}^2|\hat{\theta}|^2 + 2\gamma d$$
$$- \gamma|Y_k^{\mathrm{EM}}|^2 + \gamma^2|Y_k^{\mathrm{EM}}|^2 + (\gamma/2)|Y_k^{\mathrm{EM}}|^2$$
$$\leq (1-\gamma)|Y_k^{\mathrm{EM}}|^2 + 4\gamma(\gamma+2)|\hat{\theta}|^2 + 2\gamma d.$$

Thus,

$$\mathbb{E}\left[|Y_{k+1}^{\mathrm{EM}}|^2\right] \leq (1-\gamma)^{k+1}d + 10|\hat{\theta}|^2 + 2d. \tag{56}$$

Using Corollary 15 in 56 yields

$$\mathbb{E}\left[|Y_k^{\mathrm{EM}}|^2\right] \leq (1-\gamma)^k d + 2d + 20(e^{-2n\lambda\mathbb{E}[\sigma_\tau^2 m_\tau^2]}\mathbb{E}\left[|\theta_0-\theta^*|^2\right] + dC_{\mathsf{SGLD},1}/\beta + \lambda C_{\mathsf{SGLD},2} + |\theta^*|^2).$$

$\square$

*Proof of Lemma 18.* Using the process 11 with the approximating function $s$ given in 13, Lemma 17, Corollary 15 and $\gamma \in (0, 1/2]$, we have, for any $t \in [0, T]$,

$$
\begin{aligned}
&\mathbb{E}\left[|\widehat{Y}_t^{\mathrm{EM}} - \widehat{Y}_{\lfloor t/\gamma \rfloor \gamma}^{\mathrm{EM}}|^2\right] \\
&= \mathbb{E}\left[\left|\int_{\lfloor t/\gamma \rfloor \gamma}^t (-\widehat{Y}_{\lfloor s/\gamma \rfloor \gamma}^{\mathrm{EM}} + 2m_{T-\lfloor s/\gamma \rfloor \gamma}\hat{\theta})\mathrm{d}s + \sqrt{2}\int_{\lfloor t/\gamma \rfloor \gamma}^t \mathrm{d}\overline{B}_s\right|^2\right] \\
&\leq \gamma^2 \mathbb{E}\left[\left|-\widehat{Y}_{\lfloor t/\gamma \rfloor \gamma}^{\mathrm{EM}} + 2m_{T-\lfloor t/\gamma \rfloor \gamma}\hat{\theta}\right|^2\right] + 2d\gamma \\
&\leq 2\gamma^2 \mathbb{E}\left[|\widehat{Y}_{\lfloor t/\gamma \rfloor \gamma}^{\mathrm{EM}}|^2\right] + 8\gamma^2 \mathbb{E}\left[|\hat{\theta}|^2\right] + 2d\gamma \\
&\leq 6\gamma^2 d + 40\gamma^2 (e^{-2n\lambda \mathbb{E}[\sigma_\tau^2 m_\tau^2]}\mathbb{E}\left[|\theta_0 - \theta^*|^2\right] + dC_{\mathsf{SGLD},1}/\beta + \lambda C_{\mathsf{SGLD},2} + |\theta^*|^2) \\
&\quad + 16\gamma^2 (e^{-2n\lambda \mathbb{E}[\sigma_\tau^2 m_\tau^2]}\mathbb{E}\left[|\theta_0 - \theta^*|^2\right] + dC_{\mathsf{SGLD},1}/\beta + \lambda C_{\mathsf{SGLD},2} + |\theta^*|^2) + 2d\gamma \\
&\leq \gamma C_{\mathsf{EMose}},
\end{aligned}
$$

where $C_{\mathsf{EMose}} = 8d + 56(e^{-2n\lambda \mathbb{E}[\sigma_\tau^2 m_\tau^2]}\mathbb{E}\left[|\theta_0 - \theta^*|^2\right] + dC_{\mathsf{SGLD},1}/\beta + \lambda C_{\mathsf{SGLD},2} + |\theta^*|^2)$. $\qquad\square$

*Proof of Lemma 19.* Applying Itô's formula to the process 11 with the approximating function $s$ given in 13, we have, for any $t \in [0, T]$,

$$
\begin{aligned}
\mathrm{d}|\widehat{Y}_t^{\mathrm{EM}}|^2 &= 2\langle \widehat{Y}_t^{\mathrm{EM}}, -\widehat{Y}_{\lfloor t/\gamma \rfloor \gamma}^{\mathrm{EM}} + 2m_{T-\lfloor t/\gamma \rfloor \gamma}\hat{\theta}\rangle \mathrm{d}t + 2\langle \widehat{Y}_t^{\mathrm{EM}}, \sqrt{2}\mathrm{d}\overline{B}_t\rangle + 2d\mathrm{d}t \\
&= -2|\widehat{Y}_t^{\mathrm{EM}}|^2\mathrm{d}t + 2\langle \widehat{Y}_t^{\mathrm{EM}}, \widehat{Y}_t^{\mathrm{EM}} - \widehat{Y}_{\lfloor t/\gamma \rfloor \gamma}^{\mathrm{EM}}\rangle \mathrm{d}t + 2\langle \widehat{Y}_t^{\mathrm{EM}}, 2m_{T-\lfloor t/\gamma \rfloor \gamma}\hat{\theta}\rangle \mathrm{d}t \\
&\quad + 2\langle \widehat{Y}_t^{\mathrm{EM}}, \sqrt{2}\mathrm{d}\overline{B}_t\rangle + 2d\mathrm{d}t.
\end{aligned}
$$

Integrating both sides and taking expectation, we have

$$
\begin{aligned}
\mathbb{E}\left[|\widehat{Y}_t^{\mathrm{EM}}|^2\right] &= -2\int_0^t \mathbb{E}\left[|\widehat{Y}_s^{\mathrm{EM}}|^2\right]\mathrm{d}s + 2\int_0^t \mathbb{E}\left[\langle \widehat{Y}_s^{\mathrm{EM}}, \widehat{Y}_s^{\mathrm{EM}} - \widehat{Y}_{\lfloor s/\gamma \rfloor \gamma}^{\mathrm{EM}}\rangle\right]\mathrm{d}s \\
&\quad + 2\int_0^t \mathbb{E}\left[\langle \widehat{Y}_s^{\mathrm{EM}}, 2m_{T-\lfloor s/\gamma \rfloor \gamma}\hat{\theta}\rangle\right]\mathrm{d}s + \mathbb{E}\left[|\widehat{Y}_0^{\mathrm{EM}}|^2\right] + 2dt.
\end{aligned}
$$

Then, differentiating both sides and using Young's inequality, yield

$$
\begin{aligned}
\frac{\mathrm{d}}{\mathrm{d}t}\mathbb{E}\left[|\widehat{Y}_t^{\mathrm{EM}}|^2\right] &= -2\mathbb{E}\left[|\widehat{Y}_t^{\mathrm{EM}}|^2\right] + 2\mathbb{E}\left[\langle \widehat{Y}_t^{\mathrm{EM}}, \widehat{Y}_t^{\mathrm{EM}} - \widehat{Y}_{\lfloor t/\gamma \rfloor \gamma}^{\mathrm{EM}}\rangle\right] \\
&\quad + 2\mathbb{E}\left[\langle \widehat{Y}_t^{\mathrm{EM}}, 2m_{T-\lfloor t/\gamma \rfloor \gamma}\hat{\theta}\rangle\right] + 2d \\
&\leq -\mathbb{E}\left[|\widehat{Y}_t^{\mathrm{EM}}|^2\right] + 2\mathbb{E}\left[|\widehat{Y}_t^{\mathrm{EM}} - \widehat{Y}_{\lfloor t/\gamma \rfloor \gamma}^{\mathrm{EM}}|^2\right] + 8\mathbb{E}[|\hat{\theta}|^2] + 2d.
\end{aligned}
$$

Using Lemma 18, we have

$$
\frac{\mathrm{d}}{\mathrm{d}t}\left(e^t \mathbb{E}\left[|\widehat{Y}_t^{\mathrm{EM}}|^2\right]\right) \leq e^t (2\gamma C_{\mathsf{EMose}} + 8\mathbb{E}[|\hat{\theta}|^2] + 2d).
$$

Integrating both sides, using $\gamma \in (0, 1/2]$ and Corollary 15 yields

$$
\begin{aligned}
\mathbb{E}\left[|\widehat{Y}_t^{\mathrm{EM}}|^2\right] &\leq e^{-t}d + (1 - e^{-t})(2\gamma C_{\mathsf{EMose}} + 8\mathbb{E}[|\hat{\theta}|^2] + 2d) \\
&\leq 18d + 128(e^{-2n\lambda \mathbb{E}[\sigma_\tau^2 m_\tau^2]}\mathbb{E}\left[|\theta_0 - \theta^*|^2\right] + dC_{\mathsf{SGLD},1}/\beta + \lambda C_{\mathsf{SGLD},2} + |\theta^*|^2).
\end{aligned}
$$

$\qquad\square$

# D Proofs of the Results in the General Case

In this section, we provide the proof of Theorem 10. We start by introducing the results which will be used in the proof of Theorem 10.

## D.1 Preliminary Estimates for the General Case

Throughout this section, we fix $\epsilon \in (0, 1)$. The following auxiliary results will be used in the proof of Theorem 10 and their proofs are postponed to Appendix D.3.

We provide an upper bound for the moments of $(\widehat{Y}_t^{\mathrm{EM}})_{t \in [0,T]}$ defined in 11.

**Lemma 20.** *Let Assumptions 1 and 3.b hold. For any $p \in [2, 4]$ and $t \in [0, T - \epsilon]$,*

$$\sup_{0 \le s \le t} \mathbb{E}\left[|\widehat{Y}_s^{EM}|^p\right] \le C_{\mathsf{EM},p}(t),$$

*where*

$$C_{\mathsf{EM},p}(t) := e^{t(3p-1-\frac{2}{p}+2^{2p-1}\mathsf{K}_{Total}^p(1+T^{\alpha p}))}$$

$$\times \left( \mathbb{E}\left[|\widehat{Y}_0^{EM}|^p\right] + 2^{3p-2}\mathsf{K}_{Total}^p t(1 + \mathbb{E}[|\hat{\theta}|^p])(1 + T^{\alpha p}) + \frac{2}{p}(pM + p(p-2))^{\frac{p}{2}} t \right).$$

The following result provides an estimate for the one step error associated with $(\widehat{Y}_t^{\mathrm{EM}})_{t \in [0,T]}$ defined in 11.

**Lemma 21.** *Let Assumptions 1 and 3.b hold. For any $p \in [2, 4]$ and $t \in [0, T - \epsilon]$,*

$$\mathbb{E}\left[|\widehat{Y}_t^{EM} - \widehat{Y}_{\lfloor t/\gamma \rfloor \gamma}^{EM}|^p\right] \le \gamma^{\frac{p}{2}} C_{\mathsf{EMose},p},$$

*where*

$$C_{\mathsf{EMose},p} := 2^{p-1}(C_{\mathsf{EM},p}(T) + \mathsf{K}_{Total}^p(1 + T^{\alpha p})(2^{3p-2}C_{\mathsf{EM},p}(T) + 2^{4p-3}(1 + \mathbb{E}[|\hat{\theta}|^p])))$$

$$+ (Mp(p-1))^{\frac{p}{2}}.$$

The following result is a modification of Kumar & Sabanis (2019, Lemma 4.1).

**Lemma 22.** *Let Assumption 3.b hold and let $b : [0, T] \times \mathbb{R}^d \times \mathbb{R}^M \to \mathbb{R}^M$ such that*

$$b(t, \theta, x) := x + 2s(t, \theta, x). \tag{57}$$

*Then, for any $x, \bar{x} \in \mathbb{R}^M$, $t \in [0, T]$, $\theta \in \mathbb{R}^d$, $\alpha \in [\frac{1}{2}, 1]$, and $k = 1, \dots M$,*

$$\left| b^{(k)}(t, \theta, x) - b^{(k)}(t, \theta, \bar{x}) - \sum_{i=1}^{M} \frac{\partial b^{(k)}(t, \theta, \bar{x})}{\partial y^i}(x^i - \bar{x}^i) \right| \le \mathsf{K}_4(1 + 2|t|^\alpha)|x - \bar{x}|^2.$$

**Lemma 23.** *Let Assumption 3.b hold and let $b$ be as in 57. Then, one obtains that, for any $(t, \theta, x) \in [0, T] \times \mathbb{R}^d \times \mathbb{R}^M$ and $k = 1, \dots M$,*

$$|\nabla_x b^{(k)}(t, \theta, x)| \le 1 + 2\mathsf{K}_3(1 + 2|t|^\alpha). \tag{58}$$

## D.2 Proof of the Main Result for the General Case

*Proof of Theorem 10.* We proceed as in the proof of Theorem 1 using the splitting

$$W_2(\mathcal{L}(Y_K^{\mathrm{EM}}), \pi_{\mathsf{D}}) \le W_2(\pi_{\mathsf{D}}, \mathcal{L}(Y_{t_K})) + W_2(\mathcal{L}(Y_{t_K}), \mathcal{L}(\widetilde{Y}_{t_K}))$$

$$+ W_2(\mathcal{L}(\widetilde{Y}_{t_K}), \mathcal{L}(Y_{t_K}^{\mathrm{aux}})) + W_2(\mathcal{L}(Y_{t_K}^{\mathrm{aux}}), \mathcal{L}(Y_K^{\mathrm{EM}})), \tag{59}$$

where the first term on the right-hand side in 59 corresponds to the error made by the early stopping and the remaining terms have the same interpretation of the corresponding ones in 34.

**Upper bound on $W_2(\pi_D, \mathcal{L}(Y_{t_K}))$.** For any $t \in [0, T]$, note that

$$1 - m_t \leq \sigma_t, \qquad \sigma_t^2 = 1 - e^{-2t} \leq 2t. \tag{60}$$

Recall that $t_K = T - \epsilon$. Using the representation of the OU process 2 and the inequalities 60, we have

$$
\begin{aligned}
W_2(\pi_D, \mathcal{L}(Y_{t_K})) &= W_2(\pi_D, \mathcal{L}(X_{T-t_K})) \\
&\leq \sqrt{\mathbb{E}\left[|X_0 - X_{T-t_K}|^2\right]} \\
&\leq \sqrt{2}\left[(1 - m_{T-t_K})\sqrt{\mathbb{E}[|X_0|^2]} + \sigma_{T-t_K}\sqrt{M}\right] \\
&\leq \sqrt{2}\sigma_{T-t_K}(\sqrt{\mathbb{E}[|X_0|^2]} + \sqrt{M}) \\
&\leq 2\sqrt{\epsilon}(\sqrt{\mathbb{E}[|X_0|^2]} + \sqrt{M}).
\end{aligned}
\tag{61}
$$

**Upper bound on $W_2(\mathcal{L}(Y_{t_K}), \mathcal{L}(\widetilde{Y}_{t_K}))$.** Using Itô's formula, we have, for any $t \in [0, T - \epsilon]$,

$$
\begin{aligned}
\mathrm{d}|Y_t - \widetilde{Y}_t|^2 &= 2\langle Y_t - \widetilde{Y}_t, Y_t + 2\nabla \log p_{T-t}(Y_t) - \widetilde{Y}_t - 2\nabla \log p_{T-t}(\widetilde{Y}_t)\rangle \, \mathrm{d}t \\
&= 2|Y_t - \widetilde{Y}_t|^2 \, \mathrm{d}t + 4\langle Y_t - \widetilde{Y}_t, \nabla \log p_{T-t}(Y_t) - \nabla \log p_{T-t}(\widetilde{Y}_t)\rangle \, \mathrm{d}t.
\end{aligned}
\tag{62}
$$

By integrating, taking expectations on both sides in 62, and using Remark 4 with the lower bound $\widehat{L}_{\mathrm{MO}}$ in the estimate 19, we have

$$
\begin{aligned}
\mathbb{E}[|Y_{t_K} - \widetilde{Y}_{t_K}|^2] &\leq \mathbb{E}[|Y_0 - \widetilde{Y}_0|^2] + \int_0^{t_K} 2(1 - 2\widehat{L}_{\mathrm{MO}}) \, \mathbb{E}\left[|Y_t - \widetilde{Y}_t|^2\right] \mathrm{d}t \\
&\leq \mathbb{E}[|Y_0 - \widetilde{Y}_0|^2]e^{2(1-2\widehat{L}_{\mathrm{MO}})t_K}.
\end{aligned}
\tag{63}
$$

Using 63, the representation 2 with $Z_T \overset{\mathrm{d}}{=} \widetilde{Y}_0$ and 60, we have

$$
\begin{aligned}
\mathbb{E}[|Y_{t_K} - \widetilde{Y}_{t_K}|^2] &\leq \mathbb{E}[|Y_0 - \widetilde{Y}_0|^2]e^{2(1-2\widehat{L}_{\mathrm{MO}})t_K} \\
&= \mathbb{E}[|m_T X_0 + (\sigma_T - 1)\widetilde{Y}_0|^2] \, e^{2(1-2\widehat{L}_{\mathrm{MO}})t_K} \\
&\leq 2\left(\mathbb{E}[|X_0|^2] + M\right)e^{2(1-2\widehat{L}_{\mathrm{MO}})t_K - 2T}.
\end{aligned}
\tag{64}
$$

Using 64, we have

$$
\begin{aligned}
W_2(\mathcal{L}(Y_{t_K}), \mathcal{L}(\widetilde{Y}_{t_K})) &\leq \sqrt{\mathbb{E}[|Y_{t_K} - \widetilde{Y}_{t_K}|^2]} \\
&\leq \sqrt{2}(\sqrt{\mathbb{E}[|X_0|^2]} + \sqrt{M})e^{-2\widehat{L}_{\mathrm{MO}}(T-\epsilon)-\epsilon}.
\end{aligned}
\tag{65}
$$

**Upper bound on $W_2(\mathcal{L}(\widetilde{Y}_{t_K}), \mathcal{L}(Y_{t_K}^{\mathbf{aux}}))$.** Using Itô's formula, we have, for $t \in [0, T - \epsilon]$,

$$
\begin{aligned}
\mathrm{d}|\widetilde{Y}_t - Y_t^{\mathrm{aux}}|^2 &= 2\langle \widetilde{Y}_t - Y_t^{\mathrm{aux}}, \widetilde{Y}_t + 2\,\nabla \log p_{T-t}(\widetilde{Y}_t) - Y_t^{\mathrm{aux}} - 2\,s(T-t, \hat{\theta}, Y_t^{\mathrm{aux}})\rangle \, \mathrm{d}t \\
&= 2|\widetilde{Y}_t - Y_t^{\mathrm{aux}}|^2 \, \mathrm{d}t + 4\,\langle \widetilde{Y}_t - Y_t^{\mathrm{aux}}, \nabla \log p_{T-t}(\widetilde{Y}_t) - \nabla \log p_{T-t}(Y_t^{\mathrm{aux}})\rangle \, \mathrm{d}t \\
&\quad + 4\,\langle \widetilde{Y}_t - Y_t^{\mathrm{aux}}, \nabla \log p_{T-t}(Y_t^{\mathrm{aux}}) - s(T-t, \hat{\theta}, Y_t^{\mathrm{aux}})\rangle \, \mathrm{d}t.
\end{aligned}
\tag{66}
$$

By integrating and taking the expectation on both sides in 66, and using the Remark 4 with the lower bound $\widehat{L}_{\mathrm{MO}}$ in the estimate 19, Young's inequality with $\zeta \in (0, 1)$ and Assumption 4, we have

$$
\begin{aligned}
\mathbb{E}[|\widetilde{Y}_{T-\epsilon} - Y_{T-\epsilon}^{\mathrm{aux}}|^2] &= 2 \int_0^{T-\epsilon} \mathbb{E}[|\widetilde{Y}_s - Y_s^{\mathrm{aux}}|^2] \, \mathrm{d}s \\
&\quad + 4 \int_0^{T-\epsilon} \mathbb{E}[\langle \widetilde{Y}_s - Y_s^{\mathrm{aux}}, \nabla \log p_{T-s}(\widetilde{Y}_s) - \nabla \log p_{T-s}(Y_s^{\mathrm{aux}})\rangle] \, \mathrm{d}s \\
&\quad + 4 \int_0^{T-\epsilon} \mathbb{E}[\langle \widetilde{Y}_s - Y_s^{\mathrm{aux}}, \nabla \log p_{T-s}(Y_s^{\mathrm{aux}}) - s(T - s, \hat{\theta}, Y_s^{\mathrm{aux}})\rangle] \, \mathrm{d}s \\
&\leq \int_0^{T-\epsilon} 2(1 + \zeta - 2\widehat{L}_{\mathrm{MO}}) \mathbb{E}[|\widetilde{Y}_s - Y_s^{\mathrm{aux}}|^2] \, \mathrm{d}s \\
&\quad + 2\zeta^{-1} \int_0^{T-\epsilon} \mathbb{E}[|\nabla \log p_{T-s}(Y_s^{\mathrm{aux}}) - s(T - s, \hat{\theta}, Y_s^{\mathrm{aux}})|^2] \, \mathrm{d}s \\
&\leq \int_0^{T-\epsilon} 2(1 + \zeta - 2\widehat{L}_{\mathrm{MO}}) \, \mathbb{E}[|\widetilde{Y}_s - Y_s^{\mathrm{aux}}|^2] \, \mathrm{d}s + 2\zeta^{-1} \varepsilon_{\mathrm{SN}} \\
&\leq 2e^{2(1+\zeta-2\widehat{L}_{\mathrm{MO}})(T-\epsilon)} \zeta^{-1} \varepsilon_{\mathrm{SN}}.
\end{aligned}
\tag{67}
$$

Using 67 and $t_K = T - \epsilon$, we have

$$
\begin{aligned}
W_2(\mathcal{L}(\widetilde{Y}_{t_K}), \mathcal{L}(Y_{t_K}^{\mathrm{aux}})) &\leq \sqrt{\mathbb{E}[|\widetilde{Y}_{t_K} - Y_{t_K}^{\mathrm{aux}}|^2]} \\
&\leq \sqrt{2\zeta^{-1}} e^{(1+\zeta-2\widehat{L}_{\mathrm{MO}})(T-\epsilon)} \sqrt{\varepsilon_{\mathrm{SN}}}.
\end{aligned}
\tag{68}
$$

**Upper bound on** $W_2(\mathcal{L}(Y_{t_K}^{\mathbf{aux}}), \mathcal{L}(Y_K^{\mathbf{EM}}))$. The following bound is derived by modifying Kumar & Sabanis (2019, Lemma 4.7). For the sake of presentation, let $b : [0, T] \times \mathbb{R}^d \times \mathbb{R}^M \to \mathbb{R}^M$ such that

$$
b(t, \theta, x) = x + 2s(t, \theta, x).
\tag{69}
$$

Consequently, $(\widehat{Y}_t^{\mathrm{EM}})_{t \in [0, T]}$ can be expressed using 69 as

$$
\mathrm{d}\widehat{Y}_t^{\mathrm{EM}} = b(T - \lfloor t/\gamma \rfloor \gamma, \hat{\theta}, \widehat{Y}_{\lfloor t/\gamma \rfloor \gamma}^{\mathrm{EM}}) \, \mathrm{d}t + \sqrt{2} \, \mathrm{d}\bar{B}_t, \qquad \widehat{Y}_0^{\mathrm{EM}} \sim \pi_\infty = \mathcal{N}(0, I_M).
\tag{70}
$$

Using Itô's formula, we have, for $t \in [0, T - \epsilon]$,

$$
\begin{aligned}
&\mathrm{d}|Y_t^{\mathrm{aux}} - \widehat{Y}_t^{\mathrm{EM}}|^2 \\
&= 2\langle Y_t^{\mathrm{aux}} - \widehat{Y}_t^{\mathrm{EM}}, Y_t^{\mathrm{aux}} + 2\,s(T - t, \hat{\theta}, Y_t^{\mathrm{aux}}) - \widehat{Y}_{\lfloor t/\gamma \rfloor \gamma}^{\mathrm{EM}} - 2\,s(T - \lfloor t/\gamma \rfloor \gamma, \hat{\theta}, \widehat{Y}_{\lfloor t/\gamma \rfloor \gamma}^{\mathrm{EM}})\rangle \, \mathrm{d}t \\
&= 2|Y_t^{\mathrm{aux}} - \widehat{Y}_t^{\mathrm{EM}}|^2 \, \mathrm{d}t + 4\langle Y_t^{\mathrm{aux}} - \widehat{Y}_t^{\mathrm{EM}}, s(T - t, \hat{\theta}, Y_t^{\mathrm{aux}}) - s(T - t, \hat{\theta}, \widehat{Y}_t^{\mathrm{EM}})\rangle \, \mathrm{d}t \\
&\quad + 4\langle Y_t^{\mathrm{aux}} - \widehat{Y}_t^{\mathrm{EM}}, s(T - t, \hat{\theta}, \widehat{Y}_{\lfloor t/\gamma \rfloor \gamma}^{\mathrm{EM}}) - s(T - \lfloor t/\gamma \rfloor \gamma, \hat{\theta}, \widehat{Y}_{\lfloor t/\gamma \rfloor \gamma}^{\mathrm{EM}})\rangle \, \mathrm{d}t \\
&\quad + 2\langle Y_t^{\mathrm{aux}} - \widehat{Y}_t^{\mathrm{EM}}, b(T - t, \hat{\theta}, \widehat{Y}_t^{\mathrm{EM}}) - b(T - t, \hat{\theta}, \widehat{Y}_{\lfloor t/\gamma \rfloor \gamma}^{\mathrm{EM}}))\rangle \, \mathrm{d}t.
\end{aligned}
\tag{71}
$$

Integrating and taking the expectation on both sides in 71, using Cauchy–Schwarz inequality, Young's inequality with $\zeta \in (0, 1)$, Assumption 3.b and Remark 3, yield

$$
\begin{aligned}
&\mathbb{E}\left[|Y_t^{\text{aux}} - \widehat{Y}_t^{\text{EM}}|^2\right] \\
&= 2\int_0^t \mathbb{E}\left[|Y_s^{\text{aux}} - \widehat{Y}_s^{\text{EM}}|^2\right] \, \mathrm{d}s + 4\int_0^t \mathbb{E}\left[\langle Y_s^{\text{aux}} - \widehat{Y}_s^{\text{EM}}, s(T-s, \hat{\theta}, Y_s^{\text{aux}}) - s(T-s, \hat{\theta}, \widehat{Y}_s^{\text{EM}})\rangle\right] \, \mathrm{d}s \\
&\quad + 4\int_0^t \mathbb{E}\left[\langle Y_s^{\text{aux}} - \widehat{Y}_s^{\text{EM}}, s(T-s, \hat{\theta}, \widehat{Y}_{\lfloor s/\gamma\rfloor\gamma}^{\text{EM}}) - s(T - \lfloor s/\gamma\rfloor\gamma, \hat{\theta}, \widehat{Y}_{\lfloor s/\gamma\rfloor\gamma}^{\text{EM}})\rangle\right] \, \mathrm{d}s \\
&\quad + 2\int_0^t \mathbb{E}\left[\langle Y_s^{\text{aux}} - \widehat{Y}_s^{\text{EM}}, b(T-s, \hat{\theta}, \widehat{Y}_s^{\text{EM}}) - b(T-s, \hat{\theta}, \widehat{Y}_{\lfloor s/\gamma\rfloor\gamma}^{\text{EM}})\rangle\right] \, \mathrm{d}s \\
&\leq \int_0^t 2(1 + \zeta + 2\mathsf{K}_3(1 + 2|T-s|^\alpha)) \, \mathbb{E}\left[|Y_s^{\text{aux}} - \widehat{Y}_s^{\text{EM}}|^2\right] \, \mathrm{d}s \\
&\quad + 2\zeta^{-1}\int_0^t \mathbb{E}\left[|s(T-s, \hat{\theta}, \widehat{Y}_{\lfloor s/\gamma\rfloor\gamma}^{\text{EM}}) - s(T - \lfloor s/\gamma\rfloor\gamma, \hat{\theta}, \widehat{Y}_{\lfloor s/\gamma\rfloor\gamma}^{\text{EM}})|^2\right] \, \mathrm{d}s \\
&\quad + 2\int_0^t \mathbb{E}\left[\langle Y_s^{\text{aux}} - \widehat{Y}_s^{\text{EM}}, b(T-s, \hat{\theta}, \widehat{Y}_s^{\text{EM}}) - b(T-s, \hat{\theta}, \widehat{Y}_{\lfloor s/\gamma\rfloor\gamma}^{\text{EM}}))\rangle\right] \, \mathrm{d}s \\
&\leq \int_0^t 2(1 + \zeta + 2\mathsf{K}_3(1 + 2T^\alpha)) \, \mathbb{E}\left[|Y_s^{\text{aux}} - \widehat{Y}_s^{\text{EM}}|^2\right] \, \mathrm{d}s \\
&\quad + 2\zeta^{-1}\mathsf{K}_1^2 \, \mathbb{E}\left[(1 + 2|\hat{\theta}|)^2\right] \int_0^t |s - \lfloor s/\gamma\rfloor\gamma|^{2\alpha} \, \mathrm{d}s \\
&\quad + 2\int_0^t \mathbb{E}\left[\langle Y_s^{\text{aux}} - \widehat{Y}_s^{\text{EM}}, b(T-s, \hat{\theta}, \widehat{Y}_s^{\text{EM}}) - b(T-s, \hat{\theta}, \widehat{Y}_{\lfloor s/\gamma\rfloor\gamma}^{\text{EM}}))\rangle\right] \, \mathrm{d}s \\
&\leq \int_0^t 2(1 + \zeta + 2\mathsf{K}_3(1 + 2T^\alpha)) \, \mathbb{E}\left[|Y_s^{\text{aux}} - \widehat{Y}_s^{\text{EM}}|^2\right] \, \mathrm{d}s + 4\zeta^{-1}\mathsf{K}_1^2(1 + 8(\widetilde{\varepsilon}_{\text{AL}} + |\theta^*|^2))\gamma^{2\alpha}t \\
&\quad + 2\int_0^t \mathbb{E}\left[\langle Y_s^{\text{aux}} - \widehat{Y}_s^{\text{EM}}, b(T-s, \hat{\theta}, \widehat{Y}_s^{\text{EM}}) - b(T-s, \hat{\theta}, \widehat{Y}_{\lfloor s/\gamma\rfloor\gamma}^{\text{EM}})\rangle\right] \, \mathrm{d}s.
\end{aligned}
\tag{72}
$$

We proceed estimating the third term on the right-hand side of 72. Using 70, Young's inequality with $\zeta \in (0, 1)$, Lemma 22 and Lemma 21, yields, for any $t \in [0, T - \epsilon]$

$$
\begin{aligned}
&\int_0^t \mathbb{E}\left[\langle Y_s^{\text{aux}} - \widehat{Y}_s^{\text{EM}}, b(T-s, \hat{\theta}, \widehat{Y}_s^{\text{EM}}) - b(T-s, \hat{\theta}, \widehat{Y}_{\lfloor s/\gamma\rfloor\gamma}^{\text{EM}})\rangle\right] \mathrm{d}s \\
&= \sum_{k=1}^M \mathbb{E}\Bigg[\int_0^t \left(Y_s^{\text{aux},(k)} - \widehat{Y}_s^{\text{EM},(k)}\right)\Bigg(b^{(k)}(T-s, \hat{\theta}, \widehat{Y}_s^{\text{EM}}) - b^{(k)}(T-s, \hat{\theta}, \widehat{Y}_{\lfloor s/\gamma\rfloor\gamma}^{\text{EM}}) \\
&\hspace{6cm} - \sum_{j=1}^M \frac{\partial b^{(k)}(T-s, \hat{\theta}, \widehat{Y}_{\lfloor s/\gamma\rfloor\gamma}^{\text{EM}})}{\partial x^j}(\widehat{Y}_s^{\text{EM},(j)} - \widehat{Y}_{\lfloor s/\gamma\rfloor\gamma}^{\text{EM},(j)})\Bigg) \mathrm{d}s\Bigg] \\
&\quad + \sum_{k=1}^M \mathbb{E}\Bigg[\int_0^t \left(Y_s^{\text{aux},(k)} - \widehat{Y}_s^{\text{EM},(k)}\right)\Bigg(\sum_{j=1}^M \frac{\partial b^{(k)}(T-s, \hat{\theta}, \widehat{Y}_{\lfloor s/\gamma\rfloor\gamma}^{\text{EM}})}{\partial x^j}(\widehat{Y}_s^{\text{EM},(j)} - \widehat{Y}_{\lfloor s/\gamma\rfloor\gamma}^{\text{EM},(j)})\Bigg) \mathrm{d}s\Bigg] \\
&\leq \frac{\zeta}{2}\int_0^t \mathbb{E}\left[|Y_s^{\text{aux}} - \widehat{Y}_s^{\text{EM}}|^2\right] \, \mathrm{d}s \\
&\quad + \frac{1}{2\zeta}\sum_{k=1}^M \mathbb{E}\Bigg[\int_0^t |b^{(k)}(T-s, \hat{\theta}, \widehat{Y}_s^{\text{EM}}) - b^{(k)}(T-s, \hat{\theta}, \widehat{Y}_{\lfloor s/\gamma\rfloor\gamma}^{\text{EM}})
\end{aligned}
$$

$$
\quad - \sum_{j=1}^{M} \frac{\partial b^{(k)}(T-s,\hat{\theta},\widehat{Y}_{\lfloor s/\gamma\rfloor\gamma}^{\mathrm{EM}})}{\partial x^j} \left( \widehat{Y}_s^{\mathrm{EM},(j)} - \widehat{Y}_{\lfloor s/\gamma\rfloor\gamma}^{\mathrm{EM},(j)} \right) |^2 \,\mathrm{d}s \Bigg]
$$

$$
+ \sum_{k=1}^{M} \mathbb{E}\Bigg[ \int_0^t \left( Y_s^{\mathrm{aux},(k)} - \widehat{Y}_s^{\mathrm{EM},(k)} \right)
$$

$$
\times \left( \sum_{j=1}^{M} \frac{\partial b^{(k)}(T-s,\hat{\theta},\widehat{Y}_{\lfloor s/\gamma\rfloor\gamma}^{\mathrm{EM}})}{\partial x^j} \int_{\lfloor s/\gamma\rfloor\gamma}^{s} b^{(j)}(T-\lfloor r/\gamma\rfloor\gamma,\hat{\theta},\widehat{Y}_{\lfloor r/\gamma\rfloor\gamma}^{\mathrm{EM}}) \,\mathrm{d}r \right) \mathrm{d}s \Bigg]
$$

$$
+ \sum_{k=1}^{M} \mathbb{E}\left[ \int_0^t \left( Y_s^{\mathrm{aux},(k)} - \widehat{Y}_s^{\mathrm{EM},(k)} \right) \left( \sum_{j=1}^{M} \frac{\partial b^{(k)}(T-s,\hat{\theta},\widehat{Y}_{\lfloor s/\gamma\rfloor\gamma}^{\mathrm{EM}})}{\partial x^j} \int_{\lfloor s/\gamma\rfloor\gamma}^{s} \sqrt{2}\,\mathrm{d}\bar{B}_r^{(j)} \right) \mathrm{d}s \right]
$$

$$
\leq \frac{\zeta}{2} \int_0^t \mathbb{E}\left[ |Y_s^{\mathrm{aux}} - \widehat{Y}_s^{\mathrm{EM}}|^2 \right] \,\mathrm{d}s + \frac{\mathsf{K}_4^2}{2\zeta} \int_0^t (1+2|T-s|^\alpha)^2 \,\mathbb{E}\left[ |\widehat{Y}_s^{\mathrm{EM}} - \widehat{Y}_{\lfloor s/\gamma\rfloor\gamma}^{\mathrm{EM}}|^4 \right] \,\mathrm{d}s
$$

$$
+ \sum_{k=1}^{M} \mathbb{E}\Bigg[ \int_0^t \left( Y_s^{\mathrm{aux},(k)} - \widehat{Y}_s^{\mathrm{EM},(k)} \right)
$$

$$
\times \left( \sum_{j=1}^{M} \frac{\partial b^{(k)}(T-s,\hat{\theta},\widehat{Y}_{\lfloor s/\gamma\rfloor\gamma}^{\mathrm{EM}})}{\partial x^j} \int_{\lfloor s/\gamma\rfloor\gamma}^{s} b^{(j)}(T-\lfloor r/\gamma\rfloor\gamma,\hat{\theta},\widehat{Y}_{\lfloor r/\gamma\rfloor\gamma}^{\mathrm{EM}}) \,\mathrm{d}r \right) \mathrm{d}s \Bigg] \tag{73}
$$

$$
+ \sum_{k=1}^{M} \mathbb{E}\left[ \int_0^t \left( Y_s^{\mathrm{aux},(k)} - \widehat{Y}_s^{\mathrm{EM},(k)} \right) \left( \sum_{j=1}^{M} \frac{\partial b^{(k)}(T-s,\hat{\theta},\widehat{Y}_{\lfloor s/\gamma\rfloor\gamma}^{\mathrm{EM}})}{\partial x^j} \int_{\lfloor s/\gamma\rfloor\gamma}^{s} \sqrt{2}\,\mathrm{d}\bar{B}_r^{(j)} \right) \mathrm{d}s \right]
$$

$$
\leq \frac{\zeta}{2} \int_0^t \mathbb{E}\left[ |Y_s^{\mathrm{aux}} - \widehat{Y}_s^{\mathrm{EM}}|^2 \right] \,\mathrm{d}s + \gamma^2 \frac{\mathsf{K}_4^2}{\zeta} t(1+4T^{2\alpha}) C_{\mathsf{EMose},4}
$$

$$
+ \sum_{k=1}^{M} \mathbb{E}\Bigg[ \int_0^t \left( Y_s^{\mathrm{aux},(k)} - \widehat{Y}_s^{\mathrm{EM},(k)} \right)
$$

$$
\times \left( \sum_{j=1}^{M} \frac{\partial b^{(k)}(T-s,\hat{\theta},\widehat{Y}_{\lfloor s/\gamma\rfloor\gamma}^{\mathrm{EM}})}{\partial x^j} \int_{\lfloor s/\gamma\rfloor\gamma}^{s} b^{(j)}(T-\lfloor r/\gamma\rfloor\gamma,\hat{\theta},\widehat{Y}_{\lfloor r/\gamma\rfloor\gamma}^{\mathrm{EM}}) \,\mathrm{d}r \right) \mathrm{d}s \Bigg]
$$

$$
+ \sum_{k=1}^{M} \mathbb{E}\left[ \int_0^t \left( Y_s^{\mathrm{aux},(k)} - \widehat{Y}_s^{\mathrm{EM},(k)} \right) \left( \sum_{j=1}^{M} \frac{\partial b^{(k)}(T-s,\hat{\theta},\widehat{Y}_{\lfloor s/\gamma\rfloor\gamma}^{\mathrm{EM}})}{\partial x^j} \int_{\lfloor s/\gamma\rfloor\gamma}^{s} \sqrt{2}\,\mathrm{d}\bar{B}_r^{(j)} \right) \mathrm{d}s \right].
$$

We proceed estimating the third and fourth term on the right-hand side of 73, separately.

The third term is estimated using Young's inequality with $\zeta \in (0,1)$, Lemma 23, Remark 5, Remark 3 and Lemma 20, and for any $t \in [0, T-\epsilon]$, we obtain that

$$
\sum_{k=1}^{M} \mathbb{E}\Bigg[ \int_0^t \left( Y_s^{\mathrm{aux},(k)} - \widehat{Y}_s^{\mathrm{EM},(k)} \right)
$$

$$
\times \left( \sum_{j=1}^{M} \frac{\partial b^{(k)}(T-s,\hat{\theta},\widehat{Y}_{\lfloor s/\gamma\rfloor\gamma}^{\mathrm{EM}})}{\partial x^j} \int_{\lfloor s/\gamma\rfloor\gamma}^{s} b^{(j)}(T-\lfloor r/\gamma\rfloor\gamma,\hat{\theta},\widehat{Y}_{\lfloor r/\gamma\rfloor\gamma}^{\mathrm{EM}})\mathrm{d}r \right) \mathrm{d}s \Bigg]
$$

$$
\leq \frac{\zeta}{2} \int_0^t \mathbb{E}\left[ |Y_s^{\mathrm{aux}} - \widehat{Y}_s^{\mathrm{EM}}|^2 \right] \,\mathrm{d}s
$$

$$
+ \frac{2\gamma}{\zeta} M(1+8\mathsf{K}_3^2(1+4T^{2\alpha})) \,\mathbb{E}\left[ \int_0^t \int_{\lfloor s/\gamma\rfloor\gamma}^{s} \sum_{j=1}^{M} |b^{(j)}(T-\lfloor r/\gamma\rfloor\gamma,\hat{\theta},\widehat{Y}_{\lfloor r/\gamma\rfloor\gamma}^{\mathrm{EM}})|^2\mathrm{d}r \,\mathrm{d}s \right]
$$

$$
\begin{aligned}
&\leq \frac{\zeta}{2}\int_0^t \mathbb{E}\left[|Y_s^{\mathrm{aux}} - \widehat{Y}_s^{\mathrm{EM}}|^2\right]\,\mathrm{d}s \\
&\quad + \frac{4\gamma}{\zeta}M(1 + 8\mathsf{K}_3^2(1 + 4T^{2\alpha})) \\
&\qquad\qquad\qquad \times \int_0^t\int_{\lfloor s/\gamma\rfloor\gamma}^s \left[(1 + 16\mathsf{K}_{\mathrm{Total}}^2(1 + T^{2\alpha}))\mathbb{E}[|\widehat{Y}_{\lfloor r/\gamma\rfloor\gamma}^{\mathrm{EM}}|^2] + 32\mathsf{K}_{\mathrm{Total}}^2(1 + T^{2\alpha})(1 + \mathbb{E}[|\hat{\theta}|^2])\right]\,\mathrm{d}r\,\mathrm{d}s \\
&\leq \frac{\zeta}{2}\int_0^t \mathbb{E}\left[|Y_s^{\mathrm{aux}} - \widehat{Y}_s^{\mathrm{EM}}|^2\right]\,\mathrm{d}s \\
&\quad + \frac{\gamma^2}{\zeta}4M(1 + 8\mathsf{K}_3^2(1 + 4T^{2\alpha})) \\
&\qquad\qquad \times t\left[(1 + 16\mathsf{K}_{\mathrm{Total}}^2(1 + T^{2\alpha}))\sup_{0\leq s\leq t}\mathbb{E}[|\widehat{Y}_s^{\mathrm{EM}}|^2] + 32\mathsf{K}_{\mathrm{Total}}^2(1 + T^{2\alpha})(1 + 2\widetilde{\varepsilon}_{\mathrm{AL}} + 2|\theta^*|^2)\right] \\
&\leq \frac{\zeta}{2}\int_0^t \mathbb{E}\left[|Y_s^{\mathrm{aux}} - \widehat{Y}_s^{\mathrm{EM}}|^2\right]\,\mathrm{d}s \\
&\quad + t\frac{\gamma^2}{\zeta}4M(1 + 8\mathsf{K}_3^2(1 + 4T^{2\alpha})) \\
&\qquad\qquad \times \left[(1 + 16\mathsf{K}_{\mathrm{Total}}^2(1 + T^{2\alpha}))C_{\mathrm{EM},2}(T) + 32\mathsf{K}_{\mathrm{Total}}^2(1 + T^{2\alpha})(1 + 2\widetilde{\varepsilon}_{\mathrm{AL}} + 2|\theta^*|^2)\right].
\end{aligned}
\tag{74}
$$

We proceed with the estimate of the fourth term on the right-hand side of 73. For $k = 1, \dots, M$, we note

$$
\begin{aligned}
Y_s^{\mathrm{aux},(k)} - \widehat{Y}_s^{\mathrm{EM},(k)} = {}& Y_{\lfloor s/\gamma\rfloor\gamma}^{\mathrm{aux},(k)} - \widehat{Y}_{\lfloor s/\gamma\rfloor\gamma}^{\mathrm{EM},(k)} \\
&+ \int_{\lfloor s/\gamma\rfloor\gamma}^s (b^{(k)}(T - r, \hat{\theta}, Y_r^{\mathrm{aux}}) - b^{(k)}(T - r, \hat{\theta}, \widehat{Y}_r^{\mathrm{EM}})\,\mathrm{d}r \\
&+ \int_{\lfloor s/\gamma\rfloor\gamma}^s (b^{(k)}(T - r, \hat{\theta}, \widehat{Y}_r^{\mathrm{EM}}) - b^{(k)}(T - r, \hat{\theta}, \widehat{Y}_{\lfloor r/\gamma\rfloor\gamma}^{\mathrm{EM}}))\,\mathrm{d}r \\
&+ \int_{\lfloor s/\gamma\rfloor\gamma}^s 2(s^{(k)}(T - r, \hat{\theta}, \widehat{Y}_{\lfloor r/\gamma\rfloor\gamma}^{\mathrm{EM}}) - s^{(k)}(T - \lfloor r/\gamma\rfloor\gamma, \hat{\theta}, \widehat{Y}_{\lfloor r/\gamma\rfloor\gamma}^{\mathrm{EM}}))\,\mathrm{d}r.
\end{aligned}
\tag{75}
$$

By using 75, Cauchy–Schwarz inequality, Young's inequality, Assumption 3.b and Lemma 23, we have

$$
\begin{aligned}
&\sum_{k=1}^M \mathbb{E}\left[\int_0^t \left(Y_s^{\mathrm{aux},(k)} - \widehat{Y}_s^{\mathrm{EM},(k)}\right)\left(\sum_{j=1}^M \frac{\partial b^{(k)}(T - s, \hat{\theta}, \widehat{Y}_{\lfloor s/\gamma\rfloor\gamma}^{\mathrm{EM}})}{\partial x^j}\int_{\lfloor s/\gamma\rfloor\gamma}^s \sqrt{2}\,\mathrm{d}\bar{B}_r^{(j)}\right)\,\mathrm{d}s\right] \\
&= \sum_{k=1}^M \mathbb{E}\left[\int_0^t \left(Y_{\lfloor s/\gamma\rfloor\gamma}^{\mathrm{aux},(k)} - \widehat{Y}_{\lfloor s/\gamma\rfloor\gamma}^{\mathrm{EM},(k)}\right)\left(\sum_{j=1}^M \frac{\partial b^{(k)}(T - s, \hat{\theta}, \widehat{Y}_{\lfloor s/\gamma\rfloor\gamma}^{\mathrm{EM}})}{\partial x^j}\int_{\lfloor s/\gamma\rfloor\gamma}^s \sqrt{2}\,\mathrm{d}\bar{B}_r^{(j)}\right)\,\mathrm{d}s\right] \\
&\quad + \sum_{k=1}^M \mathbb{E}\left[\int_0^t \left(\int_{\lfloor s/\gamma\rfloor\gamma}^s (b^{(k)}(T - r, \hat{\theta}, Y_r^{\mathrm{aux}}) - b^{(k)}(T - r, \hat{\theta}, \widehat{Y}_r^{\mathrm{EM}}))\,\mathrm{d}r\right)\right. \\
&\qquad\qquad \left. \times \left(\sum_{j=1}^M \frac{\partial b^{(k)}(T - s, \hat{\theta}, \widehat{Y}_{\lfloor s/\gamma\rfloor\gamma}^{\mathrm{EM}})}{\partial x^j}\int_{\lfloor s/\gamma\rfloor\gamma}^s \sqrt{2}\,\mathrm{d}\bar{B}_r^{(j)}\right)\,\mathrm{d}s\right]
\end{aligned}
$$

$$
\begin{aligned}
&+ \sum_{k=1}^{M} \mathbb{E}\Bigg[ \int_0^t \Bigg( \int_{\lfloor s/\gamma \rfloor \gamma}^{s} (b^{(k)}(T-r,\hat{\theta},\widehat{Y}_r^{\mathrm{EM}}) - b^{(k)}(T-r,\hat{\theta},\widehat{Y}_{\lfloor r/\gamma \rfloor \gamma}^{\mathrm{EM}}))\, \mathrm{d}r \Bigg) \\
&\qquad\qquad \times \Bigg( \sum_{j=1}^{M} \frac{\partial b^{(k)}(T-s,\hat{\theta},\widehat{Y}_{\lfloor s/\gamma \rfloor \gamma}^{\mathrm{EM}})}{\partial x^j} \int_{\lfloor s/\gamma \rfloor \gamma}^{s} \sqrt{2}\, \mathrm{d}\bar{B}_r^{(j)} \Bigg)\, \mathrm{d}s \Bigg] \\
&+ \sum_{k=1}^{M} \mathbb{E}\Bigg[ \int_0^t \Bigg( \int_{\lfloor s/\gamma \rfloor \gamma}^{s} 2(s^{(k)}(T-r,\hat{\theta},\widehat{Y}_{\lfloor r/\gamma \rfloor \gamma}^{\mathrm{EM}}) - s^{(k)}(T-\lfloor r/\gamma \rfloor \gamma,\hat{\theta},\widehat{Y}_{\lfloor r/\gamma \rfloor \gamma}^{\mathrm{EM}}))\, \mathrm{d}r \Bigg) \\
&\qquad\qquad \times \Bigg( \sum_{j=1}^{M} \frac{\partial b^{(k)}(T-s,\hat{\theta},\widehat{Y}_{\lfloor s/\gamma \rfloor \gamma}^{\mathrm{EM}})}{\partial x^j} \int_{\lfloor s/\gamma \rfloor \gamma}^{s} \sqrt{2}\, \mathrm{d}\bar{B}_r^{(j)} \Bigg)\, \mathrm{d}s \Bigg] \\
&\leq \sum_{k=1}^{M} \mathbb{E}\Bigg[ \int_0^t \int_{\lfloor s/\gamma \rfloor \gamma}^{s} \gamma^{-1/2} |b^{(k)}(T-r,\hat{\theta},Y_r^{\mathrm{aux}}) - b^{(k)}(T-r,\hat{\theta},\widehat{Y}_r^{\mathrm{EM}})| \mathrm{d}r \\
&\qquad\qquad \times \gamma^{1/2} \left| \int_{\lfloor s/\gamma \rfloor \gamma}^{s} \sum_{j=1}^{M} \frac{\partial b^{(k)}(T-s,\hat{\theta},\widehat{Y}_{\lfloor s/\gamma \rfloor \gamma}^{\mathrm{EM}})}{\partial x^j} \sqrt{2}\, \mathrm{d}\bar{B}_r^{(j)} \right| \mathrm{d}s \Bigg] \\
&+ \sum_{k=1}^{M} \mathbb{E}\Bigg[ \int_0^t \int_{\lfloor s/\gamma \rfloor \gamma}^{s} |b^{(k)}(T-r,\hat{\theta},\widehat{Y}_r^{\mathrm{EM}}) - b^{(k)}(T-r,\hat{\theta},\widehat{Y}_{\lfloor r/\gamma \rfloor \gamma}^{\mathrm{EM}})| \mathrm{d}r \\
&\qquad\qquad \times \left| \int_{\lfloor s/\gamma \rfloor \gamma}^{s} \sum_{j=1}^{M} \frac{\partial b^{(k)}(T-s,\hat{\theta},\widehat{Y}_{\lfloor s/\gamma \rfloor \gamma}^{\mathrm{EM}})}{\partial x^j} \sqrt{2}\, \mathrm{d}\bar{B}_r^{(j)} \right| \mathrm{d}s \Bigg] \\
&+ \sum_{k=1}^{M} \mathbb{E}\Bigg[ \int_0^t \int_{\lfloor s/\gamma \rfloor \gamma}^{s} 2|s^{(k)}(T-r,\hat{\theta},\widehat{Y}_{\lfloor r/\gamma \rfloor \gamma}^{\mathrm{EM}}) - s^{(k)}(T-\lfloor r/\gamma \rfloor \gamma,\hat{\theta},\widehat{Y}_{\lfloor r/\gamma \rfloor \gamma}^{\mathrm{EM}})| \mathrm{d}r \\
&\qquad\qquad \times \left| \int_{\lfloor s/\gamma \rfloor \gamma}^{s} \sum_{j=1}^{M} \frac{\partial b^{(k)}(T-s,\hat{\theta},\widehat{Y}_{\lfloor s/\gamma \rfloor \gamma}^{\mathrm{EM}})}{\partial x^j} \sqrt{2}\, \mathrm{d}\bar{B}_r^{(j)} \right| \mathrm{d}s \Bigg] \\
&\leq \frac{\gamma^{-1}}{2} \sum_{k=1}^{M} \mathbb{E}\Bigg[ \int_0^t \int_{\lfloor s/\gamma \rfloor \gamma}^{s} |b^{(k)}(T-r,\hat{\theta},Y_r^{\mathrm{aux}}) - b^{(k)}(T-r,\hat{\theta},\widehat{Y}_r^{\mathrm{EM}})|^2 \mathrm{d}r\, \mathrm{d}s \Bigg] \\
&+ \frac{\gamma}{2} \sum_{k=1}^{M} \int_0^t \mathbb{E}\Bigg[ \left| \int_{\lfloor s/\gamma \rfloor \gamma}^{s} \sum_{j=1}^{M} \frac{\partial b^{(k)}(T-s,\hat{\theta},\widehat{Y}_{\lfloor s/\gamma \rfloor \gamma}^{\mathrm{EM}})}{\partial x^j} \sqrt{2}\, \mathrm{d}\bar{B}_r^{(j)} \right|^2 \mathrm{d}s \Bigg] \\
&+ \sum_{k=1}^{M} \int_0^t \left[ \mathbb{E}\Bigg( \int_{\lfloor s/\gamma \rfloor \gamma}^{s} |b^{(k)}(T-r,\hat{\theta},\widehat{Y}_r^{\mathrm{EM}}) - b^{(k)}(T-r,\hat{\theta},\widehat{Y}_{\lfloor r/\gamma \rfloor \gamma}^{\mathrm{EM}})| \mathrm{d}r \Bigg)^2 \right]^{1/2} \\
&\qquad\qquad \times \left[ \mathbb{E}\left| \int_{\lfloor s/\gamma \rfloor \gamma}^{s} \sum_{j=1}^{M} \frac{\partial b^{(k)}(T-s,\hat{\theta},\widehat{Y}_{\lfloor s/\gamma \rfloor \gamma}^{\mathrm{EM}})}{\partial x^j} \sqrt{2}\, \mathrm{d}\bar{B}_r^{(j)} \right|^2 \right]^{1/2} \mathrm{d}s \\
&+ \sum_{k=1}^{M} \int_0^t \left[ \mathbb{E}\Bigg( \int_{\lfloor s/\gamma \rfloor \gamma}^{s} 2|s^{(k)}(T-r,\hat{\theta},\widehat{Y}_{\lfloor r/\gamma \rfloor \gamma}^{\mathrm{EM}}) - s^{(k)}(T-\lfloor r/\gamma \rfloor \gamma,\hat{\theta},\widehat{Y}_{\lfloor r/\gamma \rfloor \gamma}^{\mathrm{EM}})|\, \mathrm{d}r \Bigg)^2 \right]^{1/2} \\
&\qquad\qquad \times \left[ \mathbb{E}\left| \int_{\lfloor s/\gamma \rfloor \gamma}^{s} \sum_{j=1}^{M} \frac{\partial b^{(k)}(T-s,\hat{\theta},\widehat{Y}_{\lfloor s/\gamma \rfloor \gamma}^{\mathrm{EM}})}{\partial x^j} \sqrt{2}\, \mathrm{d}\bar{B}_r^{(j)} \right|^2 \right]^{1/2} \mathrm{d}s
\end{aligned}
$$

$$\leq \mathbb{E}\left[\int_0^t \int_{\lfloor s/\gamma \rfloor \gamma}^s \gamma^{-1}(1 + 8\mathsf{K}_3^2(1 + 4T^{2\alpha}))|Y_r^{\text{aux}} - \widehat{Y}_r^{\text{EM}}|^2 \mathrm{d}r \; \mathrm{d}s\right]$$

$$+ 2\gamma^2 \sum_{k=1}^M \mathbb{E}\left[\int_0^t \sum_{j=1}^M \left|\frac{\partial b^{(k)}(T - s, \hat{\theta}, \widehat{Y}_{\lfloor s/\gamma \rfloor \gamma}^{\text{EM}})}{\partial x^j}\right|^2 \mathrm{d}s\right]$$

$$+ \gamma^{1/2}2(1 + 8\mathsf{K}_3^2(1 + 4T^{2\alpha}))^{1/2}$$

$$\times \sum_{k=1}^M \int_0^t \left[\int_{\lfloor s/\gamma \rfloor \gamma}^s \mathbb{E}[|\widehat{Y}_r^{\text{EM}} - \widehat{Y}_{\lfloor r/\gamma \rfloor \gamma}^{\text{EM}}|^2] \; \mathrm{d}r\right]^{1/2} \left[\mathbb{E}\int_{\lfloor s/\gamma \rfloor \gamma}^s \left|\sum_{j=1}^M \frac{\partial b^{(k)}(T - s, \hat{\theta}, \widehat{Y}_{\lfloor s/\gamma \rfloor \gamma}^{\text{EM}})}{\partial x^j}\right|^2 \mathrm{d}r\right]^{1/2} \mathrm{d}s$$

$$+ \gamma^{1+\alpha}4\mathsf{K}_1(1 + 8\widetilde{\varepsilon}_{\text{AL}} + 8|\theta^*|^2)^{1/2} \sum_{k=1}^M \int_0^t \left[\mathbb{E}\int_{\lfloor s/\gamma \rfloor \gamma}^s \left|\sum_{j=1}^M \frac{\partial b^{(k)}(T - s, \hat{\theta}, \widehat{Y}_{\lfloor s/\gamma \rfloor \gamma}^{\text{EM}})}{\partial x^j}\right|^2 \mathrm{d}r\right]^{1/2} \mathrm{d}s$$

$$\leq (1 + 8\mathsf{K}_3^2(1 + 4T^{2\alpha})) \int_0^t \sup_{0 \leq r \leq s} \mathbb{E}\left[|Y_r^{\text{aux}} - \widehat{Y}_r^{\text{EM}}|^2\right] \mathrm{d}s + 4\gamma^2 tM(1 + 8\mathsf{K}_3^2(1 + 4T^{2\alpha}))$$

$$\hspace{8cm} (76)$$

$$+ \gamma^{3/2}2(1 + 8\mathsf{K}_3^2(1 + 4T^{2\alpha}))^{1/2}C_{\text{EMose},2}^{1/2} \sum_{k=1}^M \int_0^t \left[\mathbb{E}\int_{\lfloor s/\gamma \rfloor \gamma}^s \left|\sum_{j=1}^M \frac{\partial b^{(k)}(T - s, \hat{\theta}, \widehat{Y}_{\lfloor s/\gamma \rfloor \gamma}^{\text{EM}})}{\partial x^j}\right|^2 \mathrm{d}r\right]^{1/2} \mathrm{d}s$$

$$+ \gamma^{1+\alpha}4\mathsf{K}_1(1 + 8\widetilde{\varepsilon}_{\text{AL}} + 8|\theta^*|^2)^{1/2} \sum_{k=1}^M \int_0^t \left[\mathbb{E}\int_{\lfloor s/\gamma \rfloor \gamma}^s \left|\sum_{j=1}^M \frac{\partial b^{(k)}(T - s, \hat{\theta}, \widehat{Y}_{\lfloor s/\gamma \rfloor \gamma}^{\text{EM}})}{\partial x^j}\right|^2 \mathrm{d}r\right]^{1/2} \mathrm{d}s$$

$$\leq (1 + 8\mathsf{K}_3^2(1 + 4T^{2\alpha})) \int_0^t \sup_{0 \leq r \leq s} \mathbb{E}\left[|Y_r^{\text{aux}} - \widehat{Y}_r^{\text{EM}}|^2\right] \; \mathrm{d}s + 4\gamma^2 tM(1 + 8\mathsf{K}_3^2(1 + 4T^{2\alpha}))$$

$$+ [\gamma^2 2(1 + 8\mathsf{K}_3^2(1 + 4T^{2\alpha}))^{1/2}C_{\text{EMose},2}^{1/2} + \gamma^{3/2+\alpha}4\mathsf{K}_1(1 + 8\widetilde{\varepsilon}_{\text{AL}} + 8|\theta^*|^2)^{1/2}]$$
$$\times [tM\sqrt{2}(1 + 8\mathsf{K}_3^2(1 + 4T^{2\alpha}))^{1/2}].$$

Using 73, 74 and 76 in 72, we have

$$\mathbb{E}\left[|Y_t^{\text{aux}} - \widehat{Y}_t^{\text{EM}}|^2\right]$$

$$\leq \int_0^t 4(1 + \zeta + \mathsf{K}_3(1 + 2T^\alpha + 4\mathsf{K}_3(1 + 4T^{2\alpha}))) \sup_{0 \leq r \leq s} \mathbb{E}\left[|Y_r^{\text{aux}} - \widehat{Y}_r^{\text{EM}}|^2\right] \; \mathrm{d}s$$

$$+ 8\gamma^2 t\zeta^{-1}M(1 + 8\mathsf{K}_3^2(1 + 4T^{2\alpha}))$$
$$\times \left[(1 + 16\mathsf{K}_{\text{Total}}^2(1 + T^{2\alpha}))C_{\text{EM},2}(T) + 32\mathsf{K}_{\text{Total}}^2(1 + T^{2\alpha})(1 + 2\widetilde{\varepsilon}_{\text{AL}} + 2|\theta^*|^2)\right]$$
$$+ 2\gamma^2 t\mathsf{K}_4^2 \zeta^{-1}(1 + 4T^{2\alpha})C_{\text{EMose},4} + 8\gamma^2 tM(1 + 8\mathsf{K}_3^2(1 + 4T^{2\alpha}))$$
$$+ \gamma^{2\alpha}t4\zeta^{-1}\mathsf{K}_1^2(1 + 8(\widetilde{\varepsilon}_{\text{AL}} + |\theta^*|^2))$$
$$+ 4[\gamma^2(1 + 8\mathsf{K}_3^2(1 + 4T^{2\alpha}))^{1/2}C_{\text{EMose},2}^{1/2} + \gamma^{3/2+\alpha}2\mathsf{K}_1(1 + 8\widetilde{\varepsilon}_{\text{AL}} + 8|\theta^*|^2)^{1/2}]$$
$$\times [tM\sqrt{2}(1 + 8\mathsf{K}_3^2(1 + 4T^{2\alpha}))^{1/2}].$$

Thus,

$$\sup_{0 \leq s \leq t} \mathbb{E}\left[|Y_s^{\text{aux}} - \widehat{Y}_s^{\text{EM}}|^2\right]$$

$$\leq \int_0^t 4(1 + \zeta + \mathsf{K}_3(1 + 2T^\alpha + 4\mathsf{K}_3(1 + 4T^{2\alpha}))) \sup_{0 \leq r \leq s} \mathbb{E}\left[|Y_r^{\text{aux}} - \widehat{Y}_r^{\text{EM}}|^2\right] \; \mathrm{d}s$$

$$
\begin{aligned}
&+ \gamma^{2\alpha} t 2 \Bigg( \mathsf{K}_4^2 \zeta^{-1} (1 + 4T^{2\alpha}) C_{\mathsf{EMose},4} + 4M(1 + 8\mathsf{K}_3^2(1 + 4T^{2\alpha})) + 2\zeta^{-1}\mathsf{K}_1^2(1 + 8(\widetilde{\varepsilon}_{\mathrm{AL}} + |\theta^*|^2)) \\
&\qquad\quad + 4\zeta^{-1}M(1 + 8\mathsf{K}_3^2(1 + 4T^{2\alpha})) \\
&\qquad\qquad \times [(1 + 16\mathsf{K}_{\mathrm{Total}}^2(1 + T^{2\alpha}))C_{\mathsf{EM},2}(T) + 32\mathsf{K}_{\mathrm{Total}}^2(1 + T^{2\alpha})(1 + 2\widetilde{\varepsilon}_{\mathrm{AL}} + 2|\theta^*|^2)] \\
&\qquad\quad + 2[(1 + 8\mathsf{K}_3^2(1 + 4T^{2\alpha}))^{1/2} C_{\mathsf{EMose},2}^{1/2} + 2\mathsf{K}_1(1 + 8\widetilde{\varepsilon}_{\mathrm{AL}} + 8|\theta^*|^2)^{1/2}] \\
&\qquad\qquad \times [M\sqrt{2}(1 + 8\mathsf{K}_3^2(1 + 4T^{2\alpha}))^{1/2}] \Bigg) \\[6pt]
&\le 2 e^{4(1 + \zeta + \mathsf{K}_3(1 + 2T^\alpha + 4\mathsf{K}_3(1 + 4T^{2\alpha})))t} \gamma^{2\alpha} t \\[4pt]
&\times \Bigg( \mathsf{K}_4^2 \zeta^{-1}(1 + 4T^{2\alpha}) C_{\mathsf{EMose},4} + 4M(1 + 8\mathsf{K}_3^2(1 + 4T^{2\alpha})) + 2\zeta^{-1}\mathsf{K}_1^2(1 + 8(\widetilde{\varepsilon}_{\mathrm{AL}} + |\theta^*|^2)) \\
&\qquad\quad + 4\zeta^{-1}M(1 + 8\mathsf{K}_3^2(1 + 4T^{2\alpha})) \\
&\qquad\qquad \times [(1 + 16\mathsf{K}_{\mathrm{Total}}^2(1 + T^{2\alpha}))C_{\mathsf{EM},2}(T) + 32\mathsf{K}_{\mathrm{Total}}^2(1 + T^{2\alpha})(1 + 2\widetilde{\varepsilon}_{\mathrm{AL}} + 2|\theta^*|^2)] \\
&\qquad\quad + 2[(1 + 8\mathsf{K}_3^2(1 + 4T^{2\alpha}))^{1/2} C_{\mathsf{EMose},2}^{1/2} + 2\mathsf{K}_1(1 + 8\widetilde{\varepsilon}_{\mathrm{AL}} + 8|\theta^*|^2)^{1/2}] \\
&\qquad\qquad \times [M\sqrt{2}(1 + 8\mathsf{K}_3^2(1 + 4T^{2\alpha}))^{1/2}] \Bigg).
\end{aligned}
\tag{77}
$$

Using 77 and $t_K = T - \epsilon$, we have

$$
\begin{aligned}
&W_2(\mathcal{L}(Y_{t_K}^{\mathrm{aux}}), \mathcal{L}(Y_K^{\mathrm{EM}})) \\
&\le \sqrt{\mathbb{E}\left[|Y_{t_K}^{\mathrm{aux}} - \widehat{Y}_{t_K}^{\mathrm{EM}}|^2\right]} \\
&\le \sqrt{2} e^{2(1 + \zeta + \mathsf{K}_3(1 + 2T^\alpha + 4\mathsf{K}_3(1 + 4T^{2\alpha})))(T - \epsilon)} \gamma^\alpha \sqrt{T - \epsilon} \\[4pt]
&\times \Bigg( \mathsf{K}_4^2 \zeta^{-1}(1 + 4T^{2\alpha}) C_{\mathsf{EMose},4} + 4M(1 + 8\mathsf{K}_3^2(1 + 4T^{2\alpha})) + 2\zeta^{-1}\mathsf{K}_1^2(1 + 8(\widetilde{\varepsilon}_{\mathrm{AL}} + |\theta^*|^2)) \\
&\qquad\quad + 4\zeta^{-1}M(1 + 8\mathsf{K}_3^2(1 + 4T^{2\alpha})) \\
&\qquad\qquad \times [(1 + 16\mathsf{K}_{\mathrm{Total}}^2(1 + T^{2\alpha}))C_{\mathsf{EM},2}(T) + 32\mathsf{K}_{\mathrm{Total}}^2(1 + T^{2\alpha})(1 + 2\widetilde{\varepsilon}_{\mathrm{AL}} + 2|\theta^*|^2)] \\
&\qquad\quad + 2[(1 + 8\mathsf{K}_3^2(1 + 4T^{2\alpha}))^{1/2} C_{\mathsf{EMose},2}^{1/2} + 2\mathsf{K}_1(1 + 8\widetilde{\varepsilon}_{\mathrm{AL}} + 8|\theta^*|^2)^{1/2}] \\
&\qquad\qquad \times [M\sqrt{2}(1 + 8\mathsf{K}_3^2(1 + 4T^{2\alpha}))^{1/2}] \Bigg)^{1/2}.
\end{aligned}
\tag{78}
$$

**Final upper bound on** $W_2(\mathcal{L}(Y_K^{\mathbf{EM}}), \pi_{\mathsf{D}})$. Substituting 61, 65, 68, and 78 into 59, we have

$$
\begin{aligned}
&W_2(\mathcal{L}(Y_K^{\mathrm{EM}}), \pi_{\mathsf{D}}) \\
&\le (\sqrt{\mathbb{E}[|X_0|^2]} + \sqrt{M})2\sqrt{\epsilon} \\
&\quad + \sqrt{2}(\sqrt{\mathbb{E}[|X_0|^2]} + \sqrt{M})e^{-2\widehat{L}_{\mathrm{MO}}(T - \epsilon) - \epsilon} \\
&\quad + \sqrt{2\zeta^{-1}} e^{(1 + \zeta - 2\widehat{L}_{\mathrm{MO}})(T - \epsilon)} \sqrt{\varepsilon_{\mathrm{SN}}} \\
&\quad + \sqrt{2} e^{2(1 + \zeta + \mathsf{K}_3(1 + 2T^\alpha + 4\mathsf{K}_3(1 + 4T^{2\alpha})))(T - \epsilon)} \gamma^\alpha \sqrt{T - \epsilon}
\end{aligned}
$$

$$\times \Bigg( \mathsf{K}_4^2 \zeta^{-1}(1 + 4T^{2\alpha})C_{\mathsf{EMose},4} + 4M(1 + 8\mathsf{K}_3^2(1 + 4T^{2\alpha})) + 2\zeta^{-1}\mathsf{K}_1^2(1 + 8(\widetilde{\varepsilon}_{\mathrm{AL}} + |\theta^*|^2))$$

$$+ 4\zeta^{-1}M(1 + 8\mathsf{K}_3^2(1 + 4T^{2\alpha}))$$
$$\times [(1 + 16\mathsf{K}_{\mathrm{Total}}^2(1 + T^{2\alpha}))C_{\mathsf{EM},2}(T) + 32\mathsf{K}_{\mathrm{Total}}^2(1 + T^{2\alpha})(1 + 2\widetilde{\varepsilon}_{\mathrm{AL}} + 2|\theta^*|^2)] \tag{79}$$
$$+ 2[(1 + 8\mathsf{K}_3^2(1 + 4T^{2\alpha}))^{1/2}C_{\mathsf{EMose},2}^{1/2} + 2\mathsf{K}_1(1 + 8\widetilde{\varepsilon}_{\mathrm{AL}} + 8|\theta^*|^2)^{1/2}]$$

$$\times [M\sqrt{2}(1 + 8\mathsf{K}_3^2(1 + 4T^{2\alpha}))^{1/2}] \Bigg)^{1/2}.$$

The bound for $W_2(\mathcal{L}(\widehat{Y}_K^{\mathrm{EM}}), \pi_{\mathsf{D}})$ in 79 can be made arbitrarily small by appropriately choosing parameters including $\epsilon, T, \varepsilon_{\mathrm{SN}}$ and $\gamma$. More precisely, for any $\delta > 0$, we first choose $0 \le \epsilon < \epsilon_\delta$ with $\epsilon_\delta$ given in Table 4 such that the first term on the right-hand side of 79 is

$$(\sqrt{\mathbb{E}[|X_0|^2]} + \sqrt{M})2\sqrt{\epsilon} < \delta/4. \tag{80}$$

Next, we choose $T > T_\delta$ with $T_\delta$ given in Table 4 such that the second term on the right-hand side of 79 is

$$\sqrt{2}(\sqrt{\mathbb{E}[|X_0|^2]} + \sqrt{M})e^{-2\widehat{L}_{\mathrm{MO}}(T-\epsilon)-\epsilon} < \delta/4. \tag{81}$$

Next, we turn to the third term on the right-hand side of 79. We choose $0 < \varepsilon_{\mathrm{SN}} < \varepsilon_{\mathrm{SN},\delta}$ with $\varepsilon_{\mathrm{SN},\delta}$ given in Table 4 such that

$$\sqrt{2\zeta^{-1}}e^{(1+\zeta-2\widehat{L}_{\mathrm{MO}})(T-\epsilon)}\sqrt{\varepsilon_{\mathrm{SN}}} < \delta/4. \tag{82}$$

Finally, we choose $0 < \gamma < \gamma_\delta$ with $\gamma_\delta$ given in Table 4 such that the fourth term on the right-hand side of 79 is

$$\sqrt{2}e^{2(1+\zeta+\mathsf{K}_3(1+2T^\alpha+4\mathsf{K}_3(1+4T^{2\alpha})))(T-\epsilon)}\gamma^\alpha\sqrt{T-\epsilon}$$

$$\times \Bigg( \mathsf{K}_4^2 \zeta^{-1}(1 + 4T^{2\alpha})C_{\mathsf{EMose},4} + 4M(1 + 8\mathsf{K}_3^2(1 + 4T^{2\alpha})) + 2\zeta^{-1}\mathsf{K}_1^2(1 + 8(\widetilde{\varepsilon}_{\mathrm{AL}} + |\theta^*|^2))$$

$$+ 4\zeta^{-1}M(1 + 8\mathsf{K}_3^2(1 + 4T^{2\alpha}))$$
$$\times [(1 + 16\mathsf{K}_{\mathrm{Total}}^2(1 + T^{2\alpha}))C_{\mathsf{EM},2}(T) + 32\mathsf{K}_{\mathrm{Total}}^2(1 + T^{2\alpha})(1 + 2\widetilde{\varepsilon}_{\mathrm{AL}} + 2|\theta^*|^2)]$$
$$+ 2[(1 + 8\mathsf{K}_3^2(1 + 4T^{2\alpha}))^{1/2}C_{\mathsf{EMose},2}^{1/2} + 2\mathsf{K}_1(1 + 8\widetilde{\varepsilon}_{\mathrm{AL}} + 8|\theta^*|^2)^{1/2}] \tag{83}$$

$$\times [M\sqrt{2}(1 + 8\mathsf{K}_3^2(1 + 4T^{2\alpha}))^{1/2}] \Bigg)^{1/2} < \delta/4.$$

Using 80, 81, 82 and 83, we obtain $W_2(\mathcal{L}(\widehat{Y}_K^{\mathrm{EM}}), \pi_{\mathsf{D}}) < \delta$. $\qquad\square$

### D.3 Proof of the Preliminary Results for the General Case

We provide the proofs of Section 3.2 and Appendix D.1.

*Proof of Remark 5.* Using Assumption 3.b, we have

$$|s(t, \theta, x)| \le |s(t, \theta, x) - s(0, 0, 0)| + |s(0, 0, 0)|$$
$$\le \mathsf{K}_1(1 + |\theta|)|t|^\alpha + \mathsf{K}_2(1 + |t|^\alpha)|\theta| + \mathsf{K}_3(1 + |t|^\alpha)|x| + |s(0, 0, 0)|$$
$$\le \mathsf{K}_{\mathrm{Total}}(1 + |t|^\alpha)(1 + |\theta| + |x|),$$

where $\mathsf{K}_{\mathrm{Total}} := \mathsf{K}_1 + \mathsf{K}_2 + \mathsf{K}_3 + |s(0, 0, 0)|$. $\qquad\square$

*Proof of Lemma 20.* Using Itô's formula, we have, for any $t \in [0, T - \epsilon]$ and $p \in [2, 4]$,

$$
\begin{aligned}
\mathrm{d}|\widehat{Y}_t^{\mathrm{EM}}|^p &= p \left\langle |\widehat{Y}_t^{\mathrm{EM}}|^{p-2}\widehat{Y}_t^{\mathrm{EM}}, \widehat{Y}_{\lfloor t/\gamma \rfloor \gamma}^{\mathrm{EM}} \right\rangle \mathrm{d}t + p \left\langle |\widehat{Y}_t^{\mathrm{EM}}|^{p-2}\widehat{Y}_t^{\mathrm{EM}}, 2\ s(T - \lfloor t/\gamma \rfloor \gamma, \hat{\theta}, \widehat{Y}_{\lfloor t/\gamma \rfloor \gamma}^{\mathrm{EM}}) \right\rangle \mathrm{d}t \\
&+ p \left\langle |\widehat{Y}_t^{\mathrm{EM}}|^{p-2}\widehat{Y}_t^{\mathrm{EM}}, \sqrt{2}\ \mathrm{d}\overline{B}_t \right\rangle + \frac{p}{2}|\widehat{Y}_t^{\mathrm{EM}}|^{p-2}(2M)\ \mathrm{d}t + \frac{p(p-2)}{2}|\widehat{Y}_t^{\mathrm{EM}}|^{p-4}\ 2|\widehat{Y}_t^{\mathrm{EM}}|^2 \mathrm{d}t.
\end{aligned}
\tag{84}
$$

Integrating and taking expectation on both sides in 84, using Young's inequality and Remark 5, we have

$$
\begin{aligned}
\mathbb{E}\left[|\widehat{Y}_t^{\mathrm{EM}}|^p\right] &= \mathbb{E}\left[|\widehat{Y}_0^{\mathrm{EM}}|^p\right] + p \int_0^t \mathbb{E}\left[\langle |\widehat{Y}_s^{\mathrm{EM}}|^{p-2}\widehat{Y}_s^{\mathrm{EM}}, \widehat{Y}_{\lfloor s/\gamma \rfloor \gamma}^{\mathrm{EM}} \rangle\right] \mathrm{d}s \\
&\quad + 2p \int_0^t \mathbb{E}\left[\langle |\widehat{Y}_s^{\mathrm{EM}}|^{p-2}\widehat{Y}_s^{\mathrm{EM}}, s(T - \lfloor s/\gamma \rfloor \gamma, \hat{\theta}, \widehat{Y}_{\lfloor s/\gamma \rfloor \gamma}^{\mathrm{EM}}) \rangle\right] \mathrm{d}s \\
&\quad + p(M + p - 2) \int_0^t \mathbb{E}\left[|\widehat{Y}_s^{\mathrm{EM}}|^{p-2}\right] \mathrm{d}s \\
&\leq \mathbb{E}\left[|\widehat{Y}_0^{\mathrm{EM}}|^p\right] + 3(p-1) \int_0^t \mathbb{E}\left[|\widehat{Y}_s^{\mathrm{EM}}|^p\right] \mathrm{d}s + \int_0^t \mathbb{E}\left[|\widehat{Y}_{\lfloor s/\gamma \rfloor \gamma}^{\mathrm{EM}}|^p\right] \mathrm{d}s \\
&\quad + 2 \int_0^t \mathbb{E}\left[|s(T - \lfloor s/\gamma \rfloor \gamma, \hat{\theta}, \widehat{Y}_{\lfloor s/\gamma \rfloor \gamma}^{\mathrm{EM}})|^p\right] \mathrm{d}s \\
&\quad + \frac{2}{p}(pM + p(p-2))^{\frac{p}{2}}t + \frac{p-2}{p} \int_0^t \mathbb{E}\left[|\widehat{Y}_s^{\mathrm{EM}}|^p\right] \mathrm{d}s \\
&\leq \mathbb{E}\left[|\widehat{Y}_0^{\mathrm{EM}}|^p\right] + \left(3p - 2 - \frac{2}{p}\right) \int_0^t \mathbb{E}\left[|\widehat{Y}_s^{\mathrm{EM}}|^p\right] \mathrm{d}s + \int_0^t \mathbb{E}\left[|\widehat{Y}_{\lfloor s/\gamma \rfloor \gamma}^{\mathrm{EM}}|^p\right] \mathrm{d}s \\
&\quad + 2^p \mathsf{K}_{\mathrm{Total}}^p \int_0^t (1 + |T - \lfloor s/\gamma \rfloor \gamma|^\alpha)^p\ \mathbb{E}\left[|\widehat{Y}_{\lfloor s/\gamma \rfloor \gamma}^{\mathrm{EM}}|^p\right] \mathrm{d}s \\
&\quad + 2^{2p-1} \mathsf{K}_{\mathrm{Total}}^p (1 + \mathbb{E}[|\hat{\theta}|^p]) \int_0^t (1 + |T - \lfloor s/\gamma \rfloor \gamma|^\alpha)^p\ \mathrm{d}s + \frac{2}{p}(pM + p(p-2))^{\frac{p}{2}}t \\
&\leq \mathbb{E}\left[|\widehat{Y}_0^{\mathrm{EM}}|^p\right] + \left(3p - 1 - \frac{2}{p} + 2^{2p-1}\mathsf{K}_{\mathrm{Total}}^p(1 + T^{\alpha p})\right) \int_0^t \sup_{0 \leq r \leq s} \mathbb{E}\left[|\widehat{Y}_r^{\mathrm{EM}}|^p\right] \mathrm{d}s \\
&\quad + 2^{3p-2} \mathsf{K}_{\mathrm{Total}}^p(1 + \mathbb{E}[|\hat{\theta}|^p])(1 + T^{\alpha p})t + \frac{2}{p}(pM + p(p-2))^{\frac{p}{2}}t.
\end{aligned}
$$

Using Grönwall's inequality, we have

$$
\begin{aligned}
\sup_{0 \leq s \leq t} \mathbb{E}\left[|\widehat{Y}_s^{\mathrm{EM}}|^p\right] &\leq \mathbb{E}\left[|\widehat{Y}_0^{\mathrm{EM}}|^p\right] + \left(3p - 1 - \frac{2}{p} + 2^{2p-1}\mathsf{K}_{\mathrm{Total}}^p(1 + T^{\alpha p})\right) \int_0^t \sup_{0 \leq r \leq s} \mathbb{E}\left[|\widehat{Y}_r^{\mathrm{EM}}|^p\right] \mathrm{d}s \\
&\quad + 2^{3p-2} \mathsf{K}_{\mathrm{Total}}^p(1 + \mathbb{E}[|\hat{\theta}|^p])(1 + T^{\alpha p})t + \frac{2}{p}(pM + p(p-2))^{\frac{p}{2}}t \\
&\leq e^{t(3p - 1 - \frac{2}{p} + 2^{2p-1}\mathsf{K}_{\mathrm{Total}}^p(1 + T^{\alpha p}))} \\
&\quad \times \left(\mathbb{E}\left[|\widehat{Y}_0^{\mathrm{EM}}|^p\right] + 2^{3p-2}\mathsf{K}_{\mathrm{Total}}^p t(1 + \mathbb{E}[|\hat{\theta}|^p])(1 + T^{\alpha p}) + \frac{2}{p}(pM + p(p-2))^{\frac{p}{2}}t\right).
\end{aligned}
$$

$\square$

*Proof of Lemma 21.* Using 11, Lemma 20 and Remark 5, we have, for any $t \in [0, T - \epsilon]$ and $p \in [2, 4]$,

$$
\begin{aligned}
&\mathbb{E}\left[|\widehat{Y}_t^{\mathrm{EM}} - \widehat{Y}_{\lfloor t/\gamma \rfloor \gamma}^{\mathrm{EM}}|^p\right] \\
&\leq \gamma^p \mathbb{E}\left[\left|\widehat{Y}_{\lfloor t/\gamma \rfloor \gamma}^{\mathrm{EM}} + 2\ s(T - \lfloor t/\gamma \rfloor \gamma, \hat{\theta}, \widehat{Y}_{\lfloor t/\gamma \rfloor \gamma}^{\mathrm{EM}})\right|^p\right] + \mathbb{E}\left[\left|\int_{\lfloor t/\gamma \rfloor \gamma}^t \sqrt{2}\ \mathrm{d}\overline{B}_s\right|^p\right]
\end{aligned}
$$

$$\leq 2^{p-1}\gamma^p \left(\mathbb{E}\left[|\widehat{Y}^{\mathrm{EM}}_{\lfloor t/\gamma\rfloor\gamma}|^p\right] + 2^p\,\mathbb{E}\left[|s(T-\lfloor t/\gamma\rfloor\gamma,\hat{\theta},\widehat{Y}^{\mathrm{EM}}_{\lfloor t/\gamma\rfloor\gamma})|^p\right]\right) + \gamma^{\frac{p}{2}}(Mp(p-1))^{\frac{p}{2}}$$

$$\leq 2^{p-1}\gamma^p(C_{\mathsf{EM},p}(t) + 2^{3p-2}\mathsf{K}^p_{\mathrm{Total}}(1+T^{\alpha p})C_{\mathsf{EM},p}(t) + 2^{4p-3}\mathsf{K}^p_{\mathrm{Total}}(1+T^{\alpha p})(1+\mathbb{E}[|\hat{\theta}|^p]))$$
$$+ \gamma^{\frac{p}{2}}(Mp(p-1))^{\frac{p}{2}}$$

$$\leq \gamma^{\frac{p}{2}}C_{\mathsf{EMose},p},$$

where

$$C_{\mathsf{EMose},p} = 2^{p-1}(C_{\mathsf{EM},p}(T) + \mathsf{K}^p_{\mathrm{Total}}(1+T^{\alpha p})(2^{3p-2}C_{\mathsf{EM},p}(T) + 2^{4p-3}(1+\mathbb{E}[|\hat{\theta}|^p]))) + (Mp(p-1))^{\frac{p}{2}}.$$

$\square$

*Proof of Lemma 22.* By the mean value theorem, for any $k = 1,\ldots,M$, we have,

$$b^{(k)}(t,\theta,x) - b^{(k)}(t,\theta,\bar{x}) = \sum_{i=1}^{M} \frac{\partial b^{(k)}(t,\theta,qx+(1-q)\bar{x})}{\partial y^i}(x^i - \bar{x}^{(i)}),$$

for some $q \in (0,1)$. Hence, for a fixed $q \in (0,1)$, we have

$$\left|b^{(k)}(t,\theta,x) - b^{(k)}(t,\theta,\bar{x}) - \sum_{i=1}^{M} \frac{\partial b^{(k)}(t,\theta,\bar{x})}{\partial y^i}(x^i - \bar{x}^i)\right|$$

$$= \left|\sum_{i=1}^{M} \frac{\partial b^{(k)}(t,\theta,qx+(1-q)\bar{x})}{\partial y^i}(x^i - \bar{x}^{(i)}) - \sum_{i=1}^{M} \frac{\partial b^{(k)}(t,\theta,\bar{x})}{\partial y^i}(x^i - \bar{x}^i)\right|$$

$$\leq \sum_{i=1}^{M} \left|\frac{\partial b^{(k)}(t,\theta,qx+(1-q)\bar{x})}{\partial y^i} - \frac{\partial b^{(k)}(t,\theta,\bar{x})}{\partial y^i}\right||x^i - \bar{x}^i|.$$

The proof is completed using Assumption 3.b. $\square$

*Proof of Lemma 23.* At $x \in \mathbb{R}^M$ and for any $v \in \mathbb{R}^M$, we have, for any $k = 1,\ldots M$,

$$\langle\nabla_x s^{(k)}(t,\theta,x),v\rangle = \lim_{h\to 0} \frac{s^{(k)}(t,\theta,x+vh) - s^{(k)}(t,\theta,x)}{h}.$$

Using Assumption 3.b, we have

$$|\langle\nabla_x s^{(k)}(t,\theta,x),v\rangle| \leq \lim_{h\to 0} \left|\frac{s^{(k)}(t,\theta,x+vh) - s^{(k)}(t,\theta,x)}{h}\right|$$

$$\leq \lim_{h\to 0} \frac{|D_3(t,t)||x+vh-x|}{|h|} \tag{85}$$

$$\leq \mathsf{K}_3(1+2|t|^\alpha)|v|.$$

Taking $v = \frac{\nabla_x s^{(k)}(t,\theta,x)}{|\nabla_x s^{(k)}(t,\theta,x)|}$ in 85, we have

$$|\nabla_x s^{(k)}(t,\theta,x)| \leq \mathsf{K}_3(1+2|t|^\alpha). \tag{86}$$

Using 57 and 86, we obtain

$$|\nabla_x b^{(k)}(t,\theta,x)| \leq 1 + 2|\nabla_x s^{(k)}(t,\theta,x)|$$
$$\leq 1 + 2\mathsf{K}_3(1+2|t|^\alpha).$$

$\square$

# E  Table of Constants

Table 4 displays full expressions for constants which appear in Theorem 10 and Remark 12.

Table 4: Explicit expressions for the constants in Theorem 10 and Remark 12
.

| CONSTANT | DEPENDENCY | FULL EXPRESSION |
|---|---|---|
| $C_1$ | $O(\sqrt{M})$ | $2(\sqrt{\mathbb{E}[|X_0|^2]} + \sqrt{M})$ |
| $C_2$ | $O(\sqrt{M})$ | $\sqrt{2}\left(\sqrt{\mathbb{E}[|X_0|^2]} + \sqrt{M}\right)$ |
| $C_3(T,\epsilon)$ | $O(e^{(1+\zeta-2\widehat{L}_{\mathrm{MO}})(T-\epsilon)})$ | $\sqrt{2\zeta^{-1}}e^{(1+\zeta-2\widehat{L}_{\mathrm{MO}})(T-\epsilon)}$ |
| $C_{\mathsf{EM},2}(T)$ | $O(Me^{T^{2\alpha+1}}T^{2\alpha+1}\widetilde{\varepsilon}_{\mathrm{AL}})$ | $e^{T(4+8\mathsf{K}_{\mathrm{Total}}^2(1+T^{2\alpha}))}$ $\times (\mathbb{E}[|\widehat{Y}_0^{\mathsf{EM}}|^2] + 16\mathsf{K}_{\mathrm{Total}}^2 T(1+2\widetilde{\varepsilon}_{\mathrm{AL}}+2|\theta^*|^2)(1+T^{2\alpha}) + 2MT)$ |
| $C_{\mathsf{EM},4}(T)$ | $O(M^2e^{T^{4\alpha+1}}T^{4\alpha+1})$ | $e^{T(\frac{21}{2}+128\mathsf{K}_{\mathrm{Total}}^4(1+T^{4\alpha}))}$ $\times (\mathbb{E}[|\widehat{Y}_0^{\mathsf{EM}}|^4] + 1024\mathsf{K}_{\mathrm{Total}}^4 T(1+\mathbb{E}[|\hat{\theta}|^4])(1+T^{4\alpha}) + 8(M^2+4M+4)T)$ |
| $C_{\mathsf{EMose},2}$ | $O(Me^{T^{2\alpha+1}}T^{4\alpha+1}\widetilde{\varepsilon}_{\mathrm{AL}})$ | $2(C_{\mathsf{EM},2}(T) + \mathsf{K}_{\mathrm{Total}}^2(1+T^{2\alpha})(16C_{\mathsf{EM},2}(T)+32(1+2\widetilde{\varepsilon}_{\mathrm{AL}}+2|\theta^*|^2))) + 2M$ |
| $C_{\mathsf{EMose},4}$ | $O(M^2e^{T^{4\alpha+1}}T^{8\alpha+1})$ | $8(C_{\mathsf{EM},4}(T) + \mathsf{K}_{\mathrm{Total}}^4(1+T^{4\alpha})(1024C_{\mathsf{EM},4}(T)+8192(1+\mathbb{E}[|\hat{\theta}|^4]))) + 144M^2$ |
| $C_4(T,\epsilon)$ | $O(Me^{T^{4\alpha+1}}T^{4\alpha+1}\widetilde{\varepsilon}_{\mathrm{AL}}^{1/4})$ | $\sqrt{2}e^{2(1+\zeta+\mathsf{K}_3(1+2T^\alpha+4\mathsf{K}_3(1+4T^{2\alpha})))(T-\epsilon)}\sqrt{T-\epsilon}$ $\times \Big(\mathsf{K}_4^2\zeta^{-1}(1+4T^{2\alpha})C_{\mathsf{EMose},4} + 4M(1+8\mathsf{K}_3^2(1+4T^{2\alpha}))$ $+ 2\zeta^{-1}\mathsf{K}_1^2(1+8(\widetilde{\varepsilon}_{\mathrm{AL}}+|\theta^*|^2))$ $+ 4\zeta^{-1}M(1+8\mathsf{K}_3^2(1+4T^{2\alpha}))$ $\times [(1+16\mathsf{K}_{\mathrm{Total}}^2(1+T^{2\alpha}))C_{\mathsf{EM},2}(T)$ $+ 32\mathsf{K}_{\mathrm{Total}}^2(1+T^{2\alpha})(1+2\widetilde{\varepsilon}_{\mathrm{AL}}+2|\theta^*|^2)]$ $+ 2[(1+8\mathsf{K}_3^2(1+4T^{2\alpha}))^{1/2}C_{\mathsf{EMose},2}^{1/2} + 2\mathsf{K}_1(1+8\widetilde{\varepsilon}_{\mathrm{AL}}+8|\theta^*|^2)^{1/2}]$ $\times [M\sqrt{2}(1+8\mathsf{K}_3^2(1+4T^{2\alpha}))^{1/2}]\Big)^{1/2}.$ |
| $\widetilde{C}_4(T,\epsilon)$ | $O(\sqrt{M}e^{T^{2\alpha+1}}T^{2\alpha+1}\widetilde{\varepsilon}_{\mathrm{AL}}^{1/2})$ | $e^{(1+(3/2)\zeta+2\mathsf{K}_3(1+2T^\alpha))(T-\epsilon)}$ $\times (\zeta^{-1/2}(T-\epsilon)^{1/2}C_{\mathsf{EMose},2}^{1/2} + 8^{1/2}\zeta^{-1/2}(T-\epsilon)^{1/2}\mathsf{K}_1(1+8\widetilde{\varepsilon}_{\mathrm{AL}}+8|\theta^*|^2)^{1/2}$ $+ 2\zeta^{-1/2}\mathsf{K}_3(1+2T^\alpha)(T-\epsilon)^{1/2}C_{\mathsf{EMose},2}^{1/2})$ |
| $\epsilon_\delta$ | - | $\delta^2/(64(\sqrt{\mathbb{E}[|X_0|^2]}+\sqrt{M})^2)$ |
| $T_\delta$ | - | $(2\widehat{L}_{\mathrm{MO}})^{-1}\left[\ln\left(4\sqrt{2}\left(\sqrt{\mathbb{E}[|X_0|^2]}+\sqrt{M}\right)/\delta\right) - \epsilon\right] + \epsilon$ |
| $\varepsilon_{\mathrm{SN},\delta}$ | - | $(\zeta\delta^2/32)e^{-2(1+\zeta-2\widehat{L}_{\mathrm{MO}})(T-\epsilon)}$ |
| $\gamma_\delta$ | - | $\min\Big\{(\delta/(4\sqrt{2}))^{1/\alpha}(T-\epsilon)^{-1/(2\alpha)}e^{-(2/\alpha)(1+\zeta+\mathsf{K}_3(1+2T^\alpha+4\mathsf{K}_3(1+4T^{2\alpha})))(T-\epsilon)}$ $\times \Big(\mathsf{K}_4^2\zeta^{-1}(1+4T^{2\alpha})C_{\mathsf{EMose},4} + 4M(1+8\mathsf{K}_3^2(1+4T^{2\alpha}))$ $+ 2\zeta^{-1}\mathsf{K}_1^2(1+8(\widetilde{\varepsilon}_{\mathrm{AL}}+|\theta^*|^2))$ $+ 4\zeta^{-1}M(1+8\mathsf{K}_3^2(1+4T^{2\alpha}))$ $\times [(1+16\mathsf{K}_{\mathrm{Total}}^2(1+T^{2\alpha}))C_{\mathsf{EM},2}(T)$ $+ 32\mathsf{K}_{\mathrm{Total}}^2(1+T^{2\alpha})(1+2\widetilde{\varepsilon}_{\mathrm{AL}}+2|\theta^*|^2)]$ $+ 2[(1+8\mathsf{K}_3^2(1+4T^{2\alpha}))^{1/2}C_{\mathsf{EMose},2}^{1/2} + 2\mathsf{K}_1(1+8\widetilde{\varepsilon}_{\mathrm{AL}}+8|\theta^*|^2)^{1/2}]$ $\times [M\sqrt{2}(1+8\mathsf{K}_3^2(1+4T^{2\alpha}))^{1/2}]\Big)^{-1/(2\alpha)}, 1\Big\}.$ |

