# OpenReview forum: "On diffusion-based generative models and their error bounds: The log-concave case with full convergence estimates"
_TMLR — Accepted by TMLR_

### Review · Reviewer_Kv1D · 2024-10-31

**Summary Of Contributions:**

The authors provide detailed convergence estimates for diffusion models under strongly log-concave data distributions, improving understanding of their theoretical stability and reliability.

Using Lipschitz continuous functions for score estimation, the paper claims to achieve fast convergence rates by controlling the Wasserstein-2 distance between generated samples and the target distribution.

By introducing an $L^2$-accurate score estimation assumption and an auxiliary process, the paper supports a range of stochastic optimizers, broadening the method's applicability in practice​.

**Audience:**

Yes

**Claims And Evidence:**

No

**Requested Changes:**

1. Futher justify the log-concave case. For example, how can you provide a implementable algorithm to be faster than Langevin based ones.

2. Estimation of score function. How to estimate the complexity there?

3. More discussion with the related works.

**Strengths And Weaknesses:**

The paper provides comprehensive study about the properties of diffusion model for log-concave distributions.

However, I am very suspecious of the gain from diffusion model for log-concave case. All the constant numbers are hidden in the estimation of the score function. For previous diffusion proofs (such as Sampling is as easy as learning the score: theory for diffusion models with minimal data assumptions), they claim the complexity of sampling is mainly about the learning process of the score and consider the general case.  To the best of my knowledge, diffusion models works well in highly multi-modal distributions with ill log-Sobolev constant. There are not even sufficient empirical evidence showing the better performance for log-concave case. More evidence and better interpretation for the assumptions are needed.

The discussion with other related work is very limited. For example, the proximal sampler based methods [1-3] and recent diffusion-based sampling works [4-6]. Some of them explicitly consider the complexity within the estimation. Some claims to be the "first" might be reviewed.


[1] Structured logconcave sampling with a restricted gaussian oracle.

[2]  Faster high-accuracy log-concave sampling via algorithmic warm starts.

[3] Improved analysis for a proximal algorithm for sampling.

[4] Reverse Diffusion Monte Carlo

[5] Linear convergence bounds for diffusion models via stochastic localization

[6] Faster Sampling without Isoperimetry via Diffusion-based Monte Carlo

---

> ### Author Response · Authors · 2024-11-28
>
> Thank you for taking the time to review our submission and for your feedback.
>
> We combine our answer to your comments made in the Section "Strengths and weaknesses" and points 1 and 2 together in the first point below.
>
> 1) The focus of our theoretical study is to provide full theoretical guarantees for the convergence of diffusion models. By connecting these models with theoretical guarantees of machine learning optimizers, we have derived, to the best of our knowledge, the first convergence bounds where the parameters involved in the sampling and optimization steps of the diffusion models appear explicitly (see for instance Section 3.1). This connection between the sampling part and the optimization part of the diffusion models was missing in previous works. We consider this to be one of the important contributions of our work. The focus of our work is not to design  new implementable algorithms faster than Langevin based ones but rather to provide the full theoretical justification for the convergence behaviour of diffusion models. In the particular case of the motivating example (Section 3.1), we have chosen a particular optimizer, known as SGLD algorithm, to solve score-matching optimization problem (6) (and therefore "estimate the score function"), and highlight the nature of the explicit constants coming from the sampling and optimization steps appearing in the upper bounds in Theorem 1.  The SGLD algorithm was chosen due to the strongly convex nature of the score-matching optimization problem (6) and its well-known theoretical guarantees in achieving global convergence. Moreover, the SGLD algorithm has allowed us to achieve  optimal dependence of the data dimension ($\sqrt{d}$) in the upper bound in Theorem  1 in the motivating example. All these points are included in the motivating example in Section 3.1. We have provided the bounds in the general case in Section 3.2. These bounds are valid for any algorithms yielding an approximate minimizer $\hat{\theta}$ for the score-matching optimization problem (6) (and therefore "estimate the score function"), as long as  they satisfy the requirements in Assumption 1. The requirements in Assumption 1 are satisfied by popular algorithms used in practice e.g. ADAM, SGD, etc. and not only Langevin based ones. In this case, in order to achieve optimal rate of convergence or order $\alpha \in [1/2,1]$ in Theorem 10, the dependence on the data dimension is  polynomial $O(M)$. All these points are included in Section 3.2. We have also added a new Remark 8 (page 7-8) and in the new Appendix B (page 16), where we provide additional details for the score-matching problem (and therefore "estimate the score function") in the motivating example and in the general case.
>
> Third point:
>
> 2) There is a long section titled "Related Work and Comparison" at pages 9-12 where we have discussed prior works known to us up until the publication of the first draft of our work on Arxiv.  This section contains two tables (pages 11-12) comparing the complexity achieved by the other relevant works. We briefly comment about the six references mentioned by the reviewer. Reference [5] is already discussed in Section 4.2 and the complexity achieved by the authors in [5] is presented in Table 3 and compared with our result. The proximal sampler based methods mentioned in references [1-3] apply the ideas behind "proximal point reduction-based methods" in convex optimization to sampling. In optimization, these methods require to solve several optimization subproblems in order to minimize a convex function. In their application to sampling, these methods  reduce a sampling problem to multiple subproblems, where each subproblem implements a Restricted Gaussian Oracle requiring to sample from a certain distribution. The complexity achieved by the works in [1-3] cannot be (directly) compared with the complexity achieved in our work since our approach does not require calling  a Restricted Gaussian Oracle at each step. For this reason, the references [1-3] are not (immediately) relevant and comparable to our work. The references [4] and [6] diverge from the typical practice of diffusion models used in our work, where the intermediate updates—the score functions—are learned using a neural network. Instead, they reformulate the score matching problem as mean estimation subproblems for posteriors and analyze the complexity of this modification. These references are not directly comparable to our work, as their approach differs from the standard score matching method commonly employed in diffusion models, which we adopt. In addition, we note that Reference [6] appeared after the first version of our work on Arxiv. Our claims to being the "first" in specific aspects are made with careful consideration of the literature available at the time of submitting the first version of our work to Arxiv. We are open to adjusting these claims if you provide such papers that demonstrate precedence predating our work.

---

### Review · Reviewer_Jfhr · 2024-11-04

**Summary Of Contributions:**

Disclaimer: I am not an expert in the convergence theory of diffusion models. Thus, I do not have a good knowledge of the literature on the topic and could not verify the proofs carefully.

* State-of-the-art convergence bounds for diffusion based models in terms of Wasserstein distance between target distribution and generative distribution (Theorem 10). This is achieved thanks to 2 contributions.

* The main technical contribution concerns the use of an auxiliary process (Equation 9) in the study of convergence bounds. This auxiliary process is simply defined as the backward process, where the true score is replaced by its approximation learned by stochastic optimization.

* The authors provide a toy example where they apply their analytical framework to bound the Wasserstein distance between a target multivariate Gaussian with unknown mean, and a distribution generated by a diffusion model. In this case, they consider the score to be learned by SGLD, and use known results of SGLD convergence.

**Audience:**

Yes

**Claims And Evidence:**

Yes

**Requested Changes:**

See weaknesses: more discussion about the impact of the novel auxiliary process, and about the links between Assumption 1 and Assumption 4.

**Strengths And Weaknesses:**

Strengths:

* Improved upper-bounds on the Wasserstein distance between generative and target distributions.

Weaknesses:

* I feel that the justification on the impact of the novel auxiliary process is unclear. The authors mainly claim that the difference is that "the approximating function s and the estimator $\hat{\theta}$ are known". I do believe this falls short in explaining the reason why this process is useful. What is the implication of considering the auxiliary process on the theoretical derivation? How does it lead to improved convergence bounds?

* Also, the link between Assumption 1 and Assumption 4 should be explained with more depth. Since Assumption 4 says that the estimator of the score is L2-accurate, why does one need Assumption 1? Is it because Assumption 4 is only defined on the auxiliary process defined in Equation 9?

* The writing could be made more direct and more concise.

---

> ### Author Response · Authors · 2024-11-28
>
> Thank you for taking the time to review our submission and for your constructive feedback. We answer your points below.
>
> 1) The use of the auxiliary process (9) does not lead to improved convergence bounds, but it just facilitates the convergence proof in the theoretical derivation since it connects the backward process (4) with the numerical scheme (11) (see the upper bounds with the auxiliary process in Appendix C.2 at pages 18-20 and Appendix D.2 at pages 26-32). The importance of the auxiliary process (9) is more on the practical side. This process can be easily simulated because we just need to choose the estimator/algorithm $\hat{\theta}$ satisfying Assumption 1 (e.g. algorithms like SGD, ADAM, SGLD, etc), the approximating function $s$ satisfying Assumption 3 (e.g. Lipschitz functions) and samples from the standard Gaussian distribution to initiate the auxiliary process (9) at time 0 and simulate it. The ease of its practical simulation allow us to use the auxiliary process in the assumption on the score approximation (Assumption 4) in contrast to what was done in previous works. Indeed, it is far more difficult to simulate the time reversal process using the forward process and changing time, due to the equality in distribution of the time reversal and forward process. Therefore, the auxiliary process makes our Assumption 4 on the score matching more "practical" since it does not need samples from data per se as the other assumptions in the works cited in Remark 7 have.
>
>
> 2) We impose Assumption 1 (in addition to Assumption 4) to introduce properties that should be satisfied by the underlying optimization algorithm. This is one of the key assumptions of our results as we investigate the connection between the sampling procedure and the optimization procedure involved in the diffusion models, which has not been addressed in the existing literature.
> To be more precise, in Section 3.1, we provide a concrete example where the expressions of the score function and its approximating function are known. Then, by choosing the SGLD algorithm as the optimizer (to solve (6)), we showed that Assumptions 1-3 are satisfied. In this particular example, the $L^2$ estimate imposed in Assumption 4 is, in fact, a result that can be deduced using arguments in the new Appendix B (at page 16) and updated Remark 8 (at page 7-8). In Theorem 1, we present a non-asymptotic convergence bound for the diffusion model involving parameters from both sampling (Euler) and optimization (SGLD) algorithms.
> In the general setting, we do not restrict ourselves to the choice of the SGLD algorithm in the optimization procedure, and any optimizer (including, e.g., ADAM and SGD) that satisfies conditions in Assumption 1 could, together with Assumptions 2-4, lead to the convergence result in Theorem 10. In addition, as we do not specify the expressions for the score function and its approximating function in the general setting, we need to explicitly assume Assumptions 2-4 so as to obtain the main result.

---

### Review · Reviewer_3tvY · 2024-11-15

**Summary Of Contributions:**

The article under consideration provides with theoretical guarantees of convergence for score-based generative models in W_2 distance assuming, among other things, that the data distribution is log-concave. This problem has been recently  deeply investigated by several other works. The main interest of this work is that guarantess of convergence are in W_2. The article suffers of several limitations as well. However, I believe it is a contribution that would interest the readers of TMLR and I cautiously recommend publication provided the authors address the comments below in full detail.

**Audience:**

Yes

**Broader Impact Concerns:**

No broader impact concerns

**Claims And Evidence:**

Yes

**Requested Changes:**

Major comments
1)I must say it seems hard to me to fulfil both assumptions 3 and 4 at the same time. How can one be sure that the L^2 approximation error can be small if the class of approximating functions s(.,.,.) satisfies the regularity properties imposed by (3)?  To convince the reader of this, one should show that the score function of a log-concave distribution meets the requirements imposed by Assumption 3, with the exception of the regularity on \theta, which I am ok with assuming. One sided Lipschitzianity of the score is granted by Log-concavity, but what about the Lipschitz upper bound? What about time regularity? And the Lipschitzianity assumption on the derivative of the score?
 Thus, I don't feel comfortable saying that Theorem 10 does not applies as such to log-concave distributions and that the title of the article is fully justified. It may be possible to verify these requirements, at least for some distributions other than the Gaussian, but it is not done here.
2)The table in section 4.2 comparing the findings of this article with other relevant contributions in the field is incomplete and not clear. First of all, it does not distinguish between result obtained that compare directly the law of the generative model with the data distribution and articles that make use of an early stopping rule, that is to say articles that compare the law of the algorithm with a smoothed version of the data. This distinction is  imporant. In fact, having a KL against smoothed data does NOT provide with a KL or W_2 bound against the data unless further (strong) assumptions are made. Moreover, even ignoring all contibutions that came after this paper first appeared, there seem to be some gaps. For example, the referenced paper by Conforti et al obtains results that are better than those of Chen et al without having to resort to early stopping. Moreover, it also contain a result that is equivalent to Benton et al. in the early stopping regime.
3) I don't understand Remark 7. I agree the assumption of the auxiliary process is new but its density is also unknown. It can be simulated easily, this I agree. However, given samples from data, even the time reversal process can be simulated by simulating the forward and changing time. So I see a clear advantage in using the auxiliary process if one can bypass the use of samples from the data. Samples from data are indeed required to train the neural network that approximates the score, so what is the real gain?
4) Assumption 1 seems to assume implicitly that there is a unique  optimal parameter \theta^*. It does not seem very realistic. Can the hypothesis be formulated differently to avoid this problem? Also, what is the significance of this hypothesis? Why is it needed? Since other papers in the field don't need it, the authors should clarify its role.
Minor comments
1) It seems to me that is not the process that has the representation (2) but rahter its marginal distributions.
2) Why not integrating directly the linear part in (11) in the Euler scheme?
3) I don’t understand the sentence at the beginning of sectionI 3.2 . I agree that the score may not be computed at t=T. However, by using the score at time T-\varepsilon one can easily define Y^{EM} at time T.
4) Pg 4: the process (9) instead of the process 9
5) Eq:10 it should be T-t_k instead of T-k
6) The notation for the auxiliary process is not the same in (9) to (20)
7) I would put the Gaussian result as example of application of theorem 10, emphasising that better constants can be obtained in this case, rather than a main result.

**Strengths And Weaknesses:**

Strength

-optimal dependence in the dimension of theoretical guarantees
-W_2 guarantees

Weaknesses
-Strong hypothesis hard to verify in practice

---

> ### Author Response · Authors · 2024-11-28
>
> Thank you for taking the time to review our submission and for your constructive feedback.
>
> We answer your main comments below.
>
> (1) We answer this point in the new Remark 8 (page 7-8) and in the new Appendix B (page 16), where we have added further details. We demonstrate that the conditions specified in Assumptions 3 and 4 are simultaneously satisfied for a strongly log-concave data distribution other than the Gaussian.
>
> (2) We agree with the reviewer about the previous table. We have created two tables for the results with and without early stopping making the distinction clearer. We also have added the result by Conforti et al. in Section 4.2 and in Table 2. In Table 3, the results in Chen et al. 2023a, Theorem 2.2 and Benton et al., Corollary 1 are stated for the smoothed version of the data, denoted by $\pi_{D}^{\epsilon} $ . Therefore for the comparison with our result, an additional error should be added to their bounds, as the distance between $\pi_{D}^{\epsilon} $ and $\pi_{D}$ scales with $\sqrt{M}$ in $W_2$ (see Appendix D.2).
>
> (3) Simulating the time-reversal process using the forward process and by changing time is considerably more challenging (in comparison to simulating the auxiliary process), as typically only the equality in distribution between the time-reversal and the forward process does not suffice. Simulating the auxiliary process (9) does not require samples from the data per se. Instead, it takes as input only the (random) estimator $\hat{\theta}$ of $\theta^{\star}$ and samples from the standard Gaussian distribution to initiate process.
>
> (4) Assumption 1 states that "$\theta^{\star}$ is a minimiser of the score-matching optimization problem (6)", since it is not known whether the optimization problem (6) is convex or non-convex. A footnote in the Assumption 1 has been added to clarify this point for the reader. Moreover, one observes that the current literature often assumes that the score-matching optimization problem (6) can be solved but offers no details or intuition for this. Assumption 1 outlines key properties that the optimization algorithm must possess to achieve improved upper bounds on the Wasserstein distance between the generative and target distributions. Most of the properties stated in Assumption 1 are satisfied by algorithms such as ADAM, SGD and variants widely used in practice.
>
> We answer your minor comments below:
>
> (1) The equation (2) is the solution of the Ornstein–Uhlenbeck process (1), where $\sigma_t Z_t = \sqrt{2} \int_0^t e^{-(t-s)} dB_s$, implying that $Z_t$ is an $M$-dimensional standard Gaussian random vector. For this reason, the representation (2) holds almost surely.
>
> (2) This is a typical Euler-Maruyama numerical scheme. We are not sure whether the reviewer is suggesting a numerical scheme with integrating factors?
>
> (3) We agree with the reviewer that the original statement was somewhat convoluted. We have revised it (page 6) to enhance clarity for the reader.
>
> (4) We are experiencing an issue with the format of eqref in the TMLR LaTeX environment and have noticed that other accepted papers have used ref instead, as we are currently doing.
>
> (5) We agree with the reviewer and we have changed to $T-t_k$ in equation (10) page 4.
>
> (6) The notation seems to be correct. Does the reviewer mean the fonts of "aux" are different in (9) and (20)? If so, this is because of the assumption environment, and it is stated before Assumption 4 that $Y^{\text{aux}}$ is the one defined in (9).
>
> (7) We have chosen to retain the motivating example at the beginning to guide the reader towards the main theorem.

---

### Author Response · Authors · 2024-11-28

We thank the reviewers for their insightful comments. We have uploaded a revised version of the paper with changes highlighted in red.

---

### Comment · Reviewer_3tvY · 2024-12-16
**On the use of the time-reversal process**

Dear authors,

Thank you for addressing my comments and remarks and improving the overall presentation.

About my main comment (3): I understand that simulating the auxiliary process does not require to sample from data. However, I don't see how one can avoid to sample from data when constructing the score approximation. So the real gain from using the auxiliary process is still unclear to me. About you comment: why is equality in distribution not good enough? If one samples a forward path then, by reverting its time in a pathwise way, he obtains sample from the time reversal. Why is this not good enough? Please clarify this.

---

> ### Author Response · Authors · 2024-12-17
>
> Thank you for your constructive comments.
>
> We agree with the reviewer that utilizing samples from the data is unavoidable for constructing  the score approximation. Thus, clearly, the use of sample data cannot be avoided.
>
> The importance of the auxiliary process (9) is understood through its role as the corresponding stochastic differential equation (SDE) to the generative algorithm, since the latter is the Euler numerical scheme of this SDE. Hence, the key role of the auxiliary process is to make transparent the relationship between the generative algorithm (an Euler numerical scheme) to its corresponding SDE, i.e. the auxiliary process (9) itself. This further allows us to unpack the relationship between the generative algorithm and the backward process $\widetilde{Y}_t  $ (4) due to the difference in their drift coefficients. Thus, it facilitates key elements of the convergence proof (see the upper bounds with the auxiliary process in Appendix C.2 at pages 18-20 and Appendix D.2 at pages 26-32).
>
> With regards to  the equality in distribution, one notes that in general such equality (between two variables) is sufficient for the generation of new samples. In this particular case though, as the reviewer points out,  a pathwise approach is required for the generation of (new) samples. This is not given by considering only equality in distribution, but it is rather facilitated by a (unique) strong pathwise solution of the time-reversed SDE and an a.s. equality with the corresponding solution of the forward process.

---

### Decision · Action_Editor_Eqjp · 2025-01-13

**Recommendation:** Accept with minor revision

**Comment:**

The main reason for the rejection of the paper was that its references were not up-to-date. I agree with this assessment and request that the authors expand and update their literature review accordingly.

The authors state that "A careful examination of [...] is still needed." I request that they provide additional details and clarification regarding this claim. Moreover, Conforti et al. also present results on early stopping, which should be acknowledged and discussed in the paper.

**Audience:**

The paper is a first step toward convergence bounds in $2$-Wasserstein distance for diffusion models, which should be of interest for TMLR's audience.

**Claims And Evidence:**

The present submission is interested in establishing error estimates for diffusion models. In contrast to previous works, it aims to establish $2$-Wasserstein bounds. To this end, the authors suppose that the target is stronlgy log-concave, condition that may be considered stringent, in particular in comparison to exisiting results on the subject.

---

> ### Author Response · Authors · 2025-02-08
>
> We thank the Action Editor and the reviewers for their comments.
>
> We have included nine additional references in the Introduction and the Related Work and Comparison section. We have acknowledged and discussed the early stopping result in Corollary 2.4 of Conforti et al. in the Related Work and Comparison section. Additionally, we have provided further details about the need for additional conditions on the score function in Chen et al. (2023a) and Benton et al. (2024) to ensure the uniqueness of the solution for the backward SDE.

---

> > ### Comment · Action_Editor_Eqjp · 2025-02-11
> >
> > I thank the authors for incorporating my recommendations.
> >
> > However, based on the included discussion, I believe that the results from Conforti et al. should be included in Table 3. Unlike Chen et al. and Benton et al., this work does not impose any conditions on the score while still presenting an improvement over Chen et al results.

---

> > > ### Author Response · Authors · 2025-02-12
> > >
> > > We thank the Action Editor for the additional suggestion. We have included the result from Conforti et al. in  Table 3 (page 13).

---

> > > > ### Comment · Action_Editor_Eqjp · 2025-02-12
> > > >
> > > > I thank the authors for their changes. However, from my understanding, Assumption 1.10 is automatically satisfied for the OU process. Did I missed something?

---

> > > > > ### Author Response · Authors · 2025-02-13
> > > > >
> > > > > We thank the Action Editor for the additional comment.
> > > > >
> > > > > We follow the notation of Cattiaux et al (2023) in the paragraph "Time reversal formula for a diffusion process." (pages 5-6). Hypothesis 1.10 is an assumption on the potential $U$ in the initial marginal measure $m(dx)=\exp(-U(x)) dx$ in the martingale problem defined in equation (1.11) for the forward process. The initial marginal measure $m(dx)=\exp(-U(x)) dx$ in equation (1.11) is the data distribution, (i.e.,  $X_0 \sim m$ in our notation) which corresponds to the distribution of the initial condition of the OU process.
> > > > > Point (ii) in Hypothesis 1.10 requires a coercivity condition on the gradient of the potential $U$ (or the gradient of $\log m$)  and Hypothesis 1.10 point (i) requires the potential $U$ (or  $\log m$) to be continuously differentiable.

---

> > > > > > ### Comment · Action_Editor_Eqjp · 2025-02-13
> > > > > >
> > > > > > I thank the authors for your response.
> > > > > >
> > > > > > However it seems to me that in  Cattiaux et al (2023), $U$ does not correspond to the potential of the data distribution but the stationary measure which here the Gaussian distribution and therefore Assumption 1.10 is satisfied.

---

> > > > > > > ### Author Response · Authors · 2025-02-14
> > > > > > >
> > > > > > > We thank the Action Editor for their response.
> > > > > > >
> > > > > > > We have modified Table 3 (page 13) to match the description in Table 3 of Corollary 2.4 of Conforti et al.